# Learning to Estimate Shapley Values with Vision Transformers

**Ian Covert**,[*] **Chanwoo Kim**[*] **& Su-In Lee**
Paul G. Allen School of Computer Science & Engineering
University of Washington
`{icovert,chanwkim,suinlee}@cs.washington.edu`

## Abstract

Transformers have become a default architecture in computer vision, but understanding what drives their predictions remains a challenging problem. Current explanation approaches rely on attention values or input gradients, but these provide a limited view of a model's dependencies. Shapley values offer a theoretically sound alternative, but their computational cost makes them impractical for large, high-dimensional models. In this work, we aim to make Shapley values practical for vision transformers (ViTs). To do so, we first leverage an attention masking approach to evaluate ViTs with partial information, and we then develop a procedure to generate Shapley value explanations via a separate, learned explainer model. Our experiments compare Shapley values to many baseline methods (e.g., attention rollout, GradCAM, LRP), and we find that our approach provides more accurate explanations than existing methods for ViTs.

## 1 Introduction

Transformers (Vaswani et al., 2017) were originally introduced for NLP, but in recent years they have been successfully adapted to a variety of other domains (Wang et al., 2020; Jumper et al., 2021). In computer vision, transformer-based models are now used for problems including image classification, object detection and semantic segmentation (Dosovitskiy et al., 2020; Touvron et al., 2021; Liu et al., 2021), and they achieve state-of-the-art performance in many tasks (Wortsman et al., 2022). The growing use of transformers in computer vision motivates the question of what drives their predictions: understanding a complex model's dependencies is an important problem in many applications, but the field has not settled on a solution for the transformer architecture.

Transformers are composed of alternating self-attention and fully-connected layers, where the self-attention operation associates attention values with every pair of tokens. In vision transformers (ViTs) (Dosovitskiy et al., 2020), the tokens represent non-overlapping image patches, typically a total of $14 \times 14 = 196$ patches each of size $16 \times 16$. It is intuitive to view attention values as indicators of feature importance (Abnar and Zuidema, 2020; Ethayarajh and Jurafsky, 2021), but interpreting transformer attention in this way is potentially misleading. Recent work has raised questions about the validity of attention as explanation (Serrano and Smith, 2019; Jain and Wallace, 2019; Chefer et al., 2021), arguing that it provides an incomplete picture of a model's dependence on each token.

If attention is not a reliable indicator of feature importance, then what is? We consider the perspective that transformers are no different from any other architecture, and that we can explain their predictions using model-agnostic approaches that are currently used for other architectures. Among these methods, Shapley values are a theoretically compelling approach with feature importance scores that are designed to satisfy many desirable properties (Shapley, 1953; Lundberg and Lee, 2017).

The main challenge for Shapley values in the transformer context is calculating them efficiently, because a naive calculation has exponential running time in the number of patches. If Shapley values are poorly approximated, they are unlikely to reflect a model's true dependencies, but calculating them with high accuracy is currently too slow to be practical. Thus, our work aims to make Shapley values practical for transformers, and for ViTs in particular. Our contributions include:

---

[*]Equal contribution.

1. We investigate several approaches for withholding input features from vision transformers, which is a key operation for computing Shapley values. We find that ViTs can accommodate missing image patches by masking attention values for held-out tokens, and that training with random masking is important for models to properly handle partial information.

2. We develop a learning-based approach to estimate Shapley values efficiently and accurately. Our approach involves fine-tuning an existing ViT using a loss function designed specifically for Shapley values (Jethani et al., 2021b), and we prove that our loss bounds the estimation error without requiring ground truth values during training. Once trained, our explainer model provides a significant speedup over methods like KernelSHAP (Lundberg and Lee, 2017).

3. Our experiments compare the Shapley value-based approach to several groups of competing methods: attention-based, gradient-based and removal-based explanations. We find that our approach provides the best overall performance, correctly identifying influential and non-influential patches for both target and non-target classes. We verify this using three image datasets, and the results are consistent across multiple metrics.

Overall, our work shows that Shapley values can be made practical for vision transformers, and that they provide a compelling alternative to current attention- and gradient-based approaches.

## 2 RELATED WORK

Understanding neural network predictions is a challenging problem that has been actively researched for the last decade (Simonyan et al., 2013; Zeiler and Fergus, 2014; Ribeiro et al., 2016). We focus on *feature attribution*, or identifying the specific input features that influence a prediction, but prior work has also considered other problems (Olah et al., 2017; Kim et al., 2018). The various techniques that have been developed can be grouped into several categories, which we describe below.

**Attention-based explanations** Transformers use self-attention to associate weights with each pair of tokens (Vaswani et al., 2017), and a natural idea is to assess which tokens receive the most attention (Clark et al., 2019; Rogers et al., 2020; Vig et al., 2020). There are several versions of this approach, including *attention rollout* and *attention flow* (Abnar and Zuidema, 2020), which analyze attention across multiple layers. Attention is a popular interpretation tool, but it is only one component in a sequence of nonlinear operations that provides an incomplete picture of a model's dependencies (Serrano and Smith, 2019; Jain and Wallace, 2019; Wiegreffe and Pinter, 2019), and direct usage of attention weights has not been shown to perform well in vision tasks (Chefer et al., 2021).

**Gradient-based methods** For other deep learning models such as CNNs, gradient-based explanations are a popular family of approaches. There are many variations on the idea of calculating input gradients, including methods that modify the input (e.g., SmoothGrad, IntGrad) (Smilkov et al., 2017; Sundararajan et al., 2017; Xu et al., 2020), operate at intermediate network layers (GradCAM) (Selvaraju et al., 2017), or design modified backpropagation rules (e.g., LRP, DeepLift) (Bach et al., 2015; Shrikumar et al., 2016; Chefer et al., 2021). Although they are efficient to compute for arbitrary network architectures, gradient-based explanations achieve mixed results in quantitative benchmarks, including for object localization and the removal of influential features (Petsiuk et al., 2018; Hooker et al., 2019; Saporta et al., 2021; Jethani et al., 2021b), and they have been shown to be insensitive to the randomization of model parameters (Adebayo et al., 2018).

**Removal-based explanations** Finally, *removal-based explanations* are those that quantify feature importance by explicitly withholding inputs from the model (Covert et al., 2021). For models that require all features to make predictions, several options for removing features include setting them to default values (Zeiler and Fergus, 2014), sampling replacement values (Agarwal and Nguyen, 2020) and blurring images (Fong and Vedaldi, 2017). These methods work with any model type, but they tend to be slow because they require making many predictions. The simplest approach of removing individual features (known as *leave-one-out*, Ethayarajh and Jurafsky 2021) is relatively fast, but the computational cost increases as we examine more feature subsets.

Shapley values (Shapley, 1953) are an influential approach within the removal-based explanation framework. By examining all feature subsets, they provide a nuanced view of each feature's influence and satisfy many desirable properties (Lundberg and Lee, 2017). They are approximated in practice using methods like TreeSHAP and KernelSHAP (Lundberg et al., 2020; Covert and Lee, 2021), but these approaches either are not applicable or do not scale to large ViTs. Recent work highlighted

a connection between attention flow and Shapley values (Ethayarajh and Jurafsky, 2021), but this approach is fundamentally different from SHAP (Lundberg and Lee, 2017): attention flow treats each feature's influence on the model as strictly additive, which is computationally convenient but fails to represent feature interactions. Our work instead focuses on the original formulation (Lundberg and Lee, 2017) and aims to make Shapley values based on feature removal practical for ViTs.

## 3 BACKGROUND

Here, we define notation used throughout the paper and briefly introduce Shapley values.

### 3.1 NOTATION

Our focus is vision transformers trained for classification tasks, where $\mathbf{x} \in \mathbb{R}^{224 \times 224 \times 3}$ denotes an image and $\mathbf{y} \in \{1, \ldots, K\}$ denotes the class. We write the image patches as $\mathbf{x} = (\mathbf{x}_1, \ldots, \mathbf{x}_d)$, where ViTs typically have $\mathbf{x}_i \in \mathbb{R}^{16 \times 16 \times 3}$ and $d = 196$. The model is given by $f(\mathbf{x}; \eta) \in [0, 1]^K$ and $f_y(\mathbf{x}; \eta) \in [0, 1]$ represents the probability for the $y$th class. Shapley values involve feature subsets, so we use $s \in \{0, 1\}^d$ to denote a subset of indices and $\mathbf{x}_s = \{\mathbf{x}_i : s_i = 1\}$ a subset of image patches. We also use $\mathbf{0}$ and $\mathbf{1} \in \mathbb{R}^d$ to denote vectors of zeros and ones, and $e_i \in \mathbb{R}^d$ is a vector with a one in the $i$th position and zeros elsewhere. Finally, bold symbols $\mathbf{x}, \mathbf{y}$ are random variables, $x, y$ are possible values, and $p(\mathbf{x}, \mathbf{y})$ denotes the data distribution.

### 3.2 SHAPLEY VALUES

Shapley values were developed in game theory for allocating credit in coalitional games (Shapley, 1953). A coalitional game is represented by a set function, and the value for each subset indicates the profit achieved when the corresponding players participate. Given a game with $d$ players, or a set function $v : \{0, 1\}^d \mapsto \mathbb{R}$, the Shapley values are denoted by $\phi_1(v), \ldots, \phi_d(v) \in \mathbb{R}$ for each player, and the value $\phi_i(v)$ for the $i$th player is defined as follows:

$$\phi_i(v) = \frac{1}{d} \sum_{s:s_i=0} \binom{d-1}{\mathbf{1}^\top s}^{-1} \big( v(s + e_i) - v(s) \big). \tag{1}$$

Intuitively, eq. (1) represents the change in profit from introducing the $i$th player, averaged across all possible subsets to which $i$ can be added. Shapley values are defined in this way to satisfy many reasonable properties: for example, the credits sum to the value when all players participate, players with equivalent contributions receive equal credit, and players with no contribution receive zero credit (Shapley, 1953). These properties make Shapley values attractive in many settings: they have been applied with coalitional games that represent a model's prediction given a subset of features (SHAP) (Štrumbelj and Kononenko, 2010; Lundberg and Lee, 2017), as well as several other use-cases in machine learning (Ghorbani and Zou, 2019; 2020; Covert et al., 2020).

There are two main challenges when using Shapley values to explain individual predictions (Chen et al., 2022). The first is properly withholding feature information, and we explore how to address this challenge in the ViT context (Section 4). The second is calculating Shapley values efficiently, because their computation scales exponentially with the number of inputs $d$. Traditionally, they are approximated using sampling-based estimators like KernelSHAP (Castro et al., 2009; Štrumbelj and Kononenko, 2010; Lundberg and Lee, 2017), but we build on a more efficient learning-based approach (FastSHAP) recently introduced by Jethani et al. (2021b) (Section 5).

## 4 EVALUATING VISION TRANSFORMERS WITH PARTIAL INFORMATION

The basic idea behind Shapley values, as well as other removal-based explanations (Covert et al., 2021), is to evaluate the model with partial feature information and analyze how a prediction changes. Most models need values for all the features to make predictions, so in practice we require a mechanism to represent feature removal. For example, we can set held-out image regions to zero, or we can average the prediction across randomly sampled replacement values.

With vision transformers, the options for removing features are slightly different. Recent work has demonstrated the robustness of ViTs to randomly zeroed pixel values (Naseer et al., 2021), but the self-attention operation enables a more elegant approach: we can simply ignore tokens for image patches we wish to remove (Jain et al., 2021). We achieve this by masking attention values at each self-attention layer, or setting them to a large negative value before applying the softmax operation

(see Appendix A). This resembles causal attention masking in transformer language models like GPT-3 (Brown et al., 2020), but we use masking for a different purpose. Alternatively, we could use a unique token value as in masked language models such as BERT (Devlin et al., 2018), which would involve simply setting held-out tokens to the mask value.

Using this attention masking approach, we can evaluate a ViT model $f(\mathbf{x}; \eta)$ given subsets of image patches, denoted by $\mathbf{x_s}$. However, because these partial inputs represent off-manifold examples, the predictions with partial information may not behave as desired. We have two options to correct this: 1) we can ensure that the model is trained with random masking, or 2) we can fine-tune the model to encourage sensible behavior with missing patches. The first option is more direct, but it does not allow us to explain models trained without masking. For the latter option, we can create an updated model denoted by $g(\mathbf{x_s}; \beta)$ that we fine-tune using the following loss,

$$\min_{\beta} \quad \mathbb{E}_{p(\mathbf{x})} \mathbb{E}_{p(\mathbf{s})} \Big[ D_{\mathrm{KL}}\big(f(\mathbf{x}; \eta) \, || \, g(\mathbf{x_s}; \beta)\big)\Big], \tag{2}$$

where $p(\mathbf{s})$ is a distribution over subsets. In practice, we sample the cardinality $\mathbf{m} = \mathbf{1}^\top \mathbf{s}$ from $\mathbf{m} \sim \mathrm{Unif}(0, d)$ and then sample $\mathbf{m}$ patches uniformly at random. Intuitively, eq. (2) encourages $g(\mathbf{x_s}; \beta)$ to preserve the original model's predictions even with missing features. We use this loss because it satisfies the desirable property that the optimal model $g(\mathbf{x_s}; \beta^*)$ outputs the expected prediction given the available information (Covert et al., 2021), or

$$g(x_s; \beta^*) = \mathbb{E}[f(\mathbf{x}; \eta) \mid \mathbf{x}_s = x_s]. \tag{3}$$

Note that this represents a best-effort prediction, because if $f(\mathbf{x}; \eta) = p(\mathbf{y} \mid \mathbf{x})$ then we have $g(\mathbf{x_s}; \beta^*) = p(\mathbf{y} \mid \mathbf{x}_s)$. Similarly, in the case where $f(\mathbf{x}_s; \eta)$ is trained directly with random masking, the training process estimates $f(\mathbf{x}_s; \eta) \approx p(\mathbf{y} \mid \mathbf{x}_s)$ (see Appendix B). We refer to the fine-tuned model $g(\mathbf{x_s}; \beta)$ as a *surrogate*, following the naming in prior work (Frye et al., 2020). Whether we use the original model or a version fine-tuned with random masking, our attention masking approach enables us to probe how individual predictions change as we remove groups of image patches.

## 5 LEARNING TO ESTIMATE SHAPLEY VALUES

Given our approach for evaluating ViTs with partial information, we can use Shapley values to identify influential image patches for an input $x$ and class $y$. This involves evaluating the model with many feature subsets $x_s$, so we define a coalitional game $v_{xy}(s) = g_y(x_s; \beta)$. Alternatively, if we use a model trained with masking, we can define the coalitional game as $v_{xy}(s) = f_y(x_s; \eta)$. Common Shapley value approximations are based on sampling feature permutations (Castro et al., 2009; Štrumbelj and Kononenko, 2010) or fitting a weighted least squares model (Lundberg and Lee, 2017; Covert and Lee, 2021), but these can require hundreds or thousands of model evaluations to explain a single prediction.[1] Instead, we develop a learning-based estimation approach for ViTs.

Our goal is to obtain an explainer model that estimates Shapley values directly. To do so, we train a new vision transformer $\phi_{\mathrm{ViT}}(\mathbf{x}, \mathbf{y}; \theta) \in \mathbb{R}^d$ that outputs approximate Shapley values for an input-output pair $(x, y)$ in a single forward pass. Crucially, rather than training the model using a dataset of ground truth Shapley value explanations, we train it by minimizing the following objective,

$$\mathcal{L}(\theta) = \mathbb{E}_{p(\mathbf{x}, \mathbf{y})} \mathbb{E}_{p_{\mathrm{Sh}}(\mathbf{s})} \Big[ \big(v_{\mathbf{xy}}(\mathbf{s}) - v_{\mathbf{xy}}(\mathbf{0}) - \mathbf{s}^\top \phi_{\mathrm{ViT}}(\mathbf{x}, \mathbf{y}; \theta)\big)^2 \Big] \tag{4}$$

$$\text{s.t.} \quad \mathbf{1}^\top \phi_{\mathrm{ViT}}(x, y; \theta) = v_{xy}(\mathbf{1}) - v_{xy}(\mathbf{0}) \quad \forall \, (x, y),$$

where $p_{\mathrm{Sh}}(\mathbf{s})$ is a distribution defined as $p_{\mathrm{Sh}}(s) \propto (\mathbf{1}^\top s - 1)!(d - \mathbf{1}^\top s - 1)!$ for $0 < \mathbf{1}^\top s < d$ and $p_{\mathrm{Sh}}(\mathbf{1}) = p_{\mathrm{Sh}}(\mathbf{0}) = 0$. Intuitively, eq. (4) encourages the explainer model to output feature scores that provide an additive approximation for the predictions with partial information, where the predictions are represented by $v_{\mathbf{xy}}(\mathbf{s})$ and the additive approximation by $v_{\mathbf{xy}}(\mathbf{0}) + \mathbf{s}^\top \phi_{\mathrm{ViT}}(\mathbf{x}, \mathbf{y}; \theta)$.

The loss in eq. (4) was introduced by Jethani et al. (2021b) and is derived from an optimization-based characterization of the Shapley value (Charnes et al., 1988). To rigorously justify this training approach, we derive new results that show how this objective controls the Shapley value estimation error. Proofs are in Appendix D. First, we show that the explainer's loss for a single input is strongly convex in the prediction, a result that implies the existence of unique optimal predictions.

---

[1] The number of model evaluations depends on how fast the estimators converge, and we find that KernelSHAP requires >100,000 samples to converge for ViTs (Appendix H).

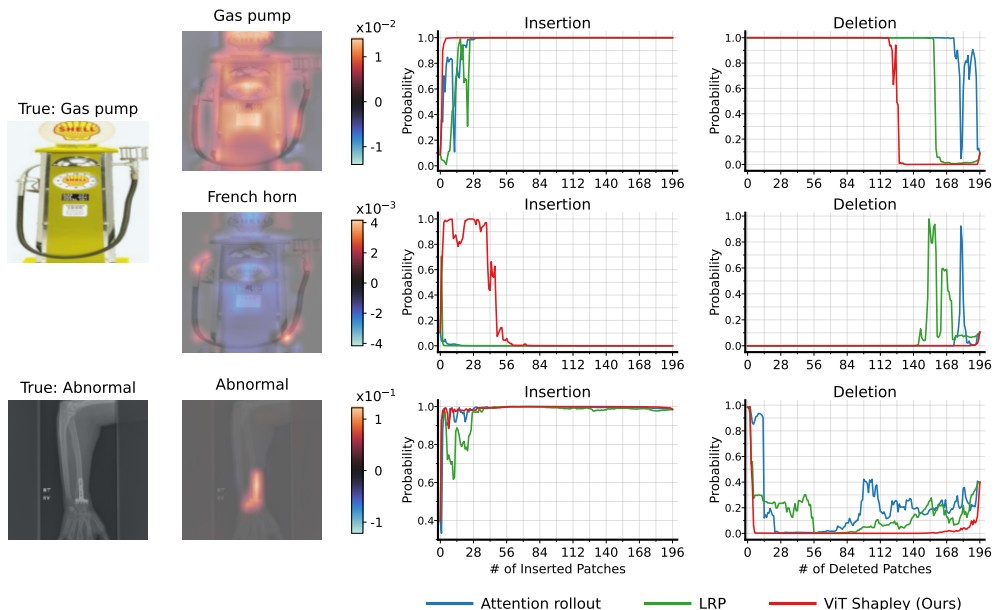

Figure 1: Explanations where our approach identifies relevant information for target and non-target classes. **Left:** original images from the ImageNette and MURA datasets. **Middle left:** explanations generated by ViT Shapley for specific classes. **Right:** probability of the class being explained after the insertion or deletion of important patches (higher is better for insertion, lower for deletion).

**Lemma 1.** *For a single input-output pair $(x, y)$, the expected loss under eq.* (4) *for the prediction $\phi_{ViT}(x, y; \theta)$ is $\mu$-strongly convex with $\mu = H_{d-1}^{-1}$, where $H_{d-1}$ is the $(d-1)$th harmonic number.*

Next, we utilize the strong convexity property from Lemma 1 to prove our main result: that the explainer model's loss function upper bounds the distance between the exact and approximated Shapley values. This is notable because we do not utilize ground truth values during training.

**Theorem 1.** *For a model $\phi_{ViT}(\mathbf{x}, \mathbf{y}; \theta)$ whose predictions satisfy the constraint in eq.* (4)*, the objective value $\mathcal{L}(\theta)$ upper bounds the Shapley value estimation error as follows,*

$$\mathbb{E}_{p(\mathbf{x},\mathbf{y})}\left[\left|\left|\phi_{ViT}(\mathbf{x}, \mathbf{y}; \theta) - \phi(v_{\mathbf{xy}})\right|\right|_2\right] \leq \sqrt{2H_{d-1}\Big(\mathcal{L}(\theta) - \mathcal{L}^*\Big)},$$

*where $\mathcal{L}^*$ represents the loss achieved by the exact Shapley values.*

This shows that our objective is a viable approach for training without exact Shapley values, because optimizing eq. (4) minimizes an upper bound on the estimation error. In other words, if we can iteratively optimize the explainer model so that its loss approaches the optimum obtained by the exact Shapley values ($\mathcal{L}(\theta) \to \mathcal{L}^*$), our estimation error will go to zero.

In practice, we train the explainer model $\phi_{ViT}(\mathbf{x}, \mathbf{y}; \theta)$ using stochastic gradient descent, and several other steps are important during training. First, we normalize the explainer's unconstrained predictions in order to satisfy the objective's constraint in eq. (4); this ensures that the Shapley value's *efficiency* property holds (Shapley, 1953). Next, rather than training the explainer from scratch, we fine-tune an existing model that can be either the original classifier or a ViT pre-trained on a different supervised or self-supervised learning task (Touvron et al., 2021; He et al., 2021); ViTs are more difficult to train than convolutional networks, and we find that fine-tuning is important to train the explainer effectively (Table 3). Finally, we simplify the architecture by estimating Shapley values for all classes simultaneously. Our training approach is described in more detail in Appendix C.

By using a ViT to estimate Shapley values, we model the true explanation function and learn rich representations that capture not only which class is represented, but where key information is located. And by fine-tuning an existing model, we allow the explainer to re-use visual features that were informative for other challenging tasks. Ultimately, the explainer cannot guarantee exact Shapley values, but no approximation algorithm can; instead, it offers a favorable trade-off between accuracy and efficiency, and we find empirically that this approach offers a powerful alternative to the methods currently used for ViTs.

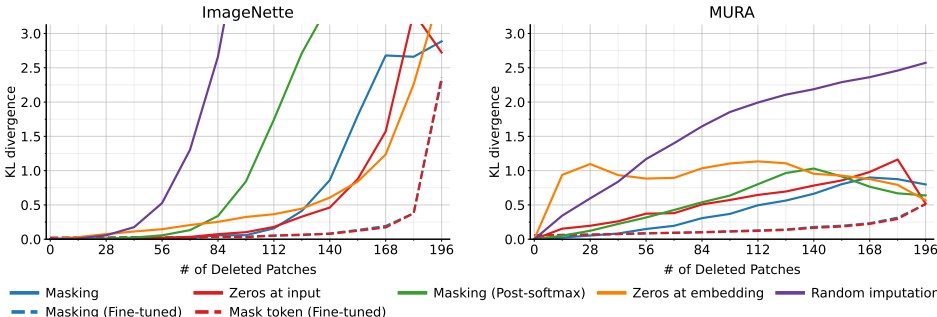

Figure 2: ViT predictions given partial information. We delete patches at random using several removal mechanisms, and then measure the quality of the resulting predictions via the KL divergence relative to the original, full-image predictions (lower is better).

# 6 EXPERIMENTS

We now demonstrate the effectiveness of our approach, termed *ViT Shapley*.[2] First, we evaluate attention masking for handling held-out patches in ViTs (Section 6.1). Next, we compare explanations from ViT Shapley to several existing methods (Section 6.2). Our baselines include attention-, gradient- and removal-based explanations, and we compare these methods via several metrics for explanation quality, including insertion/deletion of important features (Petsiuk et al., 2018), sensitivity-n (Ancona et al., 2018), faithfulness (Bhatt et al., 2021) and ROAR (Hooker et al., 2019).

Our experiments are based on three image datasets: ImageNette, a natural image dataset consisting of ten ImageNet classes (Howard and Gugger, 2020; Deng et al., 2009), MURA, a medical image dataset of musculoskeletal radiographs classified as normal or abnormal (Rajpurkar et al., 2017), and the Oxford IIIT-Pets dataset, which has 37 classes (Parkhi et al., 2012). See Figure 1 for example images. The main text shows results for ImageNette and MURA, and Pets results are in Appendix H. We use ViT-Base models (Wightman, 2019) as classifiers for all datasets, unless otherwise specified.

## 6.1 EVALUATING IMAGE PATCH REMOVAL

Our initial experiments test whether attention masking is effective for handling held-out image patches. We fine-tuned the classifiers for each dataset following the procedure described in Section 4, and we also tested several approaches without performing any fine-tuning: attention masking, attention masking applied after the softmax operation (how dropout is often implemented for ViTs, Wightman 2019), setting input patches to zero (Naseer et al., 2021), setting token embeddings to zero, and replacing with random patches from the dataset. Finally, we performed identical fine-tuning while replacing input patches with zeros, which is equivalent to introducing a fixed mask token.

As a measure of how well missing patches are handled, we calculated the KL divergence relative to the full-image predictions as random patches are removed. This can be interpreted as a divergence measure between the masked predictions and the predictions with patches marginalized out (see Appendix B), or how close we are to correctly removing patch information. The metric is calculated with randomly generated patch subsets, and it represents whether the model makes reasonable predictions given partial inputs. Similar results for top-1 accuracy are in Appendix H.

Figure 2 shows the results. Most methods perform well with <25% of patches missing, leading to only small increases in KL divergence. This is especially true for ImageNette, where large objects make the model more robust to missing patches. However, the methods with no fine-tuning begin to diverge as larger numbers of patches are removed and the partial inputs become increasingly off-manifold. Thus, fine-tuning becomes necessary to properly account for partial inputs as more patches are removed.

For all datasets, we find that fine-tuning with either attention masking or input patches set to zero provide comparable performance, and that these perform best across all numbers of patches. This means that fine-tuning makes attention masking significantly more effective for marginalizing out missing patches, and these results suggest that training ViTs with held-out tokens may be necessary to enable robustness to partial information. As prior work suggests, properly handling held-out

---

[2]https://github.com/suinleelab/vit-shapley

Table 1: Evaluating ViT Shapley using standard explanation metrics, with explanations calculated for the target class only. Methods that fail to outperform the random baseline are shown in gray, and the best results are shown in bold (accounting for 95% confidence intervals).

| | ImageNette | | | MURA | | |
|---|---|---|---|---|---|---|
| | Ins. ($\uparrow$) | Del. ($\downarrow$) | Faith. ($\uparrow$) | Ins. ($\uparrow$) | Del. ($\downarrow$) | Faith. ($\uparrow$) |
| Attention last | 0.962 (0.004) | 0.793 (0.013) | **0.694 (0.015)** | 0.890 (0.010) | 0.592 (0.013) | 0.635 (0.016) |
| Attention rollout | 0.938 (0.005) | 0.880 (0.010) | **0.704 (0.015)** | 0.845 (0.011) | 0.692 (0.014) | 0.618 (0.016) |
| GradCAM | 0.914 (0.006) | 0.937 (0.008) | 0.680 (0.015) | 0.899 (0.009) | 0.681 (0.015) | 0.631 (0.016) |
| IntGrad | 0.967 (0.004) | 0.930 (0.008) | 0.403 (0.024) | 0.897 (0.010) | 0.796 (0.015) | 0.201 (0.022) |
| Vanilla | 0.950 (0.004) | 0.808 (0.013) | **0.703 (0.015)** | 0.890 (0.010) | 0.537 (0.014) | 0.629 (0.016) |
| SmoothGrad | 0.947 (0.005) | 0.942 (0.006) | **0.703 (0.015)** | 0.870 (0.010) | 0.813 (0.011) | 0.617 (0.016) |
| VarGrad | 0.949 (0.005) | 0.946 (0.005) | **0.700 (0.015)** | 0.857 (0.011) | 0.823 (0.011) | 0.615 (0.016) |
| LRP | 0.967 (0.004) | 0.779 (0.014) | **0.705 (0.015)** | 0.900 (0.009) | 0.551 (0.013) | 0.646 (0.016) |
| Leave-one-out | 0.969 (0.002) | 0.917 (0.010) | 0.140 (0.040) | 0.926 (0.008) | 0.694 (0.017) | 0.308 (0.032) |
| RISE | 0.977 (0.001) | 0.860 (0.014) | **0.704 (0.015)** | 0.957 (0.004) | 0.573 (0.018) | 0.618 (0.016) |
| **ViT Shapley** | **0.985 (0.002)** | **0.691 (0.014)** | **0.711 (0.015)** | **0.971 (0.002)** | **0.307 (0.013)** | **0.707 (0.013)** |
| Random | 0.951 (0.005) | 0.951 (0.005) | - | 0.849 (0.010) | 0.847 (0.010) | - |

information is crucial for generating informative explanations (Frye et al., 2020; Covert et al., 2021), so the remainder of our experiments proceed with the fine-tuned attention masking approach.

## 6.2 Evaluating explanation accuracy

Next, we implemented ViT Shapley by training explainer models for both datasets. We used the fine-tuned classifiers from Section 6.1 to handle partial information, and we used the ViT-Base architecture with extra output layers to generate Shapley values for all patches. The explainer models were trained by optimizing eq. (4) using stochastic gradient descent (see details in Appendix C), and once trained, the explainer outputs approximate Shapley values in a single forward pass (Figure 1).

As comparisons for ViT Shapley, we considered a large number of baselines. For attention-based methods, we use attention rollout and the last layer's attention directed to the class token (Abnar and Zuidema, 2020). Similar to prior work (Chefer et al., 2021), we did not use attention flow due to the computational cost. Next, for gradient-based methods, we use Vanilla Gradients (Simonyan et al., 2013), IntGrad (Sundararajan et al., 2017), SmoothGrad (Smilkov et al., 2017), VarGrad (Hooker et al., 2019), LRP (Chefer et al., 2021) and GradCAM (Selvaraju et al., 2017). For removal-based methods, we use the leave-one-out approach (Zeiler and Fergus, 2014) and RISE (Petsiuk et al., 2018). Appendix F describes the baselines in more detail, including how several were modified to provide patch-level results, and Appendix H shows the running time for each method.

Given our set of baselines, we used several metrics to evaluate ViT Shapley. Evaluating explanation accuracy is difficult when the true importance is not known a priori, so we rely on metrics that test how removing (un)important features affects a model's predictions. Intuitively, removing influential features for a particular class should reduce the class probability, and removing non-influential features should not affect or even increase the class probability. Removal-based explanations are implicitly related to such metrics (Covert et al., 2021), but attention- and gradient-based methods may be hoped to provide strong performance with lower computational cost.

First, we implemented the widely used *insertion* and *deletion* metrics (Petsiuk et al., 2018). For these, we generate predictions while inserting/removing features in order of most to least important, and we then evaluate the area under the curve of prediction probabilities (see Figure 1). Here, we average the results across 1,000 images for their true class. We use random test set images for ImageNette, and for MURA we use test examples that were classified as abnormal because these are more important in practice. When removing information, we use the fine-tuned classifier because this represents the closest approximation to properly removing information from the model (Section 4).

Table 1 displays the results, and we find that ViT Shapley offers the best performance on both datasets. RISE and LRP tend to be the most competitive baselines, and perhaps surprisingly, certain other methods fail to outperform a random baseline (GradCAM, SmoothGrad, VarGrad). The baselines are sometimes competitive with ViT Shapley on insertion, but the gap for the deletion metric is larger.

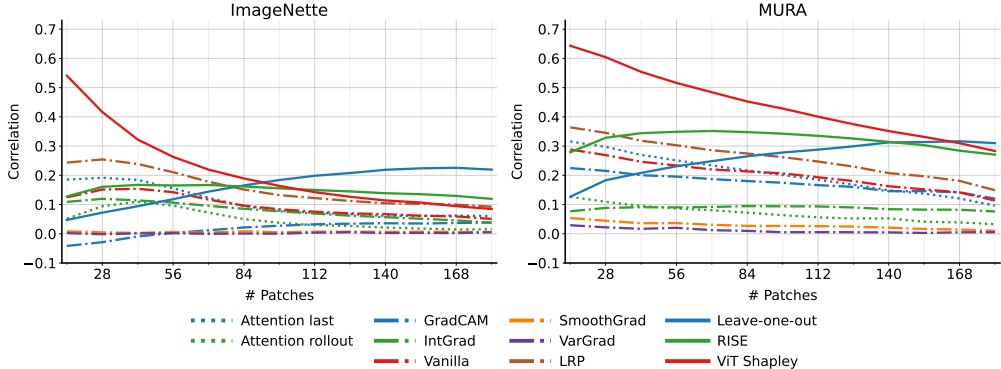

Figure 3: Sensitivity-n evaluation for different subset sizes. The metric is generated separately for a range of subset sizes, whereas faithfulness is calculated jointly over subsets of all sizes.

Practically, this means that ViT Shapley identifies important features that quickly drive the prediction towards a given class, and that quickly reduce the prediction probability when deleted.

Next, we modified these metrics to address a common issue with model explanations: that their results are not specific to each class (Rudin, 2019). ViT Shapley produces separate explanations for each class, so it can identify relevant patches even for non-target classes (see Figure 1). Table 2 shows insertion/deletion results averaged across all non-target classes for ImageNette. The attention-based methods do not produce class-specific explanations, and the remaining baselines generally provide poor results. Empirically, this is because the explanations are often similar across classes (see Appendix I). ViT Shapley performs best, particularly on insertion, and RISE is the best-performing baseline. Appendix H shows results for MURA, as well as the curves used to calculate these results.

Table 2: Evaluating ViT Shapley for explaining non-target classes. Methods that fail to outperform the random baseline are shown in gray, and the best results are shown in bold (accounting for 95% confidence intervals).

|  | ImageNette | | |
| --- | --- | --- | --- |
|  | Ins. ($\uparrow$) | Del. ($\downarrow$) | Faith. ($\uparrow$) |
| Attention last | - | - | - |
| Attention rollout | - | - | - |
| GradCAM | 0.021 (0.002) | 0.005 (0.000) | -0.672 (0.015) |
| IntGrad | 0.008 (0.001) | 0.004 (0.000) | 0.294 (0.022) |
| Vanilla | 0.006 (0.001) | 0.020 (0.001) | -0.682 (0.015) |
| SmoothGrad | 0.006 (0.001) | 0.006 (0.001) | -0.683 (0.015) |
| VarGrad | 0.006 (0.001) | 0.006 (0.001) | -0.680 (0.015) |
| LRP | 0.004 (0.001) | 0.022 (0.001) | -0.680 (0.015) |
| Leave-one-out | 0.013 (0.002) | 0.003 (0.000) | -0.017 (0.028) |
| RISE | 0.023 (0.003) | 0.002 (0.000) | -0.681 (0.015) |
| **ViT Shapley** | **0.093 (0.004)** | **0.001 (0.000)** | **0.672 (0.014)** |
| Random | 0.005 (0.001) | 0.005 (0.001) | - |

The insertion and deletion metrics only test importance rankings, so we require other metrics to test the specific attribution values. *Sensitivity-n* (Ancona et al., 2018) was proposed for this purpose, and it measures whether attributions correlate with the impact on a model's prediction when a feature is removed. The correlation is typically calculated across subsets of a fixed size, and then averaged across many predictions. *Faithfulness* (Bhatt et al., 2021) is a similar metric where the correlation is calculated across subsets of all sizes.

Table 1 and Table 2 show faithfulness results. Among the baselines, RISE and LRP remain most competitive, but ViT Shapley again performs best for both datasets. Figure 3 shows sensitivity-n results calculated across a range of subset sizes. Leave-one-out naturally performs best for large subset sizes, but ViT Shapley performs the best overall, particularly with smaller subsets. The sensitivity-n results focus on the target class, but Appendix H shows results for non-target classes where ViT Shapley's advantage over many baselines (including LRP) is even larger.

Finally, we performed an evaluation inspired by ROAR (Hooker et al., 2019), which tests how a model's accuracy degrades as important features are removed. ROAR suggests retraining with masked inputs, but this is unnecessary here because the fine-tuned classifier is designed to handle held-out patches. We therefore generated multiple versions of the metric. First, we evaluated accuracy while using the fine-tuned classifier to handle masked patches. Second, we repeated the evaluation using a separate evaluator model trained directly with held-out patches, similar to EVAL-X (Jethani et al.,

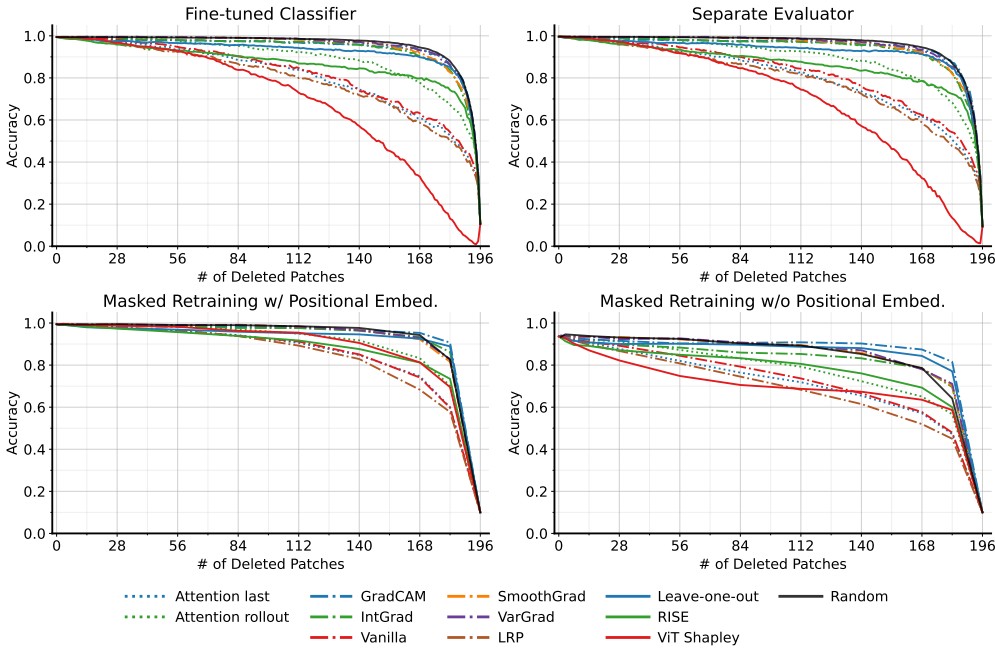

Figure 4: ImageNette accuracy when removing features in order of their importance, run with four evaluation strategies: fine-tuned classifier, separate evaluator, masked retraining, and masked retraining without positional embeddings.

2021a). Third, we performed masked retraining as described by ROAR. The first version represents the original classifier's best-effort prediction, and the second is a best-effort prediction disconnected from the original model; masked retraining is similar, but the retrained model can exploit information communicated by the masking, such as the shape and position of the removed object.

Figure 4 shows the results when removing important patches. ViT Shapley consistently outperforms the baselines across the first two versions of the metric, yielding faster degradation when important patches are removed. ViT Shapley also performs best when inserting important patches, yielding a faster increase in accuracy (Appendix H). It is outperformed by several baselines with masked retraining in the deletion direction (Figure 4 bottom left), but we find that this is likely due to spatial information leaked by ViT Shapley's deleted patches; indeed, when we retrained *without positional embeddings*, we found that ViT Shapley achieved the fastest degradation with a small number of deleted patches (Figure 4 bottom right). Interestingly, positional embeddings in ViTs offer a unique approach to alleviate ROAR's known information leakage issue (Jethani et al., 2021a).

In addition to these experiments, we include many further results in the supplement (Appendix H). First, we observe similar benefits for ViT Shapley when using the Oxford-IIIT Pets dataset. Next, regarding the choice of architecture, we observe consistent results when replacing ViT-Base with ViT-Tiny, -Small or -Large. We also replicate our results when using a classifier trained directly with random masking, an approach discussed in prior work to accommodate partial input information (Covert et al., 2021). We then generated metrics comparing ViT Shapley's approximation quality with KernelSHAP (Lundberg and Lee, 2017), and we found that ViT Shapley's accuracy is equivalent to running KernelSHAP for roughly 120,000 model evaluations (Appendix H). Lastly, we provide qualitative examples in Appendix I, including comparisons with the baselines. Overall, these results show that ViT Shapley is a practical and effective approach for explaining ViT predictions.

## 7 CONCLUSION

In this work, we developed a learning-based approach to generate Shapley values for ViTs. Our approach involves training a separate explainer model using an objective that does not require ground truth supervision, and ViT Shapley outperforms a variety of attention- and gradient-based methods across a range of accuracy metrics. Our experiments provide empirical support for Shapley values as an alternative to attention-based approaches, which remain popular for transformer models. Future directions involve extending ViT Shapley to NLP models, operating with arbitrary token groups or superpixels, and accelerating or otherwise improving the explainer model's training.

## Acknowledgements

We thank Mukund Sudarshan, Neil Jethani, Chester Holtz and the Lee Lab for helpful discussions. This work was funded by NSF DBI-1552309 and DBI-1759487, NIH R35-GM-128638 and R01-NIA-AG-061132.

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

# A    ATTENTION MASKING

This section describes our attention masking approach in detail. First, recall that ViTs use query-key-value self-attention (Vaswani et al., 2017; Dosovitskiy et al., 2020), which accepts a set of input tokens and produces a weighted sum of learned token values. Given an input $\mathbf{z} \in \mathbb{R}^{d \times h}$ and parameters $U_{qkv} \in \mathbb{R}^{h \times 3h'}$, we compute the self attention output $\text{SA}(\mathbf{z})$ for a single head as follows:

$$[\mathbf{Q}, \mathbf{K}, \mathbf{V}] = \mathbf{z} U_{qkv} \tag{5}$$

$$\mathbf{A} = \text{softmax}(\mathbf{Q}\mathbf{K}^\top / \sqrt{h'}) \tag{6}$$

$$\text{SA}(\mathbf{z}) = \mathbf{A}\mathbf{V}. \tag{7}$$

In multihead self-attention, we perform this operation in parallel over $k$ attention heads and project the concatenated outputs. Denoting each head's output as $\text{SA}_i(\mathbf{z})$ and the projection matrix as $U_{msa} \in \mathbb{R}^{k \cdot h' \times h}$, the multihead self-attention output $\text{MSA}(\mathbf{z})$ is

$$\text{MSA}(\mathbf{z}) = [\text{SA}_1(\mathbf{z}), \dots, \text{SA}_k(\mathbf{z})] U_{msa}. \tag{8}$$

Multihead self-attention can operate with any number of tokens, so given a subset $\mathbf{s} \in \{0, 1\}^d$ and an input $\mathbf{x}$, we can evaluate a ViT using only tokens for the patches $\mathbf{x_s} = \{\mathbf{x}_i : s_i = 1\}$. However, for implementation purposes it is preferable to maintain the same number of tokens within a minibatch. We therefore provide all tokens to the model and achieve the same effect using attention masking. Our exact approach is described below.

Let $\mathbf{z} \in \mathbb{R}^{d \times h}$ represent the full token set for an input $\mathbf{x}$ and let $\mathbf{s}$ be a subset. At each self-attention layer, we construct a mask matrix $\mathbf{S} = [\mathbf{s}, \dots, \mathbf{s}]^\top \in \{0, 1\}^{d \times d}$ and calculate the masked self-attention output $\text{SA}(\mathbf{z}, \mathbf{s})$ as follows:

$$\mathbf{A} = \text{softmax}((\mathbf{Q}\mathbf{K}^\top - (1 - \mathbf{S}) \cdot \infty) / \sqrt{h'}) \tag{9}$$

$$\text{SA}(\mathbf{z}, \mathbf{s}) = \mathbf{A}\mathbf{V}. \tag{10}$$

The masked multihead self-attention output is then calculated similarly to the original version:

$$\text{MSA}(\mathbf{z}, \mathbf{s}) = [\text{SA}_1(\mathbf{z}, \mathbf{s}), \dots, \text{SA}_k(\mathbf{z}, \mathbf{s})] U_{msa}. \tag{11}$$

Due to the masking in eq. (9), each output token in $\text{MSA}(\mathbf{z}, \mathbf{s})$ is guaranteed not to attend to tokens from $\mathbf{x_{1-s}} = \{\mathbf{x}_i : s_i = 0\}$. We use masked self-attention in all layers of the network, so that the tokens for $\mathbf{x_s}$ remain invariant to those for $\mathbf{x_{1-s}}$ throughout the entire model, including after the layer norm and fully-connected layers. When the final prediction is calculated using the class token, the output is equivalent to using only the tokens for $\mathbf{x_s}$. If the final prediction is instead produced using global average pooling (Beyer et al., 2022), we can modify the average to account only for tokens we wish to include.

# B    MASKED TRAINING

In this section, we provide proofs to justify training a ViT classifier with held-out tokens, either as part of the original training or as part of a post-hoc fine-tuning procedure (the surrogate model training described in Section 4). Our proofs are similar to those in prior work that discusses marginalizing out features using their conditional distribution (Covert et al., 2021).

First, consider a model trained directly with masking. Given a subset distribution $p(\mathbf{s})$ and the data distribution $p(\mathbf{x}, \mathbf{y})$, we can train a model $f(\mathbf{x_s}; \eta)$ with cross-entropy loss and random masking by minimizing the following:

$$\min_{\eta} \quad \mathbb{E}_{p(\mathbf{x},\mathbf{y})} \mathbb{E}_{p(\mathbf{s})} [-\log f_{\mathbf{y}}(\mathbf{x_s}; \eta)]. \tag{12}$$

To understand the global optimizer for this loss function, consider the expected loss for the prediction given a fixed model input $x_s$:

$$\mathbb{E}_{p(\mathbf{y}, \mathbf{x_{1-s}}|x_s)}[-\log f_{\mathbf{y}}(x_s; \eta)] = \mathbb{E}_{p(\mathbf{y}|x_s)}[-\log f_{\mathbf{y}}(x_s; \eta)]. \tag{13}$$

The expression in eq. (13) is equal to the KL divergence $D_{\mathrm{KL}}(p(\mathbf{y} \mid x_s) \mid\mid f(x_s; \eta))$ up to a constant value, so the prediction that minimizes this loss is $p(\mathbf{y} \mid x_s)$. For any subset $s \in \{0, 1\}^d$ where $p(s) > 0$, we then have the following result for the model $f(\mathbf{x_s}; \eta^*)$ that minimizes eq. (12):

$$f_y(x_s; \eta^*) = p(y \mid x_s) \text{ a.e. in } p(\mathbf{x}).$$

Intuitively, this means that training the original model with masking estimates $f(\mathbf{x_s}; \eta) \approx p(\mathbf{y} \mid \mathbf{x_s})$. In practice, we use a subset distribution $p(\mathbf{s})$ where $p(\mathbf{s}) > 0$ for all $s \in \{0, 1\}^d$: we set $p(\mathbf{s})$ by sampling the cardinality uniformly at random and then sampling the members, which is equivalent to defining $p(\mathbf{s})$ as

$$p(s) = \frac{1}{\binom{d}{\mathbf{1}^\top s} \cdot (d + 1)}.$$

Alternatively, we can use a model $f(\mathbf{x}; \eta)$ trained without masking and fine-tune it to better handle held-out features. In our case, this yields a *surrogate model* (Frye et al., 2020) denoted as $g(\mathbf{x_s}; \beta)$ that we fine-tune by minimizing the following loss:

$$\min_{\beta} \quad \mathbb{E}_{p(\mathbf{x})}\mathbb{E}_{p(\mathbf{s})}\Big[D_{\mathrm{KL}}\big(f(\mathbf{x}; \eta) \mid\mid g(\mathbf{x_s}; \beta)\big)\Big]. \tag{14}$$

To understand the global optimizer for the above loss, we can again consider the expected loss given a fixed input $x_s$:

$$\mathbb{E}_{p(\mathbf{x_{1-s}}|x_s)}\Big[D_{\mathrm{KL}}\big(f(\mathbf{x}; \eta) \mid\mid g(x_s; \beta)\big)\Big] = D_{\mathrm{KL}}\big(\mathbb{E}[f(\mathbf{x}; \eta) \mid x_s] \mid\mid g(x_s; \beta)\big) + \text{const.} \tag{15}$$

The distribution that minimizes this loss is the expected output given the available features, or $\mathbb{E}[f(\mathbf{x}; \eta) \mid x_s]$. By the same argument presented above, we then have the following result for the optimal surrogate $g(\mathbf{x_s}; \beta^*)$ that minimizes eq. (14):

$$g(x_s; \beta^*) = \mathbb{E}[f(\mathbf{x}; \eta) \mid x_s] \text{ a.e. in } p(\mathbf{x}).$$

Notice that if the initial model is optimal, or $f(\mathbf{x}; \eta) = p(\mathbf{y} \mid \mathbf{x})$, then the optimal surrogate satisfies $g(\mathbf{x_s}; \beta^*) = p(\mathbf{y} \mid \mathbf{x_s})$.

## C EXPLAINER TRAINING APPROACH

In this section, we summarize our approach for training the explainer model and describe several design choices. Recall that the explainer is a vision transformer $\phi_{\mathrm{ViT}}(\mathbf{x}, \mathbf{y}; \theta) \in \mathbb{R}^d$ that we train by minimizing the following loss:

$$\min_{\theta} \quad \mathbb{E}_{p(\mathbf{x}, \mathbf{y})}\mathbb{E}_{p_{\mathrm{Sh}}(\mathbf{s})}\Big[\big(v_{\mathbf{xy}}(\mathbf{s}) - v_{\mathbf{xy}}(\mathbf{0}) - \mathbf{s}^\top \phi_{\mathrm{ViT}}(\mathbf{x}, \mathbf{y}; \theta)\big)^2\Big]$$
$$\text{s.t.} \quad \mathbf{1}^\top \phi_{\mathrm{ViT}}(x, y; \theta) = v_{xy}(\mathbf{1}) - v_{xy}(\mathbf{0}) \quad \forall (x, y).$$

**Additive efficient normalization**   The constraint on the explainer predictions is necessary to ensure that the global optimizer outputs the exact Shapley values, and we use the same approach as prior work to enforce this constraint (Jethani et al., 2021b). We allow the model to make unconstrained predictions that we then modify using the following transformation:

$$\phi_{\text{ViT}}(x, y; \theta) \leftarrow \phi_{\text{ViT}}(x, y; \theta) + \frac{v_{xy}(\mathbf{1}) - v_{xy}(\mathbf{0}) - \mathbf{1}^\top \phi_{\text{ViT}}(x, y; \theta)}{d}. \tag{16}$$

This operation is known as the *additive efficient normalization* (Ruiz et al., 1998), and it can be interpreted as projecting the predictions onto the hyperplane where the constraint holds (Jethani et al., 2021b). We implement it as an output activation function, similar to how softmax is used to ensure valid probabilistic predictions for classification models.

**Subset distribution**   The specific distribution $p_{\text{Sh}}(\mathbf{s})$ in our loss function is motivated by the Shapley value's weighted least squares characterization (Charnes et al., 1988; Lundberg and Lee, 2017). This result states that the Shapley values for a game $v : \{0, 1\}^d \mapsto \mathbb{R}$ are the solution to the following optimization problem:

$$\min_{\phi \in \mathbb{R}^d} \sum_{0 < \mathbf{1}^\top s < d} \frac{d - 1}{\binom{d}{\mathbf{1}^\top s}(\mathbf{1}^\top s)(d - \mathbf{1}^\top s)} \left( v(s) - v(\mathbf{0}) - s^\top \phi \right)^2$$
$$\text{s.t.} \quad \mathbf{1}^\top \phi = v(\mathbf{1}) - v(\mathbf{0}).$$

We obtain $p_{\text{Sh}}(\mathbf{s})$ by normalizing the weighting term in the summation, and doing so yields a distribution $p_{\text{Sh}}(s) \propto (\mathbf{1}^\top s - 1)!(d - \mathbf{1}^\top s - 1)!$ for $0 < \mathbf{1}^\top s < d$ and $p_{\text{Sh}}(\mathbf{1}) = p_{\text{Sh}}(\mathbf{0}) = 0$. To sample from $p_{\text{Sh}}(\mathbf{s})$, we calculate the probability mass on each cardinality, sample a cardinality $\mathbf{m}$ from this multinomial distribution, and then select $\mathbf{m}$ indices uniformly at random.

**Stochastic gradient descent**   As is common in deep learning, we optimize our objective using stochastic gradients rather than exact gradients. To estimate our objective, we require a set of tuples $(\mathbf{x}, \mathbf{y}, \mathbf{s})$ that we obtain as follows. First, we sample an input $x \sim p(\mathbf{x})$. Next, we sample multiple subsets $s \sim p_{\text{Sh}}(\mathbf{s})$. To reduce gradient variance, we use the paired sampling trick (Covert and Lee, 2021) and pair each subset $s$ with its complement $\mathbf{1} - s$. Then, we use our explainer to output Shapley values simultaneously for all classes $y \in \{1, \ldots, K\}$. Finally, we minibatch this procedure across multiple inputs $x$ and calculate our loss across the resulting set of tuples $(\mathbf{x}, \mathbf{y}, \mathbf{s})$.

**Fine-tuning**   Rather than training the ViT explainer from scratch, we find that fine-tuning an existing model leads to better performance. This is consistent with recent work that finds ViTs challenging to train from scratch (Dosovitskiy et al., 2020). We have several options for initializing the explainer: we can use 1) the original classifier $f(\mathbf{x}; \eta)$, 2) the fine-tuned classifier $g(\mathbf{x}_s; \beta)$, or 3) a ViT pre-trained on another task. We treat this choice as a hyperparameter, selecting the initialization that yields the best performance. We also experiment with freezing certain layers in the model, but we find that training all the parameters leads to the best performance.

**Explainer architecture**   We use standard ViT architectures for the explainer. These typically append a class token to the set of image tokens (Dosovitskiy et al., 2020), and we find it beneficial to preserve this token in pre-trained architectures even though it is unnecessary for Shapley value estimation. We require a separate output head from the pre-trained architecture, and our explainer head consists of one additional self-attention block followed by three fully-connected layers. Each image patch yields one Shapley value estimate per class, and we discard the results for the class token.

**Hyperparameter tuning**   To select hyperparaters related to the learning rate, initialization and architecture, we use a pre-computed set of tuples $(\mathbf{x}, \mathbf{y}, \mathbf{s})$ to calculate a validation loss. These are generated using inputs $x$ that were not used for training, so our validation loss can be interpreted as an unbiased estimator of the objective function. This approach serves as an inexpensive alternative to comparing with ground truth Shapley values for a large number of samples.

**Training pseudocode** Algorithm 1 shows a simplified version of our training algorithm, without minibatching, sampling multiple subsets $s$, or parallelizing across the classes $y$.

---

**Algorithm 1:** Explainer training

---

**Input:** Coalitional game $v_{\mathbf{xy}}(\mathbf{s})$, learning rate $\alpha$
**Output:** Explainer $\phi_{\mathrm{ViT}}(\mathbf{x}, \mathbf{y}; \theta)$
initialize $\phi_{\mathrm{ViT}}(\mathbf{x}, \mathbf{y}; \theta)$
**while** *not converged* **do**
$\quad$ sample $(x, y) \sim p(\mathbf{x}, \mathbf{y})$, $s \sim p_{\mathrm{Sh}}(\mathbf{s})$
$\quad$ predict $\phi \leftarrow \phi_{\mathrm{ViT}}(x, y; \theta)$
$\quad$ set $\phi \leftarrow \phi + d^{-1} \left( v_{xy}(\mathbf{1}) - v_{xy}(\mathbf{0}) - \mathbf{1}^{\top}\phi \right)$
$\quad$ calculate $\mathcal{L} \leftarrow \left( v_{xy}(s) - v_{xy}(\mathbf{0}) - s^{\top}\phi \right)^2$
$\quad$ update $\theta \leftarrow \theta - \alpha\nabla_{\theta}\mathcal{L}$
**end**

---

### C.1 HYPERPARAMETER CHOICES

When training the original classifier and fine-tuned classifier models, we used a learning rate of $10^{-5}$ and trained for 25 epochs and 50 epochs, respectively. The MURA classifier was trained with an upweighted loss for negative examples to account for class imbalance. The best model was selected based on the validation criterion, where we used 0-1 accuracy for ImageNette and Oxford-IIIT Pets, and Cohen Kappa for MURA.

When training the explainer model, we used the same ViT-Base architecture as the original classifier and initialized using the fine-tuned classifier, as this gave the best results. We used the AdamW optimizer (Loshchilov and Hutter, 2018) with a cosine learning rate schedule and a maximum learning rate of $10^{-4}$, and we trained the model for 100 epochs, selecting the best model based on the validation loss. We used standard data augmentation steps: random resized crops, vertical flips, horizontal flips, and color jittering including brightness, contrast, saturation, and hue. We used minibatches of size 64 with 32 subset samples $s$ per $x$ sample, and we found that using a tanh nonlinearity on the explainer predictions was helpful to stabilize training.

Finally, we modified the ViT architecture to output Shapley values for each token and each class: we removed the classification head and added an extra attention layer, followed by three fully-connected layers with width 4 times the embedding dimension, and we fine-tuned the entire ViT backbone. These choices were determined by an ablation study with different model configurations, and we also compared with training training the ViT from scratch and training a separate U-Net explainer model (Ronneberger et al., 2015) (see Table 3).

Table 3: Ablation experiments for ViT Shapley explainer architecture on the ImageNette dataset, with and without fine-tuning.

| Fine-tuning config. | Extra attention block | Frozen backbone | Val loss | Test loss |
|:---:|:---:|:---:|:---:|:---:|
| A | False | True | 4.332 | 4.351 |
| B | False | False | 4.331 | 4.351 |
| C | True | True | 4.319 | 4.339 |
| D | True | False | **4.309** | **4.318** |
| ViT trained from scratch | | | 4.331 | 4.341 |
| U-Net trained from scratch | | | 4.332 | 4.338 |

We used a machine with 2 GeForce RTX 2080Ti GPUs to train the explainer model, and due to GPU memory constraints we loaded the classifier and explainer to separate GPUs and trained with mixed precision using PyTorch Lightning.[3] Training the explainer model required roughly 19 hours for the ImageNette dataset and 60 hours for the MURA dataset.

---

[3]`https://github.com/PyTorchLightning/pytorch-lightning`

# D PROOFS

Here, we re-state and prove our main results from Section 5.

**Lemma 1.** *For a single input-output pair $(x, y)$, the expected loss under eq. (4) for the prediction $\phi_{ViT}(x, y; \theta)$ is $\mu$-strongly convex with $\mu = H_{d-1}^{-1}$, where $H_{d-1}$ is the $(d - 1)$th harmonic number.*

*Proof.* For an input-output pair $(x, y)$, the expected loss for the prediction $\phi = \phi_{ViT}(x, y; \theta)$ under our objective is given by

$$h_{xy}(\phi) = \phi^\top \mathbb{E}_{p_{Sh}(\mathbf{s})}[\mathbf{ss}^\top]\phi - 2\phi^\top \mathbb{E}_{p_{Sh}(\mathbf{s})}\Big[\mathbf{s}\big(v_{xy}(\mathbf{s}) - v_{xy}(\mathbf{0})\big)\Big] + \mathbb{E}_{p_{Sh}(\mathbf{s})}\Big[\big(v_{xy}(\mathbf{s}) - v_{xy}(\mathbf{0})\big)^2\Big].$$

This is a quadratic function of $\phi$ with its Hessian given by

$$\nabla_\phi^2 h_{xy}(\phi) = 2 \cdot \mathbb{E}_{p_{Sh}(\mathbf{s})}[\mathbf{ss}^\top].$$

The convexity of $h_{xy}(\phi)$ is determined by the Hessian's eigenvalues, and its entries can be derived from the subset distribution $p_{Sh}(\mathbf{s})$; see similar results in Simon and Vincent (2020) and Covert and Lee (2021). The distribution assigns equal probability to subsets of equal cardinality, so we define the shorthand notation $p_k \equiv p_{Sh}(s)$ for $s$ such that $\mathbf{1}^\top s = k$. Specifically, we have

$$p_k = Q^{-1}\frac{d-1}{\binom{d}{k}k(d-k)} \qquad \text{and} \qquad Q = \sum_{k=1}^{d-1}\frac{d-1}{k(d-k)}.$$

We can then write $A \equiv \mathbb{E}_{p_{Sh}(\mathbf{s})}[\mathbf{ss}^\top]$ and derive its entries as follows:

$$A_{ii} = \Pr(\mathbf{s}_i = 1) = \sum_{k=1}^{d}\binom{d-1}{k-1}p_k$$

$$= Q^{-1}\sum_{k=1}^{d-1}\frac{(d-1)}{d(d-k)}$$

$$= \frac{\sum_{k=1}^{d-1}\frac{d-1}{d(d-k)}}{\sum_{k=1}^{d-1}\frac{d-1}{k(d-k)}}$$

$$A_{ij} = \Pr(\mathbf{s}_i = \mathbf{s}_j = 1) = \sum_{k=2}^{d}\binom{d-2}{k-2}p_k$$

$$= Q^{-1}\sum_{k=2}^{d-1}\frac{k-1}{d(d-k)}$$

$$= \frac{\sum_{k=2}^{d-1}\frac{k-1}{d(d-k)}}{\sum_{k=1}^{d-1}\frac{d-1}{k(d-k)}}$$

Based on this, we can see that $A$ has the structure $A = (b - c)I_d + c\mathbf{1}\mathbf{1}^\top$ for $b \equiv A_{ii} - A_{ij}$ and $c \equiv A_{ij}$. Following the argument by Simon and Vincent (2020), the minimum eigenvalue is then given by $\lambda_{\min}(A) = b - c$. Deriving the specific value shows that it depends on the $(d - 1)$th harmonic number, $H_{d-1}$:

$$\lambda_{\min}(A) = b - c = A_{ii} - A_{ij}$$

$$= \frac{\frac{1}{d} + \sum_{k=2}^{d-1}\left(\frac{d-1}{d(d-k)} - \frac{k-1}{d(d-k)}\right)}{\sum_{k=1}^{d-1}\frac{d-1}{k(d-k)}}$$

$$= \frac{1}{d\sum_{k=1}^{d-1}\frac{1}{k(d-k)}}$$

$$= \frac{1}{2\sum_{k=1}^{d-1}\frac{1}{k}}$$

$$= \frac{1}{2H_{d-1}}.$$

The minimum eigenvalue is therefore strictly positive, and this implies that $h_{xy}(\phi)$ is $\mu$-strongly convex with $\mu$ given by

$$\mu = 2 \cdot \lambda_{\min}(A) = H_{d-1}^{-1}.$$

Note that the strong convexity constant $\mu$ does not depend on $(x, y)$ and is determined solely by the number of features $d$.

□

**Theorem 1.** *For a model $\phi_{ViT}(\mathbf{x}, \mathbf{y}; \theta)$ whose predictions satisfy the constraint in eq. (4), the objective value $\mathcal{L}(\theta)$ upper bounds the Shapley value estimation error as follows,*

$$\mathbb{E}_{p(\mathbf{x},\mathbf{y})}\left[\big|\big|\phi_{ViT}(\mathbf{x}, \mathbf{y}; \theta) - \phi(v_{\mathbf{xy}})\big|\big|_2\right] \leq \sqrt{2H_{d-1}\Big(\mathcal{L}(\theta) - \mathcal{L}^*\Big)},$$

*where $\mathcal{L}^*$ represents the loss achieved by the exact Shapley values.*

*Proof.* We first consider a single input-output pair $(x, y)$ with prediction given by $\phi = \phi_{ViT}(x, y; \theta)$. Rather than writing the expected loss $h_{xy}(\phi)$, we now write the Lagrangian $\mathcal{L}_{xy}(\phi, \gamma)$ to account for the linear constraint in our objective, see eq. (4):

$$\mathcal{L}_{xy}(\phi, \gamma) = h_{xy}(\phi) + \gamma\big(v_{xy}(\mathbf{1}) - v_{xy}(\mathbf{0}) - \mathbf{1}^\top\phi\big).$$

Regardless of the Lagrange multiplier value $\gamma \in \mathbb{R}$, the Lagrangian $\mathcal{L}_{xy}(\phi, \gamma)$ is a $\mu$-strongly convex quadratic with the same Hessian as $h_{xy}(\phi)$:

$$\nabla_\phi^2 \mathcal{L}_{xy}(\phi, \gamma) = \nabla_\phi^2 h_{xy}(\phi).$$

Strong convexity enables us to bound $\phi$'s distance to the global minimizer via the Lagrangian's value. First, we denote the Lagrangian's optimizer as $(\phi^*, \gamma^*)$, where $\phi^*$ is given by the exact Shapley values (Charnes et al., 1988):

$$\phi^* = \phi(v_{xy}).$$

Next, we use the first-order strong convexity condition to write the following inequality:

$$\mathcal{L}_{xy}(\phi, \gamma^*) \geq \mathcal{L}_{xy}(\phi^*, \gamma^*) + (\phi - \phi^*)^\top \nabla_\phi \mathcal{L}_{xy}(\phi^*, \gamma^*) + \frac{\mu}{2}||\phi - \phi^*||^2.$$

According to the Lagrangian's KKT conditions (Boyd et al., 2004), we have the property that $\nabla_\phi \mathcal{L}_{xy}(\phi^*, \gamma^*) = 0$. The inequality therefore simplifies to

$$\mathcal{L}_{xy}(\phi, \gamma^*) \geq \mathcal{L}_{xy}(\phi^*, \gamma^*) + \frac{\mu}{2}||\phi - \phi^*||_2^2,$$

or equivalently,

$$||\phi - \phi^*||_2^2 \leq \frac{2}{\mu}\Big(\mathcal{L}_{xy}(\phi, \gamma^*) - \mathcal{L}_{xy}(\phi^*, \gamma^*)\Big).$$

If we constrain $\phi$ to be a feasible solution (i.e., it satisfies our objective's linear constraint), the KKT primal feasibility condition implies that the inequality further simplifies to

$$||\phi - \phi^*||_2^2 \leq \frac{2}{\mu}\Big(h_{xy}(\phi) - h_{xy}(\phi^*)\Big). \tag{17}$$

Now, we can consider this bound in expectation over $p(\mathbf{x}, \mathbf{y})$. First, we denote our full training loss as $\mathcal{L}(\theta)$, which is equal to

$$\mathcal{L}(\theta) = \mathbb{E}_{p(\mathbf{x},\mathbf{y})}\mathbb{E}_{p_{\text{Sh}}(\mathbf{s})}\Big[\big(v_{\mathbf{x}\mathbf{y}}(\mathbf{s}) - v_{\mathbf{x}\mathbf{y}}(\mathbf{0}) - \mathbf{s}^\top \phi_{\text{ViT}}(\mathbf{x}, \mathbf{y}; \theta)\big)^2\Big] = \mathbb{E}_{p(\mathbf{x},\mathbf{y})}\Big[h_{\mathbf{x}\mathbf{y}}\big(\phi_{\text{ViT}}(\mathbf{x}, \mathbf{y}; \theta)\big)\Big].$$

Next, we denote $\mathcal{L}^*$ as the training loss achieved by the exact Shapley values, or

$$\mathcal{L}^* = \mathbb{E}_{p(\mathbf{x},\mathbf{y})}\Big[h_{\mathbf{x}\mathbf{y}}\big(\phi(v_{\mathbf{x}\mathbf{y}})\big)\Big].$$

Given a network $\phi_{\text{ViT}}(\mathbf{x}, \mathbf{y}; \theta)$ whose predictions are constrained to satisfy the linear constraint, taking the bound from eq. (17) in expectation yields the following bound on the distance between the predicted and exact Shapley values:

$$\mathbb{E}_{p(\mathbf{x},\mathbf{y})}\Big[\big|\big|\phi_{\text{ViT}}(\mathbf{x}, \mathbf{y}; \theta) - \phi(v_{\mathbf{x}\mathbf{y}})\big|\big|_2^2\Big] \leq \frac{2}{\mu}\Big(\mathcal{L}(\theta) - \mathcal{L}^*\Big).$$

Applying Jensen's inequality to the left side, we can rewrite the bound as follows:

$$\mathbb{E}_{p(\mathbf{x},\mathbf{y})}\Big[\big|\big|\phi_{\text{ViT}}(\mathbf{x}, \mathbf{y}; \theta) - \phi(v_{\mathbf{x}\mathbf{y}})\big|\big|_2\Big] \leq \sqrt{\frac{2}{\mu}\Big(\mathcal{L}(\theta) - \mathcal{L}^*\Big)}.$$

Substituting in the strong convexity parameter $\mu$ from Lemma 1, we arrive at the final bound:

$$\mathbb{E}_{p(\mathbf{x},\mathbf{y})}\Big[\big|\big|\phi_{\text{ViT}}(\mathbf{x}, \mathbf{y}; \theta) - \phi(v_{\mathbf{x}\mathbf{y}})\big|\big|_2\Big] \leq \sqrt{2H_{d-1}\Big(\mathcal{L}(\theta) - \mathcal{L}^*\Big)}.$$

$\square$

We also present a corollary to Theorem 1. This result formalizes the intuition that if we can iteratively optimize the explainer such that its loss approaches the optimum, our Shapley value estimation error will go to zero.

**Corollary 1.** *Given a sequence of models $\phi_{\text{ViT}}(\mathbf{x}, \mathbf{y}; \theta_1), \phi_{\text{ViT}}(\mathbf{x}, \mathbf{y}; \theta_2), \ldots$ whose predictions satisfy the constraint in eq. (4) and where $\mathcal{L}(\theta_n) \to \mathcal{L}^*$, the Shapley value estimation error goes to zero:*

$$\lim_{n\to\infty} \mathbb{E}_{p(\mathbf{x},\mathbf{y})}\Big[\big|\big|\phi_{\text{ViT}}(\mathbf{x}, \mathbf{y}; \theta_n) - \phi(v_{\mathbf{x}\mathbf{y}})\big|\big|_2\Big] = 0.$$

*Proof.* Fix $\epsilon > 0$. By assumption, there exists a value $n'$ such that $\mathcal{L}(\theta_n) - \mathcal{L}^* < \frac{\mu\epsilon^2}{2}$ for $n > n'$. Following the result in Theorem 1, we have $\mathbb{E}_{p(\mathbf{x},\mathbf{y})}\left[\left|\left|\phi_{\text{ViT}}(\mathbf{x}, \mathbf{y}; \theta_n) - \phi(v_{\mathbf{xy}})\right|\right|_2\right] < \epsilon$ for $n > n'$.

$\square$

Finally, we also consider the role of our loss function in quantifying the Shapley value estimation error, which we define for a given explainer model $\phi_{\text{ViT}}(\mathbf{x}, \mathbf{y}; \theta)$ as

$$\text{SVE} = \mathbb{E}_{p(\mathbf{x},\mathbf{y})}\left[\left|\left|\phi_{\text{ViT}}(\mathbf{x}, \mathbf{y}; \theta) - \phi(v_{\mathbf{xy}})\right|\right|_2\right].$$

One natural approach is to use an external dataset (e.g., the test data) consisting of samples $(x_i, y_i)$ for $i = 1, \ldots, n$, calculate their exact Shapley values $\phi(v_{x_i y_i})$, and generate a Monte Carlo estimate as follows:

$$\hat{\text{SVE}}_n = \frac{1}{n} \sum_{i=1}^{n} \left|\left|\phi_{\text{ViT}}(x_i, y_i; \theta) - \phi(v_{x_i y_i})\right|\right|_2.$$

While standard concentration inequalities allow us to bound SVE using $\hat{\text{SVE}}_n$, generating the ground truth values can be computationally costly, particularly for large $n$. Instead, another approach is to use our result from Theorem 1, which bypasses the need for ground truth Shapley values. For this, recall that $\mathcal{L}(\theta)$ represents our weighted least squares loss function, where we assume that the explainer $\phi_{\text{ViT}}(\mathbf{x}, \mathbf{y}; \theta)$ satisfies the constraint in eq. (4) for all predictions. If we know $\mathcal{L}(\theta)$ exactly, then Theorem 1 yields the following bound with probability 1:

$$\text{SVE} \leq \sqrt{2H_{d-1}\Big(\mathcal{L}(\theta) - \mathcal{L}^*\Big)}.$$

If we do not know $\mathcal{L}(\theta)$ exactly, we can instead form a Monte Carlo estimate $\hat{\mathcal{L}}(\theta)_n$ using samples $(x_i, y_i, s_i)$ for $i = 1, \ldots, n$. Then, using concentration inequalities like Chebyshev or Hoeffding (the latter only applies if we assume bounded errors), we can get probabilistic bounds of the form $P(|\mathcal{L}(\theta) - \hat{\mathcal{L}}_n| > \epsilon) \leq \delta$. With these, we can say with probability at least $1 - \delta$ that $\mathcal{L}(\theta) \leq \hat{\mathcal{L}}(\theta)_n + \epsilon$. Finally, combining this with the last steps of our Theorem 1 proof, we obtain the following bound with probability at least $1 - \delta$:

$$\text{SVE} \leq \sqrt{2H_{d-1}\Big(\hat{\mathcal{L}}(\theta)_n - \mathcal{L}^* + \epsilon\Big)}. \tag{18}$$

Naturally, $\delta$ is a function of $\epsilon$ and the number of samples $n$ used to estimate $\hat{\mathcal{L}}(\theta)_n$, with the rate of convergence to probability 1 depending on the choice of concentration inequality (Chebyshev or Hoeffding). Although this procedure yields an inexpensive upper bound on the Shapley value estimation error, the bound's looseness, as well as the fact that we do not know $\mathcal{L}^*$ a priori, make it unappealing as an evaluation metric. The more important takeaways are 1) that training with the loss $\mathcal{L}(\theta)$ effectively minimizes an upper bound on the Shapley value estimation error, and 2) that comparing explainer models via their validation loss, which is effectively $\hat{\mathcal{L}}(\theta)_n$, is a principled approach to perform model selection and hyperparameter tuning.

## E  DATASETS

The ImageNette dataset contains 9,469 training examples and 3,925 validation examples, and we split the validation data to obtain validation and test sets containing 1,962 examples each. The MURA dataset contains 36,808 training examples and 3,197 validation examples. We use the validation examples as a test set, and we split the training examples to obtain train and validation sets containing 33,071 and 3,737 examples, ensuring that images from the same patient belong to a single split. The Oxford-IIIT Pets dataset contains 7,349 examples for 37 classes, and we split the data to obtain train, validation, and test sets containing 5,879, 735, and 735 examples, respectively. For all datasets,

Table 4: Performance metrics for target-class explanations with additional baselines. Methods that fail to outperform the random baseline are shown in gray, and the best results are shown in bold (accounting for 95% confidence intervals).

| | ImageNette | | | MURA | | |
|---|---|---|---|---|---|---|
| | Ins. (↑) | Del. (↓) | Faith. (↑) | Ins. (↑) | Del. (↓) | Faith. (↑) |
| Attention last | 0.962 (0.004) | 0.793 (0.013) | **0.694 (0.015)** | 0.890 (0.010) | 0.592 (0.013) | 0.635 (0.016) |
| Attention rollout | 0.938 (0.005) | 0.880 (0.010) | **0.704 (0.015)** | 0.845 (0.011) | 0.692 (0.014) | 0.618 (0.016) |
| GradCAM (LN) | 0.914 (0.006) | 0.937 (0.008) | 0.680 (0.015) | 0.899 (0.009) | 0.681 (0.015) | 0.631 (0.016) |
| GradCAM (Attn) | 0.938 (0.006) | 0.948 (0.006) | 0.656 (0.014) | 0.843 (0.012) | 0.835 (0.011) | 0.580 (0.016) |
| IntGrad (Pixel) | 0.967 (0.004) | 0.930 (0.008) | 0.403 (0.024) | 0.897 (0.010) | 0.796 (0.015) | 0.201 (0.022) |
| IntGrad (Embed.) | 0.967 (0.004) | 0.930 (0.008) | 0.403 (0.024) | 0.897 (0.010) | 0.796 (0.015) | 0.201 (0.022) |
| Vanilla (Pixel) | 0.938 (0.005) | 0.860 (0.011) | **0.700 (0.015)** | 0.890 (0.010) | 0.561 (0.014) | 0.627 (0.016) |
| Vanilla (Embed.) | 0.950 (0.004) | 0.808 (0.013) | **0.703 (0.015)** | 0.890 (0.010) | 0.537 (0.014) | 0.629 (0.016) |
| SmoothGrad (Pixel) | 0.960 (0.005) | 0.779 (0.013) | **0.706 (0.015)** | 0.873 (0.010) | 0.634 (0.014) | 0.618 (0.016) |
| SmoothGrad (Embed.) | 0.947 (0.005) | 0.942 (0.006) | **0.703 (0.015)** | 0.870 (0.010) | 0.813 (0.011) | 0.617 (0.016) |
| VarGrad (Pixel) | 0.958 (0.005) | 0.796 (0.013) | **0.682 (0.015)** | 0.871 (0.010) | 0.660 (0.013) | 0.577 (0.015) |
| VarGrad (Embed.) | 0.949 (0.005) | 0.946 (0.005) | **0.700 (0.015)** | 0.857 (0.011) | 0.823 (0.011) | 0.615 (0.016) |
| LRP | 0.967 (0.004) | 0.779 (0.014) | **0.705 (0.015)** | 0.900 (0.009) | 0.551 (0.013) | 0.646 (0.016) |
| Leave-one-out | 0.969 (0.002) | 0.917 (0.010) | 0.140 (0.040) | 0.926 (0.008) | 0.694 (0.017) | 0.308 (0.032) |
| RISE | 0.977 (0.001) | 0.860 (0.014) | **0.704 (0.015)** | 0.957 (0.004) | 0.573 (0.018) | 0.618 (0.016) |
| **ViT Shapley** | **0.985 (0.002)** | **0.691 (0.014)** | **0.711 (0.015)** | **0.971 (0.002)** | **0.307 (0.013)** | **0.707 (0.013)** |
| Random | 0.951 (0.005) | 0.951 (0.005) | - | 0.849 (0.010) | 0.847 (0.010) | - |

the training and validation data were used to train the original classifiers, fine-tuned classifiers and explainer models, and the test data was used only when calculating performance metrics.

# F  BASELINE METHODS

This section provides implementation details for the baseline explanation methods. We used a variety of attention-, gradient- and removal-based methods as comparisons for ViT Shapley, and we modified several approaches to arrive at patch-level feature attribution scores.

**Attention last**  This approach calculates the attention directed from each image token into the class token in the final self-attention layer, summed across attention heads (Abnar and Zuidema, 2020; Chefer et al., 2021). The results are provided at the patch-level, but they are not generated separately for each output class.

**Attention rollout**  This approach accounts for the flow of attention between tokens by summing across attention heads and multiplying the resulting attention matrices at each layer (Abnar and Zuidema, 2020). Like the previous method, results are not generated separately for each output class. We used an implementation provided by prior work (Chefer et al., 2021).

**Common gradient-based methods**  Several methods that operate via input gradients are Vanilla gradients (Simonyan et al., 2013), SmoothGrad (Smilkov et al., 2017), VarGrad (Hooker et al., 2019), and IntGrad (Sundararajan et al., 2017). These methods were run using the Captum package (Kokhlikyan et al., 2020), and we used 10 samples per image for SmoothGrad, VarGrad and IntGrad. We tried applying these at the level of pixels and patch embeddings, and in both cases we created class-specific, patch-level attributions by summing across the unnecessary dimensions. We calculated the absolute value before summing for Vanilla and SmoothGrad, VarGrad automatically produces non-negative values, and we preserved the sign for IntGrad because it should be meaningful.

**GradCAM**  Originally designed for intermediate convolutional layers (Selvaraju et al., 2017), GradCAM has since been generalized to the ViT context. The main operations remain the same, only the representation being analyzed is the layer-normed input to the final self-attention layer, and the aggregation is across the embedding dimension rather than convolutional channels (GradCAM LN) (Gildenblat and contributors, 2021). We also experimented with using a different internal layer for

generating explanations (the attention weights computed in the final self-attention layer, denoted as GradCAM Attn. (Chefer et al., 2021)).

**Layer-wise relevance propagation (LRP)**    Originally described as a set of constraints for a modified backpropagation routine (Bach et al., 2015), LRP has since been implemented for a variety of network layers and architectures, and it was recently adapted to ViTs (Chefer et al., 2021). We used an implementation provided by prior work (Chefer et al., 2021).

**Leave-one-out**    The importance scores in this approach are the difference in prediction probability for the full-image and the iamge with a single patch removed. We removed patches by setting pixels to zero, similar to the original version for CNNs (Zeiler and Fergus, 2014).

**RISE**    This approach involves sampling many occlusion masks and reporting the mean prediction when each patch is included. The original version for CNNs (Petsiuk et al., 2018) used a complex approach to generate masks, but we simply sampled subsets of patches. As in the original work, we sample from all subsets with equal probability, and we use 2,000 mask samples per sample to be explained. We occlude patches by setting pixel values to zero, similar to the original work.

**Random**    Finally, we included a random baseline as a comparison for the insertion, deletion and ROAR metrics. These metrics only require a ranking of important patches, so we generated ten random orderings and averaged the results across these orderings.

Table 4 shows the same metrics as Table 1 with additional results for alternative implementations of several baselines. For the methods based on input gradients, we experimented with generating explanations at both the pixel level and embedding level; the preferred approach depends on the method and metric, but both versions tend to underperform ViT Shapley, with the exception of faithfulness on ImageNette where the 95% confidence intervals overlap for many methods. We also experimented with two versions of GradCAM (described above) and find that the GradCAM LN implementation generally performs slightly better. In the main text, we present results only for GradCAM LN and the remaining gradient-based methods generated at the embedding level.

## G    METRICS DETAILS

This section provides additional details about the performance metrics used in the main text experiments (Section 6).

**Insertion/deletion**    These metrics involve repeatedly making predictions while either inserting or deleting features in order of most to least important (Petsiuk et al., 2018). While the original work removed features by setting them to zero, we use the fine-tuned classifier that was trained to handle partial information. We calculated the area under the curve for individual predictions and then averaged the results across 1,000 test set examples; we used random examples for ImageNette, and only examples that were predicted to be abnormal for MURA. Table 1 presents results for the true class only, and Table 2 presents results averaged across all the remaining classes.

**Sensitivity-n**    This metric samples feature subsets at random and calculates the correlation between the prediction with each subset and the sum of the corresponding features' attribution scores (Ancona et al., 2018). It typically considers subsets of a fixed size $n$, which means sampling from the following subset distribution $p_n(s)$:

$$p_n(s) = \mathbb{1}(\mathbf{1}^\top s = n)\binom{d}{n}^{-1}.$$

Mathematically, the metric is defined for a model $f(\mathbf{x})$, an individual sample $x$ and label $y$, feature attributions $\phi \in \mathbb{R}^d$ and subset size $n$ as follows:

$$\text{Sens}(f, x, y, \phi, n) = \text{Corr}_{p_n(\mathbf{s})}\big(\mathbf{s}^\top\phi, f_y(x) - f_y(x_{\mathbf{1-s}})\big).$$

Similar to insertion/deletion, we use the fine-tuned classifier to handle held-out patches and calculate the metric across 1,000 test set images. We use subset sizes ranging from 14 to 182 patches with step size 14, and we estimate the correlation for each example and subset size using 1,000 subset samples.

**Faithfulness**   This metric is nearly identical to sensitivity-n, only it calculates the correlation across subsets of all sizes (Bhatt et al., 2021). Mathematically, it is defined as

$$\text{Faith}(f, x, y, \phi) = \text{Corr}_{p(\mathbf{s})}\big(\mathbf{s}^\top \phi, f_y(x) - f_y(x_{\mathbf{1-s}})\big),$$

and we sample from a distribution with equal probability mass on all cardinalities, or

$$p(s) = \frac{1}{\binom{d}{\mathbf{1}^\top s} \cdot (d + 1)}.$$

We use the fine-tuned classifier to handle held-out patches, and we compute faithfulness across 1,000 test set images and with 1,000 subset samples per image.

**ROAR**   Finally, ROAR evaluates the model's accuracy after removing features in order from most to least important (Hooker et al., 2019). We also experimented with *inserting* features in order of most to least important. Crucially, the ROAR authors propose handling held-out features by retraining the model with masked inputs. We performed masked retraining by performing test-time augmentations for all training, validation and test set images, generating explanations to identify the most important patches for the true class, and setting the corresponding pixels to zero.

Because masked retraining leaks information through the masking, we also replicated this metric using the fine-tuned classifier model, and with a separate evaluator model trained directly with random masking; the evaluator model trained with random masking has been used in prior work (Jethani et al., 2021a;b). We generated results for each number of inserted/deleted patches (1, 3, 7, 14, 28, 56, 84, 112, 140, 168, and 182) with the final accuracy computed across the entire test set.

**Ground truth metrics**   Previous work has considered evaluations involving comparison with ground truth importance, where the ground truth is either identified by humans (Chefer et al., 2021) or introduced via synthetic dataset modifications (Zhou et al., 2022). An important issue with such methods is that they test explanations against what a model should depend on rather than what it does depend on, so the results do not reflect the explanation's accuracy for the specific model (Petsiuk et al., 2018). We thus decided against including such metrics.

## H   ADDITIONAL RESULTS

This section provides additional experimental results. We first show results involving similar baselines and metrics as in the main text, and we then show results comparing ViT Shapley to KernelSHAP.

### H.1   MAIN BASELINES AND METRICS

Figure 5 shows our evaluation of attention masking for handling held-out image patches using two separate metrics: 1) KL divergence relative to the full-image predictions (also shown in the main text), and 2) top-1 accuracy relative to the true labels. The former can be understood as a divergence measure between the predictions with masked inputs and the predictions with patches marginalized out using their conditional distribution (see Appendix B). The latter is a more intuitive measure of how much the performance degrades given partial inputs. The results are similar between the two metrics, showing that the predictions diverge more quickly if the model is not fine-tuned with random masking.

Table 5 shows insertion, deletion and faithfulness results for the MURA dataset with examples that were predicted to be normal, but while evaluating explanations for the abnormal class. ViT Shapley outperforms the baseline methods, reflecting that our explanations correctly identify patches that influence the prediction towards and against the abnormal class even for normal examples.

Table 6 shows insertion, deletion and faithfulness results for the Pets dataset. We observe that ViT Shapley outperforms other methods for all metrics with the exception of faithfulness for target classes, where 95% confidence intervals overlap for many methods (similar to the other datasets).

Table 7, Table 8, and Table 9 show insertion, deletion, and faithfulness metrics for ImageNette when using other ViT architectures (i.e., ViT-Tiny, -Small, and -Large, respectively) (Wightman, 2019; Dosovitskiy et al., 2020) for the classifier and explainer. They show results for target-class

explanations and non-target-class explanations, respectively. The results are consistent with those obtained for ViT-Base, and ViT Shapley outperforms the baseline methods across all three metrics. This shows that our explainer model can be trained successfully with architectures of different sizes, including when using a relatively small number of parameters.

Table 10 shows insertion, deletion and faithfulness results for a ViT classifier trained directly with random masking. Whereas our Section 6 experiments utilize a fine-tuned classifier to handle missing patches, a classifier trained with random masking allows us to bypass the fine-tuning stage and train the explainer directly. The results are similar to Table 1, and we find that ViT Shapley consistently achieves the best performance.

Figure 6, Figure 7, and Figure 8 show the average curves used to generate the insertion/deletion AUC results. All sets of plots reflect that explanations from ViT Shapley identify relevant patches that quickly move the prediction towards or away from a given class. In the case of ImageNette and Pets, we observe that this holds for both target and non-target classes.

Figure 9 shows the sensitivity-n metric evaluated for non-target classes on the ImageNette dataset. Similarly, these results show that ViT Shapley generates attribution scores that represent the impact of withholding features from a model, even for non-target classes. In this case, RISE and leave-one-out are more competitive with ViT Shapley, but their performance is less competitive when the correlation is calculated for subsets of all sizes (see faithfulness in Table 2).

Next, Figure 10 shows ROAR results generated in both the insertion and deletion directions, using the four patch removal approaches: 1) the fine-tuned classifier, 2) the separate evaluator model trained directly with random masking, 3) masked retraining, and 4) masked retraining without positional embeddings. The results show that in addition to strong performance in the deletion direction, ViT Shapley consistently achieves the best results in the insertion direction, even in the case of masked retraining with positional embeddings.

Figure 11 shows ROAR results for the same settings, but when using the ViT-Small architecture. We observe the same results obtained with ViT-Base. Except for the deletion direction with masked retraining and positional embeddings enabled, ViT Shapley achieves the best performance among all methods.

Finally, Table 11 shows the time required to generate explanations using each approach. Because ViT Shapley requires a single forward pass through the explainer model, it is among the fastest approaches and is paralleled only by the attention-based methods. The gradient-based methods require forward and backward passes for all classes, and sometimes for many altered inputs (e.g., with noise injected for SmoothGrad). RISE is the slowest of all the approaches tested because it requires making several thousand predictions to explain each sample. Our evaluation was conducted on a GeForce RTX 2080 Ti GPU, with minibatches of 16 samples for attention last, attention rollout and ViT Shapley; batch size of 1 for Vanilla Gradients, GradCAM, LRP, leave-one-out and RISE; and internal minibatching for SmoothGrad, IntGrad and VarGrad (implemented via Captum (Kokhlikyan et al., 2020)).

ViT Shapley is the only method considered here to require training time, and as described in Appendix C, training the explainer models required roughly 0.8 days for ImageNette and 2.5 days for MURA. The training time is not insignificant, but investing time in training the explainer is worthwhile if 1) high-quality explanations are required, 2) there are many examples to be explained (e.g., an entire dataset), or 3) fast explanations are required during a model's deployment.

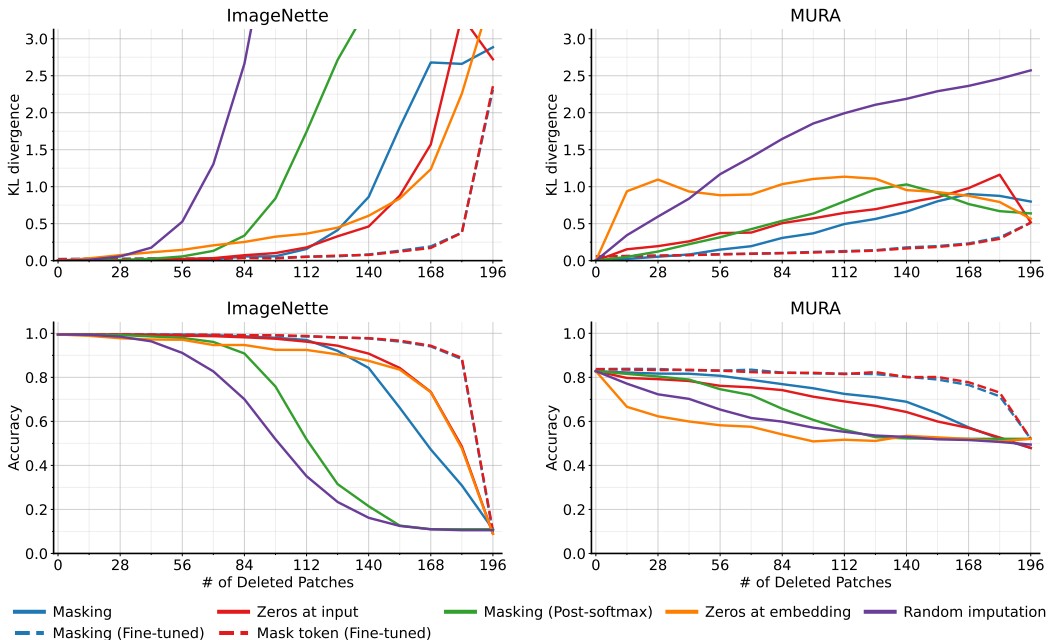

Figure 5: ViT predictions given partial information. We delete patches at random using several removal mechanisms, and we then measure the quality of the resulting predictions via two metrics: the KL divergence relative to the original, full-image predictions (top), and the top-1 accuracy relative to the true labels (bottom).

Table 5: MURA non-target metrics for images that were predicted to be normal. Methods that fail to outperform the random baseline are shown in gray, and the best results are shown in bold (accounting for 95% confidence intervals).

|  | MURA | | |
| --- | --- | --- | --- |
|  | Ins. (↑) | Del. (↓) | Faith. (↑) |
| Attention last | 0.157 (0.011) | 0.210 (0.009) | -0.455 (0.022) |
| Attention rollout | 0.199 (0.011) | 0.169 (0.010) | -0.480 (0.023) |
| GradCAM | 0.197 (0.012) | 0.152 (0.010) | -0.449 (0.022) |
| IntGrad | 0.209 (0.014) | 0.151 (0.010) | 0.103 (0.023) |
| Vanilla | 0.174 (0.011) | 0.197 (0.009) | -0.477 (0.023) |
| SmoothGrad | 0.169 (0.011) | 0.176 (0.011) | -0.480 (0.023) |
| VarGrad | 0.166 (0.011) | 0.175 (0.011) | -0.478 (0.023) |
| LRP | 0.169 (0.012) | 0.200 (0.009) | -0.461 (0.023) |
| Leave-one-out | 0.326 (0.016) | 0.093 (0.007) | 0.272 (0.029) |
| RISE | 0.406 (0.018) | 0.075 (0.006) | -0.479 (0.023) |
| **ViT Shapley** | **0.580 (0.016)** | **0.039 (0.003)** | **0.642 (0.014)** |
| Random | 0.169 (0.011) | 0.170 (0.011) | - |

Table 6: Performance metrics for ViT-Base on Pets. Methods that fail to outperform the random baseline are shown in gray, and the best results are shown in bold (accounting for 95% confidence intervals).

| | Target | | | Non-Target | | |
|---|---|---|---|---|---|---|
| | Ins. (↑) | Del. (↓) | Faith. (↑) | Ins. (↑) | Del. (↓) | Faith. (↑) |
| Attention last | 0.876 (0.011) | 0.494 (0.016) | 0.545 (0.017) | - | - | - |
| Attention rollout | 0.837 (0.012) | 0.672 (0.017) | **0.559 (0.017)** | - | - | - |
| GradCAM (Attn) | 0.837 (0.013) | 0.737 (0.018) | 0.527 (0.016) | 0.005 (0.000) | 0.008 (0.000) | -0.499 (0.013) |
| GradCAM (LN) | 0.837 (0.011) | 0.717 (0.019) | 0.553 (0.017) | 0.012 (0.001) | 0.004 (0.000) | -0.511 (0.014) |
| IntGrad (Pixel) | 0.899 (0.010) | 0.802 (0.016) | 0.437 (0.017) | 0.006 (0.001) | 0.003 (0.000) | 0.126 (0.008) |
| IntGrad (Embed.) | 0.899 (0.010) | 0.802 (0.016) | 0.437 (0.017) | 0.006 (0.001) | 0.003 (0.000) | 0.126 (0.008) |
| Vanilla (Pixel) | 0.862 (0.011) | 0.596 (0.018) | **0.559 (0.017)** | 0.004 (0.000) | 0.011 (0.000) | -0.526 (0.014) |
| Vanilla (Embed.) | 0.876 (0.011) | 0.519 (0.017) | **0.561 (0.017)** | 0.004 (0.000) | 0.013 (0.000) | -0.529 (0.014) |
| SmoothGrad (Pixel) | 0.889 (0.011) | 0.488 (0.016) | **0.564 (0.017)** | 0.003 (0.000) | 0.014 (0.000) | -0.532 (0.014) |
| SmoothGrad (Embed.) | 0.840 (0.013) | 0.834 (0.013) | **0.558 (0.017)** | 0.004 (0.000) | 0.005 (0.000) | -0.526 (0.014) |
| VarGrad (Pixel) | 0.891 (0.011) | 0.513 (0.016) | **0.557 (0.017)** | 0.003 (0.000) | 0.013 (0.000) | -0.522 (0.014) |
| VarGrad (Embed.) | 0.847 (0.012) | 0.835 (0.013) | 0.555 (0.017) | 0.004 (0.000) | 0.005 (0.000) | -0.523 (0.014) |
| LRP | 0.890 (0.011) | 0.466 (0.016) | **0.567 (0.017)** | 0.004 (0.000) | 0.013 (0.000) | -0.529 (0.014) |
| Leave-one-out | **0.936 (0.006)** | 0.653 (0.022) | 0.215 (0.036) | 0.018 (0.001) | 0.001 (0.000) | 0.138 (0.023) |
| RISE | **0.946 (0.005)** | 0.505 (0.022) | **0.559 (0.017)** | 0.032 (0.002) | 0.001 (0.000) | -0.523 (0.014) |
| **ViT Shapley** | **0.945 (0.006)** | **0.388 (0.017)** | **0.590 (0.016)** | **0.052 (0.002)** | **0.001 (0.000)** | **0.529 (0.012)** |
| Random | 0.848 (0.012) | 0.845 (0.012) | - | 0.004 (0.000) | 0.004 (0.000) | - |

Table 7: Performance metrics for ViT-Tiny on ImageNette. Methods that fail to outperform the random baseline are shown in gray, and the best results are shown in bold (accounting for 95% confidence intervals).

| | Target | | | Non-Target | | |
|---|---|---|---|---|---|---|
| | Ins. (↑) | Del. (↓) | Faith. (↑) | Ins. (↑) | Del. (↓) | Faith. (↑) |
| Attention last | 0.917 (0.008) | 0.702 (0.015) | 0.649 (0.014) | - | - | - |
| Attention rollout | 0.901 (0.009) | 0.698 (0.015) | 0.658 (0.015) | - | - | - |
| GradCAM (LN) | 0.918 (0.007) | 0.809 (0.014) | 0.659 (0.014) | 0.034 (0.002) | 0.006 (0.000) | -0.616 (0.013) |
| GradCAM (Attn) | 0.898 (0.009) | 0.814 (0.013) | 0.636 (0.014) | 0.011 (0.001) | 0.020 (0.001) | -0.607 (0.012) |
| IntGrad (Pixel) | 0.928 (0.006) | 0.852 (0.013) | 0.293 (0.025) | 0.018 (0.002) | 0.008 (0.001) | 0.094 (0.014) |
| IntGrad (Embed.) | 0.928 (0.006) | 0.852 (0.013) | 0.293 (0.025) | 0.018 (0.002) | 0.008 (0.001) | 0.094 (0.014) |
| Vanilla (Pixel) | 0.884 (0.009) | 0.801 (0.013) | 0.656 (0.015) | 0.015 (0.001) | 0.020 (0.001) | -0.629 (0.013) |
| Vanilla (Embed.) | 0.902 (0.008) | 0.752 (0.015) | 0.659 (0.015) | 0.012 (0.001) | 0.025 (0.001) | -0.632 (0.013) |
| SmoothGrad (Pixel) | 0.908 (0.009) | 0.733 (0.015) | 0.659 (0.015) | 0.011 (0.001) | 0.029 (0.002) | -0.632 (0.013) |
| SmoothGrad (Embed.) | 0.917 (0.008) | 0.756 (0.014) | 0.659 (0.015) | 0.010 (0.001) | 0.025 (0.002) | -0.632 (0.013) |
| VarGrad (Pixel) | 0.914 (0.008) | 0.754 (0.014) | 0.648 (0.014) | 0.011 (0.001) | 0.027 (0.002) | -0.620 (0.012) |
| VarGrad (Embed.) | 0.921 (0.008) | 0.777 (0.014) | 0.648 (0.014) | 0.010 (0.001) | 0.023 (0.001) | -0.618 (0.012) |
| LRP | 0.938 (0.007) | 0.648 (0.016) | **0.673 (0.014)** | 0.014 (0.001) | 0.024 (0.001) | -0.623 (0.013) |
| Leave-one-out | 0.956 (0.004) | 0.737 (0.019) | 0.221 (0.039) | 0.052 (0.004) | 0.003 (0.000) | 0.145 (0.025) |
| RISE | 0.970 (0.003) | 0.602 (0.019) | 0.658 (0.015) | 0.092 (0.005) | 0.002 (0.000) | -0.628 (0.013) |
| **ViT Shapley** | **0.981 (0.002)** | **0.457 (0.015)** | **0.694 (0.014)** | **0.198 (0.006)** | **0.001 (0.000)** | **0.641 (0.011)** |
| Random | 0.908 (0.008) | 0.907 (0.008) | - | 0.010 (0.001) | 0.010 (0.001) | - |

Table 8: Performance metrics for ViT-Small on ImageNette. Methods that fail to outperform the random baseline are shown in gray, and the best results are shown in bold (accounting for 95% confidence intervals).

| | Target | | | Non-Target | | |
|---|---|---|---|---|---|---|
| | Ins. (↑) | Del. (↓) | Faith. (↑) | Ins. (↑) | Del. (↓) | Faith. (↑) |
| Attention last | 0.949 (0.006) | 0.706 (0.014) | **0.781 (0.011)** | - | - | - |
| Attention rollout | 0.914 (0.007) | 0.814 (0.013) | **0.787 (0.011)** | - | - | - |
| GradCAM (LN) | 0.908 (0.006) | 0.898 (0.011) | 0.770 (0.010) | 0.027 (0.002) | 0.006 (0.000) | -0.740 (0.009) |
| GradCAM (Attn.) | 0.904 (0.009) | 0.911 (0.009) | 0.724 (0.010) | 0.008 (0.001) | 0.016 (0.001) | -0.717 (0.009) |
| IntGrad (Pixel) | 0.951 (0.006) | 0.908 (0.010) | 0.541 (0.022) | 0.011 (0.001) | 0.005 (0.001) | 0.384 (0.018) |
| IntGrad (Embed.) | 0.951 (0.006) | 0.908 (0.010) | 0.541 (0.022) | 0.011 (0.001) | 0.005 (0.001) | 0.384 (0.018) |
| Vanilla (Pixel) | 0.905 (0.007) | 0.830 (0.012) | **0.784 (0.010)** | 0.011 (0.001) | 0.018 (0.001) | -0.754 (0.009) |
| Vanilla (Embed.) | 0.921 (0.007) | 0.793 (0.013) | **0.786 (0.011)** | 0.009 (0.001) | 0.022 (0.001) | -0.756 (0.009) |
| SmoothGrad (Pixel) | 0.943 (0.006) | 0.744 (0.014) | **0.786 (0.011)** | 0.007 (0.001) | 0.027 (0.001) | -0.756 (0.009) |
| SmoothGrad (Embed.) | 0.946 (0.006) | 0.753 (0.014) | **0.787 (0.011)** | 0.006 (0.001) | 0.026 (0.001) | -0.757 (0.009) |
| VarGrad (Pixel) | 0.946 (0.006) | 0.766 (0.013) | 0.742 (0.010) | 0.007 (0.001) | 0.025 (0.001) | -0.709 (0.009) |
| VarGrad (Embed.) | 0.947 (0.006) | 0.774 (0.013) | 0.758 (0.010) | 0.007 (0.001) | 0.024 (0.001) | -0.723 (0.009) |
| LRP | 0.957 (0.005) | 0.695 (0.015) | **0.793 (0.010)** | 0.007 (0.001) | 0.027 (0.001) | -0.753 (0.009) |
| Leave-one-out | 0.967 (0.002) | 0.855 (0.015) | 0.044 (0.042) | 0.024 (0.002) | 0.003 (0.000) | 0.079 (0.030) |
| RISE | 0.976 (0.002) | 0.752 (0.018) | **0.787 (0.010)** | 0.049 (0.004) | 0.002 (0.000) | -0.755 (0.009) |
| **ViT Shapley** | **0.982 (0.002)** | **0.591 (0.015)** | **0.801 (0.010)** | **0.130 (0.005)** | **0.001 (0.000)** | **0.747 (0.009)** |
| Random | 0.933 (0.006) | 0.934 (0.006) | - | 0.007 (0.001) | 0.007 (0.001) | - |

Table 9: Performance metrics for ViT-Large on ImageNette. Methods that fail to outperform the random baseline are shown in gray, and the best results are shown in bold (accounting for 95% confidence intervals).

| | Target | | | Non-Target | | |
|---|---|---|---|---|---|---|
| | Ins. (↑) | Del. (↓) | Faith. (↑) | Ins. (↑) | Del. (↓) | Faith. (↑) |
| Attention last | 0.915 (0.005) | 0.896 (0.008) | 0.401 (0.022) | - | - | - |
| Attention rollout | 0.929 (0.005) | 0.928 (0.007) | **0.423 (0.023)** | - | - | - |
| GradCAM (LN) | 0.955 (0.004) | 0.924 (0.009) | **0.424 (0.023)** | 0.012 (0.001) | 0.005 (0.000) | -0.372 (0.020) |
| GradCAM (Attn) | 0.935 (0.005) | 0.930 (0.007) | 0.395 (0.022) | 0.007 (0.001) | 0.006 (0.001) | -0.351 (0.018) |
| IntGrad (Pixel) | 0.965 (0.003) | 0.947 (0.006) | 0.195 (0.020) | 0.006 (0.001) | 0.004 (0.000) | 0.114 (0.013) |
| IntGrad (Embed.) | 0.965 (0.003) | 0.948 (0.006) | 0.198 (0.020) | 0.006 (0.001) | 0.004 (0.000) | 0.117 (0.013) |
| Vanilla (Pixel) | 0.898 (0.007) | 0.921 (0.007) | **0.411 (0.022)** | 0.012 (0.001) | 0.008 (0.001) | -0.370 (0.020) |
| Vanilla (Embed.) | 0.911 (0.006) | 0.905 (0.009) | **0.415 (0.023)** | 0.010 (0.001) | 0.010 (0.001) | -0.373 (0.020) |
| SmoothGrad (Pixel) | 0.949 (0.004) | 0.861 (0.011) | **0.424 (0.023)** | 0.006 (0.001) | 0.014 (0.001) | -0.382 (0.020) |
| SmoothGrad (Embed.) | 0.956 (0.004) | 0.857 (0.011) | **0.423 (0.023)** | 0.005 (0.000) | 0.015 (0.001) | -0.380 (0.020) |
| VarGrad (Pixel) | 0.942 (0.005) | 0.878 (0.010) | 0.387 (0.021) | 0.007 (0.001) | 0.013 (0.001) | -0.350 (0.018) |
| VarGrad (Embed.) | 0.944 (0.005) | 0.875 (0.010) | 0.383 (0.020) | 0.007 (0.001) | 0.013 (0.001) | -0.343 (0.017) |
| LRP | 0.946 (0.004) | 0.869 (0.010) | **0.424 (0.023)** | 0.006 (0.001) | 0.013 (0.001) | -0.381 (0.020) |
| Leave-one-out | 0.970 (0.003) | 0.938 (0.007) | 0.123 (0.031) | 0.009 (0.001) | 0.003 (0.000) | 0.118 (0.020) |
| RISE | 0.978 (0.002) | 0.882 (0.013) | **0.425 (0.023)** | 0.019 (0.003) | 0.002 (0.000) | -0.381 (0.020) |
| **ViT Shapley** | **0.986 (0.002)** | **0.742 (0.013)** | **0.439 (0.023)** | **0.077 (0.003)** | **0.001 (0.000)** | **0.391 (0.019)** |
| Random | 0.957 (0.004) | 0.956 (0.004) | - | 0.005 (0.000) | 0.005 (0.000) | - |

Table 10: Performance metrics for target classes explanations for a ViT classifier trained with random masking. Methods that fail to outperform the random baseline are shown in gray, and the best results are shown in bold (accounting for 95% confidence intervals).

| | ImageNette | | |
|---|---|---|---|
| | Ins. (↑) | Del. (↓) | Faith. (↑) |
| Attention last | 0.963 (0.003) | 0.839 (0.011) | **0.593 (0.017)** |
| Attention rollout | 0.945 (0.004) | 0.903 (0.008) | **0.597 (0.017)** |
| GradCAM (LN) | 0.955 (0.003) | 0.898 (0.011) | **0.585 (0.016)** |
| GradCAM (Attn) | 0.945 (0.005) | 0.942 (0.006) | 0.564 (0.016) |
| IntGrad (Pixel) | 0.977 (0.002) | 0.933 (0.008) | 0.474 (0.017) |
| IntGrad (Embed.) | 0.977 (0.002) | 0.933 (0.008) | 0.474 (0.017) |
| Vanilla (Pixel) | 0.934 (0.005) | 0.901 (0.009) | **0.593 (0.017)** |
| Vanilla (Embed.) | 0.948 (0.004) | 0.864 (0.011) | **0.595 (0.017)** |
| SmoothGrad (Pixel) | 0.966 (0.004) | 0.809 (0.012) | **0.599 (0.017)** |
| SmoothGrad (Embed.) | 0.954 (0.004) | 0.950 (0.005) | **0.597 (0.017)** |
| VarGrad (Pixel) | 0.964 (0.004) | 0.826 (0.011) | **0.573 (0.016)** |
| VarGrad (Embed.) | 0.953 (0.005) | 0.954 (0.004) | **0.594 (0.017)** |
| LRP | 0.972 (0.002) | 0.806 (0.013) | **0.603 (0.017)** |
| Leave-one-out | 0.977 (0.001) | 0.881 (0.014) | 0.134 (0.034) |
| RISE | 0.982 (0.001) | 0.793 (0.017) | **0.599 (0.017)** |
| **ViT Shapley** | **0.988 (0.001)** | **0.736 (0.013)** | **0.605 (0.017)** |
| Random | 0.957 (0.004) | 0.957 (0.004) | - |

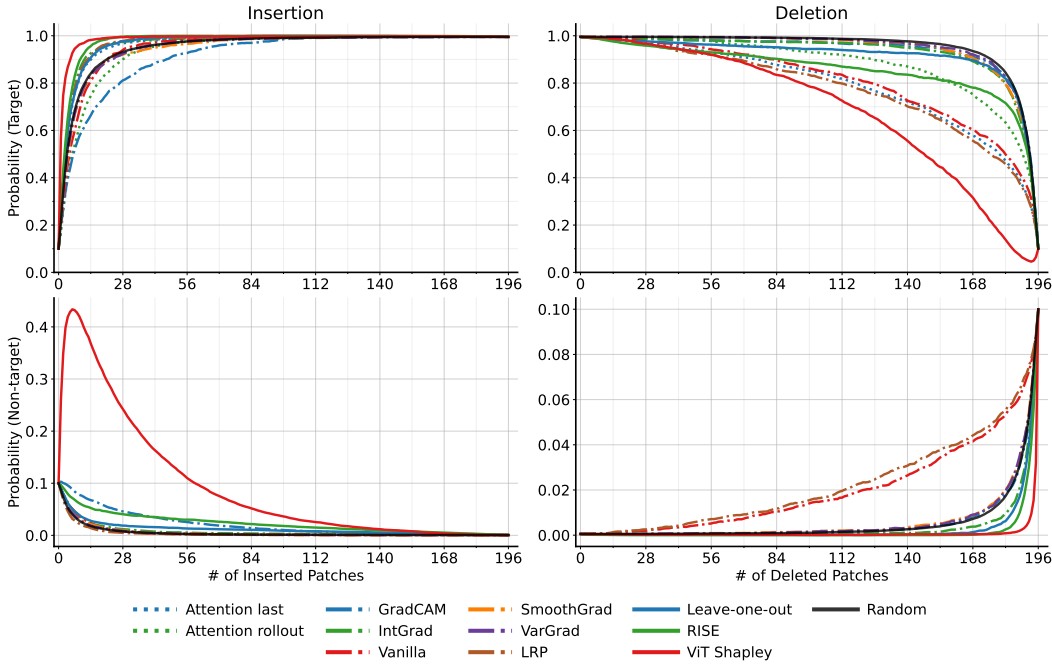

Figure 6: ImageNette average insertion/deletion curves. **Top:** the mean prediction probability for the target class as features are inserted or deleted in order of most to least important. **Bottom:** the mean prediction probability, averaged across all non-target classes.

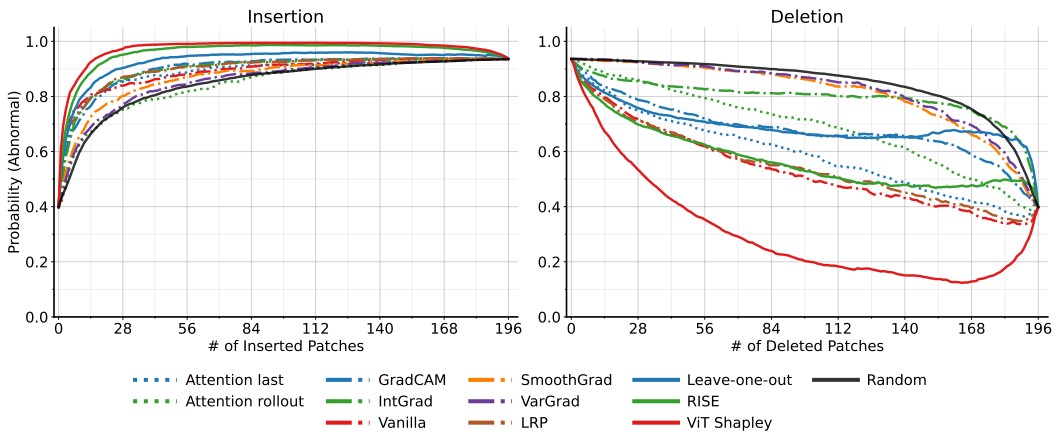

Figure 7: MURA average insertion/deletion curves. The mean prediction probability for the target class is plotted as features are inserted or deleted in order of most to least important.

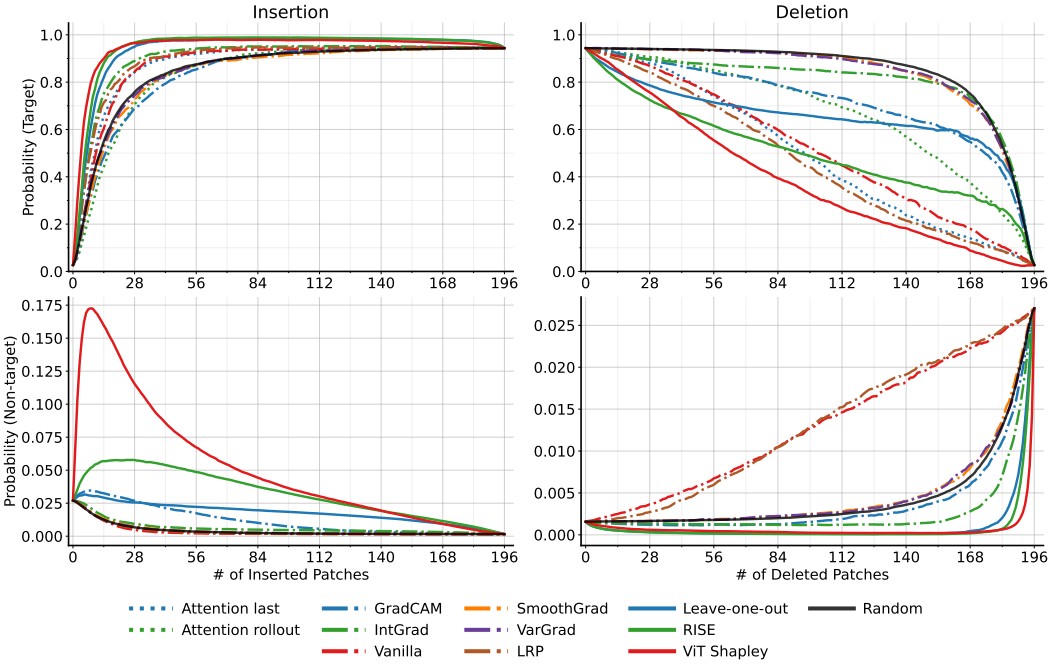

Figure 8: Pets average insertion/deletion curves. **Top:** the mean prediction probability for the target class as features are inserted or deleted in order of most to least important. **Bottom:** the mean prediction probability, averaged across all non-target classes.

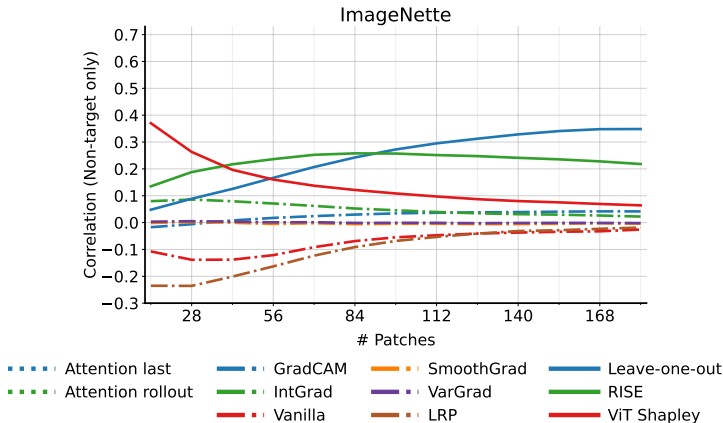

Figure 9: Sensitivity-n evaluation for non-target classes. The results are generated separately for each subset size and averaged across all non-target classes.

Table 11: Time to generate explanations for a single sample (in milliseconds). For class-specific explanations, the running time involves generating explanations for all classes.

|  | ImageNette | MURA | Pets |
|---|---:|---:|---:|
| Attention last | 6.8 | 6.4 | 6.2 |
| Attention rollout | 10.6 | 10.0 | 11.1 |
| GradCAM | 275.9 | 26.2 | 986.4 |
| IntGrad | 1236.8 | 123.3 | 5004.9 |
| Vanilla | 230.1 | 23.0 | 810.7 |
| SmoothGrad | 1218.0 | 121.2 | 4854.8 |
| VarGrad | 1218.9 | 121.3 | 4882.5 |
| LRP | 1551.4 | 155.7 | 5583.5 |
| Leave-one-out | 810.0 | 815.9 | 922.5 |
| RISE | 8213.4 | 8171.1 | 8919.8 |
| **ViT Shapley** | 10.1 | 10.2 | 10.8 |

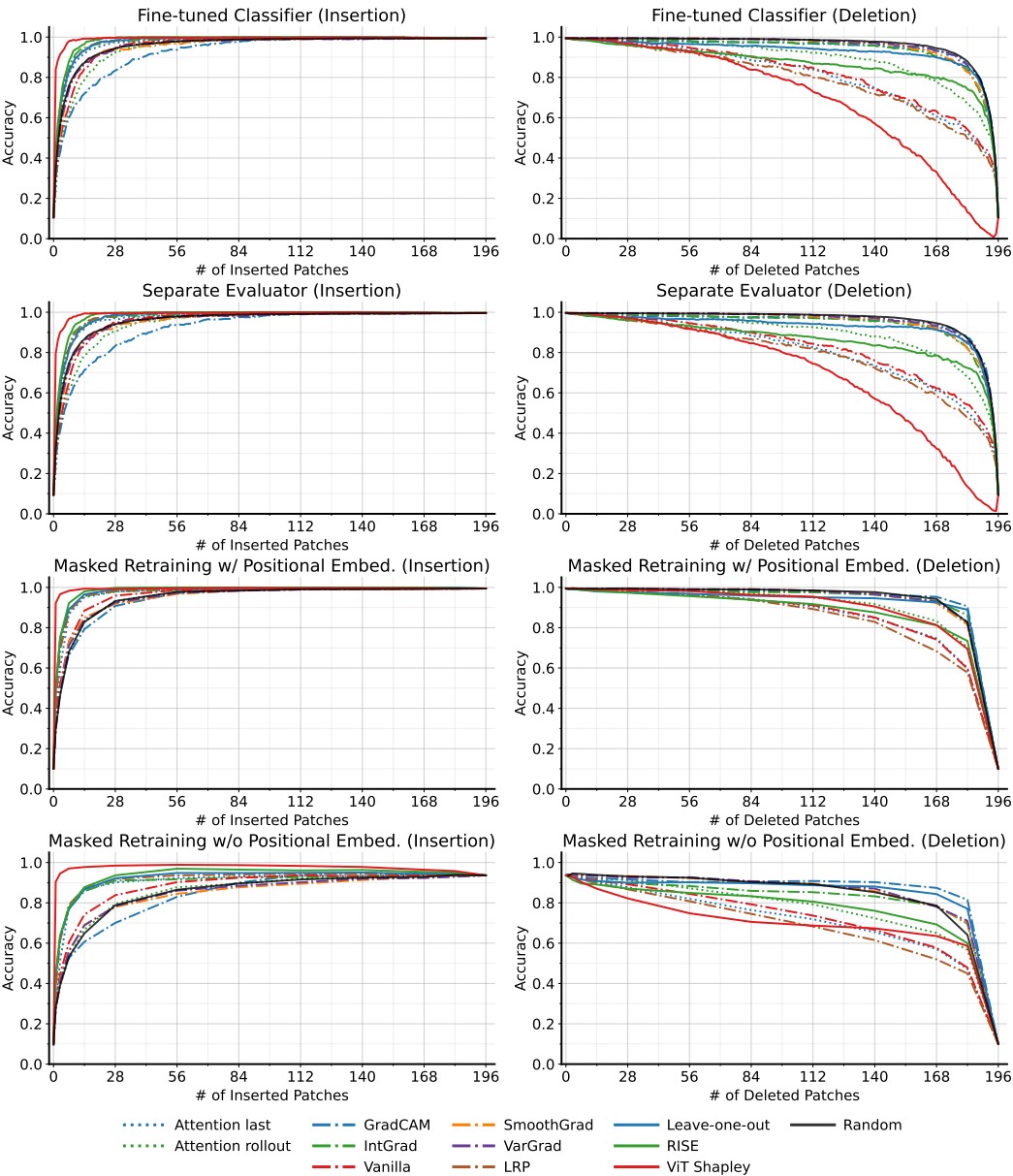

Figure 10: ImageNette accuracy when removing features in order of their importance, run with four evaluation strategies (fine-tuned classifier, separate evaluator, masked retraining and masked retraining without positional embeddings) on the ViT-Base architecture. **Left:** inserting features from most to least important. **Right:** removing features from most to least important.

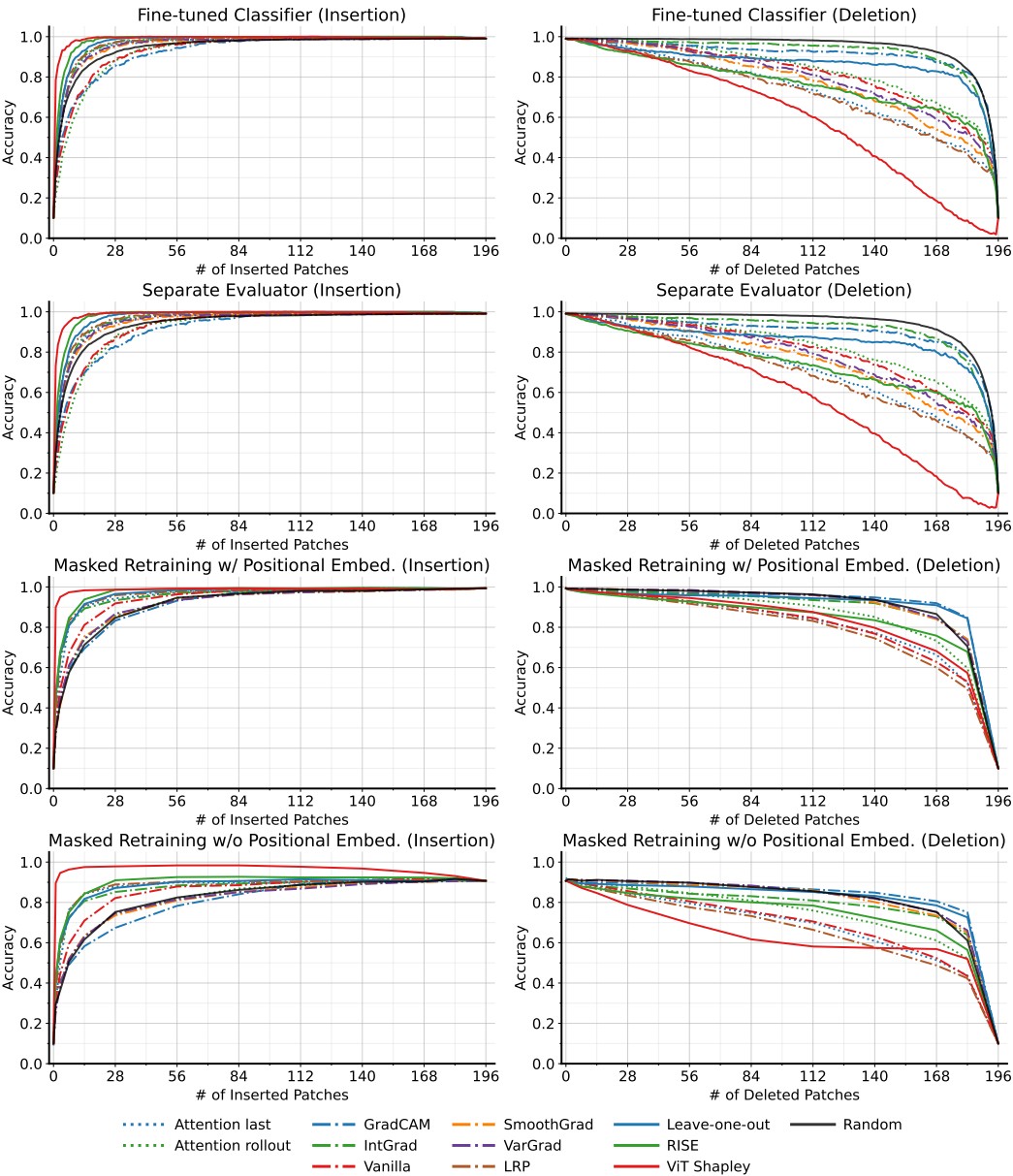

Figure 11: ImageNette accuracy when removing features in order of their importance, run with four evaluation strategies (fine-tuned classifier, separate evaluator, masked retraining and masked retraining without positional embeddings) on the ViT-Small architecture. **Left:** inserting features from most to least important. **Right:** removing features from most to least important.

Table 12: Comparing the quality of Shapley value estimates obtained using ViT Shapley and KernelSHAP via insertion/deletion scores.

|  | Target | | Non-target | |
| --- | --- | --- | --- | --- |
|  | Ins. ($\uparrow$) | Del. ($\downarrow$) | Ins. ($\uparrow$) | Del. ($\downarrow$) |
| ViT Shapley | 0.985 (0.002) | 0.691 (0.014) | 0.093 (0.004) | 0.001 (0.000) |
| KernelSHAP | 0.990 (0.002) | 0.589 (0.053) | 0.158 (0.021) | 0.001 (0.000) |

## H.2  KERNELSHAP COMPARISONS

Here, we provide two results comparing ViT Shapley with KernelSHAP.

First, Figure 12 compares the approximation quality of Shapley value estimates produced by ViT Shapley and KernelSHAP. The estimates are evaluated in terms of L2 distance, Pearson correlation and Spearman (rank) correlation, and our ground truth is generated by running KernelSHAP for a large number of iterations. Specifically, we use the convergence detection approach described by Covert and Lee (2021) with a threshold of $t = 0.1$. The results are computed using just 100 randomly selected ImageNette images due to the significant computational cost.

Based on Figure 12, we observe that the original version of KernelSHAP takes roughly 120,000 model evaluations to reach the accuracy that ViT Shapley reaches with a single model evaluation. KernelSHAP with paired sampling (Covert and Lee, 2021) converges faster, and it requires roughly 40,000 model evaluations on average. ViT Shapley's estimates are not perfect, but they reach nearly 0.8 correlation with the ground truth for the target class, and nearly 0.7 correlation on average across non-target classes.

Next, Table 12 compares the ViT Shapley estimates and the fully converged estimates from KernelSHAP via the insertion and deletion metrics. The fully converged KernelSHAP estimates performed better than ViT Shapley on both metrics, and the gap is largest for deletion with the target class. The results were also computed using only 100 ImageNette examples due to the computational cost. These results reflect that there is room for further improvement if ViT Shapley's estimates can be made more accurate. KernelSHAP itself is not a viable option in practice, as we found that its estimates took between 30 minutes and 2 hours to converge when using paired sampling (Covert and Lee, 2021) (equivalent to roughly 300k and 1,200k model evaluations), but it represents an upper bound on how well ViT Shapley could perform with near-perfect estimation quality.

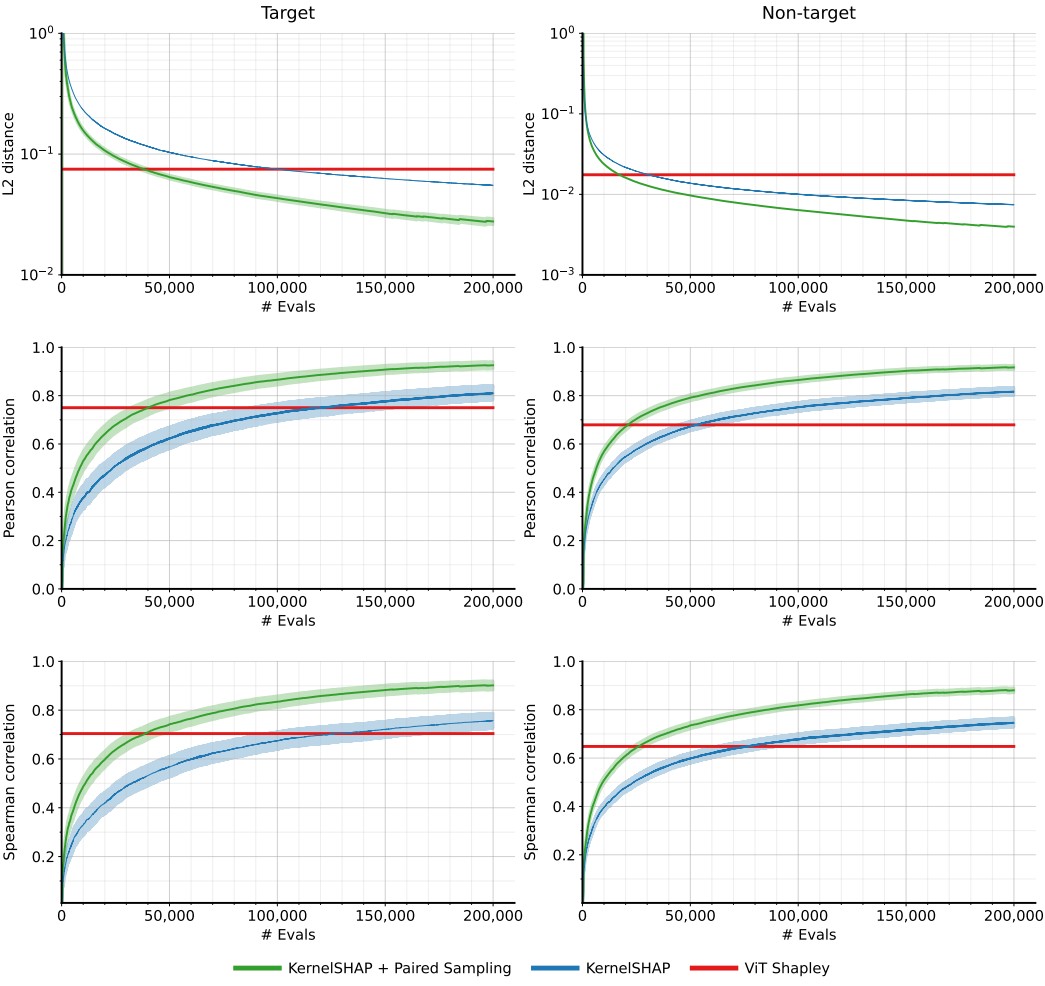

Figure 12: Comparing the quality of Shapley value estimates obtained by ViT Shapley and KernelSHAP. The shaded areas represent 95% confidence intervals.

## I   QUALITATIVE EXAMPLES

This section provides qualitative examples for ViT Shapley and the baseline methods. When visualizing explanations from each method, we used the *icefire* color palette, a diverging colormap implemented in the Python Seaborn package (Waskom, 2021). Negative influence is an important feature for ViT Shapley, and a diverging colormap allows us to highlight both positive and negative contributions. To generate the plots shown in this paper, we first calculated the maximum absolute value of an explanation and then rescaled the values to each end of the color map; next, we plotted the original image with an alpha of 0.85, and finally we performed bilinear upsampling on the explanation and overlaid the color-mapped result with an alpha of 0.9. The alpha values can be tuned to control the visibility of the original image.

Figure 13 shows a comparison between ViT Shapley and KernelSHAP explanations for several examples from the ImageNette dataset. The results are nearly identical, but ViT Shapley produces explanations in a single forward pass while KernelSHAP is considerably slower. Determining the number of samples required for KernelSHAP to converge is challenging, and we used the approach proposed by prior work with a convergence threshold of $t = 0.2$ (Covert and Lee, 2021). With this setting and with acceleration using paired sampling, the KernelSHAP explanations required between 30 minutes to 1 hour to generate per image, versus a single forward pass for ViT Shapley.

Figure 14, Figure 15 and Figure 16 show comparisons between ViT Shapley and the baselines on ImageNette samples. We only show results for attention last, attention rollout, Vanilla Gradients, Integrated Gradients, SmoothGrad, LRP, leave-one-out, and ViT Shapley; we excluded VarGrad, GradCAM and RISE because their results were less visually appealing. The explanations are shown only for the target class, and we observe that ViT Shapley often highlights the main object of interest. We also find that the model is prone to confounders, as ViT Shapley often highlights parts of the background that are correlated with the true class (e.g., the face of a man holding a tench, the clothes of a man holding a chainsaw, the sky in a parachute image).

Similarly, Figure 17, Figure 18 and Figure 19 compare ViT Shapley to the baselines on example images from the MURA dataset. The ViT Shapley explanations almost always highlight clear signs of abnormality, which are only sometimes highlighted by the baselines. Among those shown here, LRP is most similar to ViT Shapley, but they disagree in several cases.

Next, Figure 20, Figure 21, Figure 22, and Figure 23 compare ViT Shapley to the baselines on example images from the Pets dataset. We showed one randomly sampled image per each class (i.e., breed). We observe that ViT Shapley highlights distinctive features of breeds (e.g., the mouth of Boxer or the fur pattern of Egyptian Mau), and rarely puts significant importance on background patches.

Finally, Figure 24 shows several examples of non-target class explanations. These results show that ViT Shapley highlights patches that can push the model's prediction towards certain non-target classes, which is not the case for other methods. These results corroborate those in Table 2 and Table 5, which show that ViT Shapley offers the most accurate class-specific explanations.

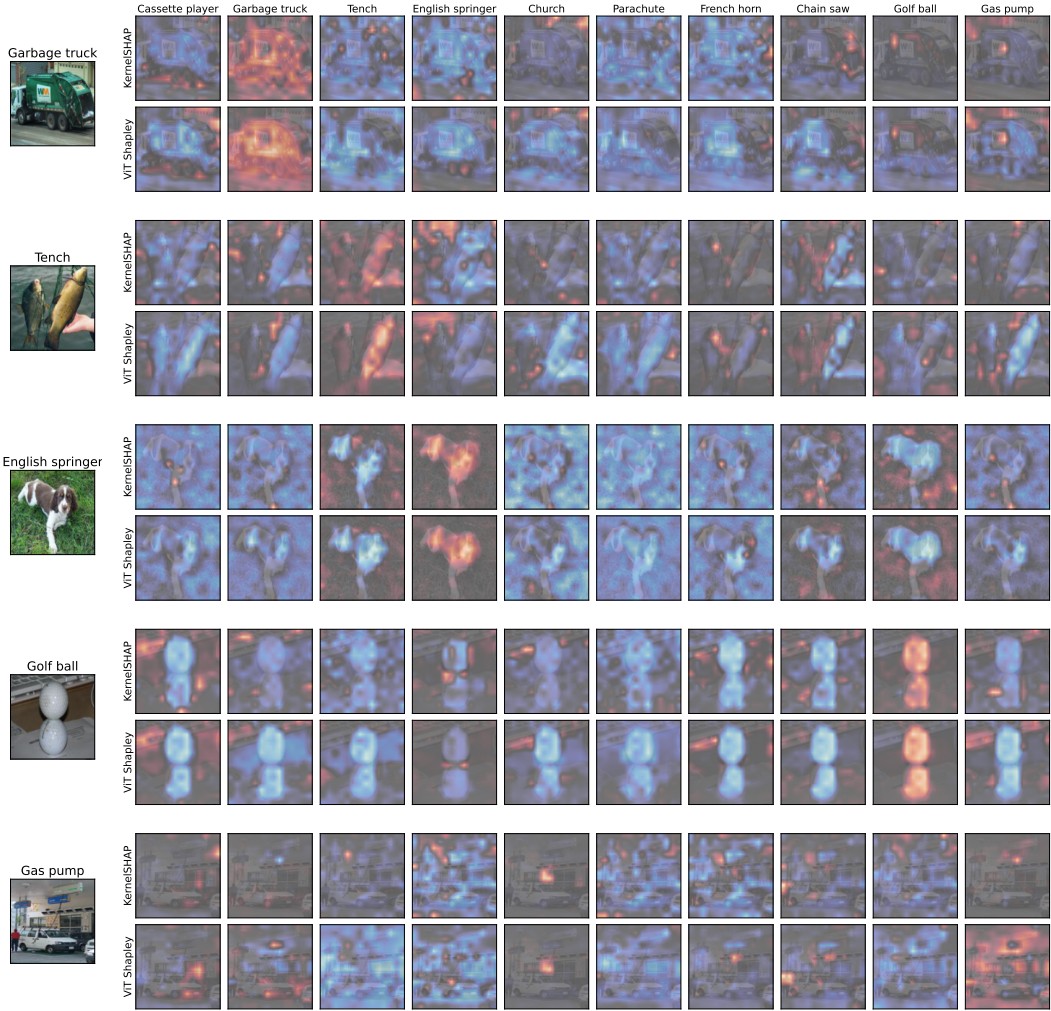

Figure 13: ViT Shapley vs. KernelSHAP comparison (ImageNette).

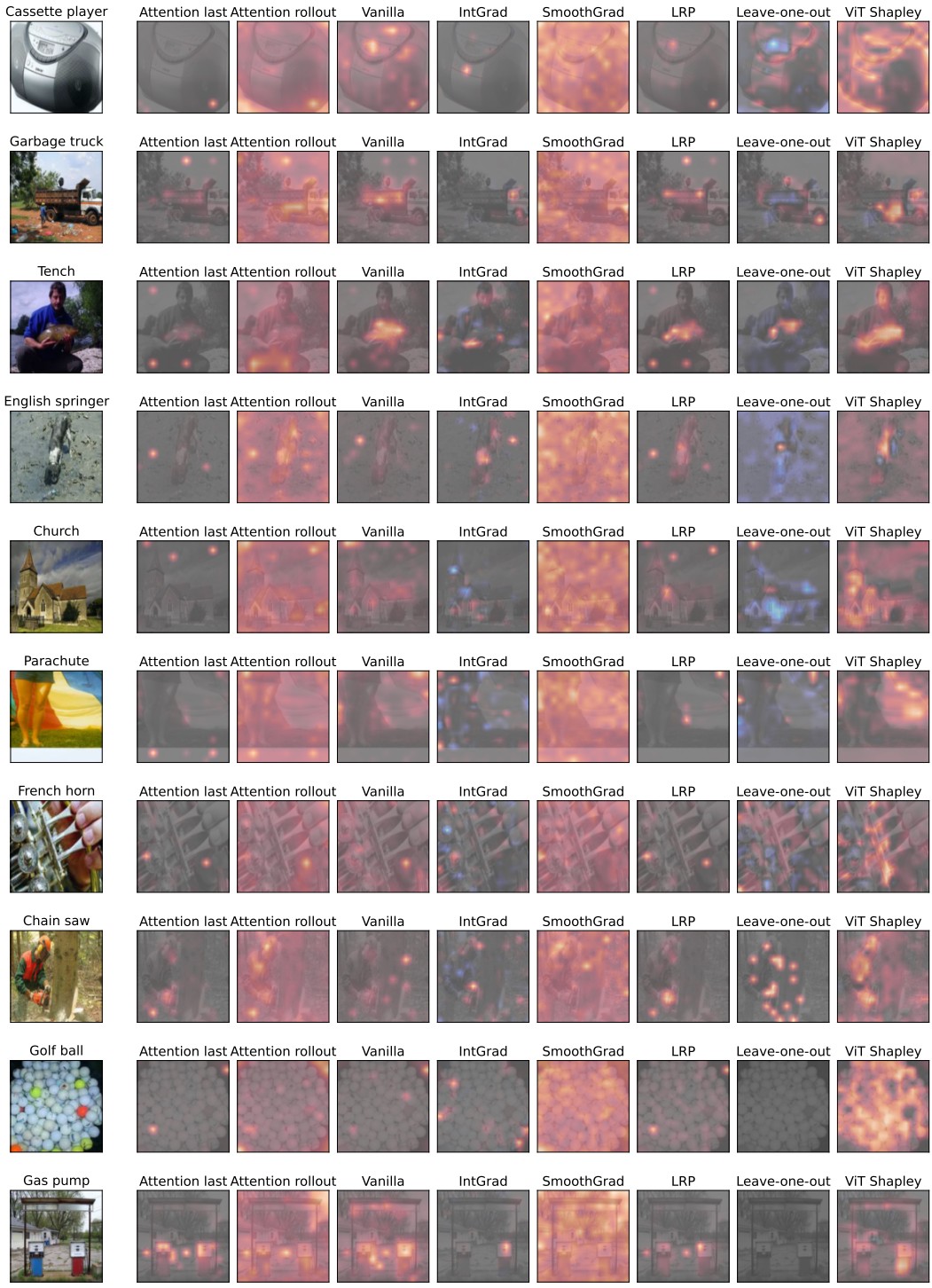

Figure 14: ViT Shapley vs. baselines comparison (ImageNette).

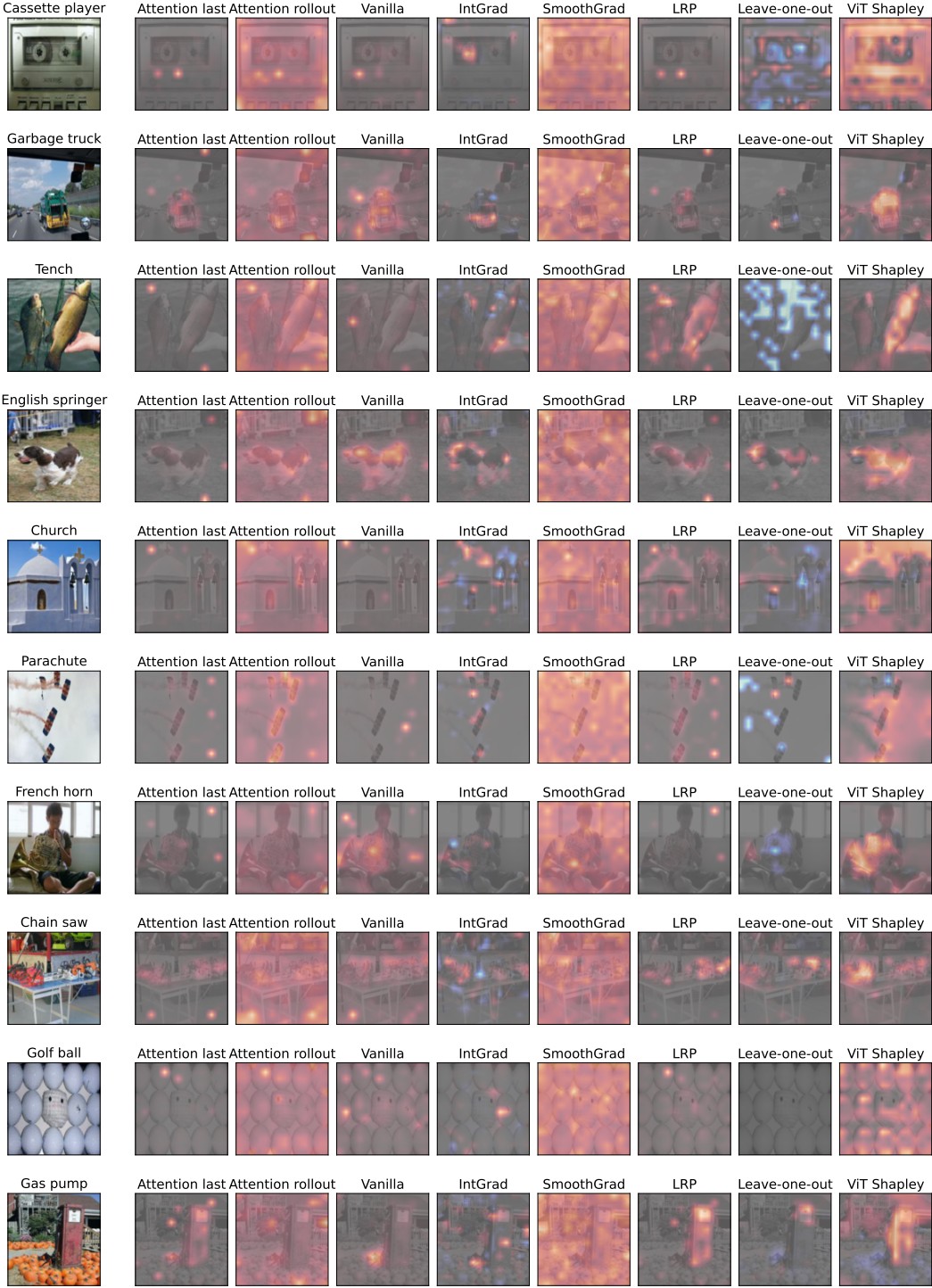

Figure 15: ViT Shapley vs. baselines comparison (ImageNette).

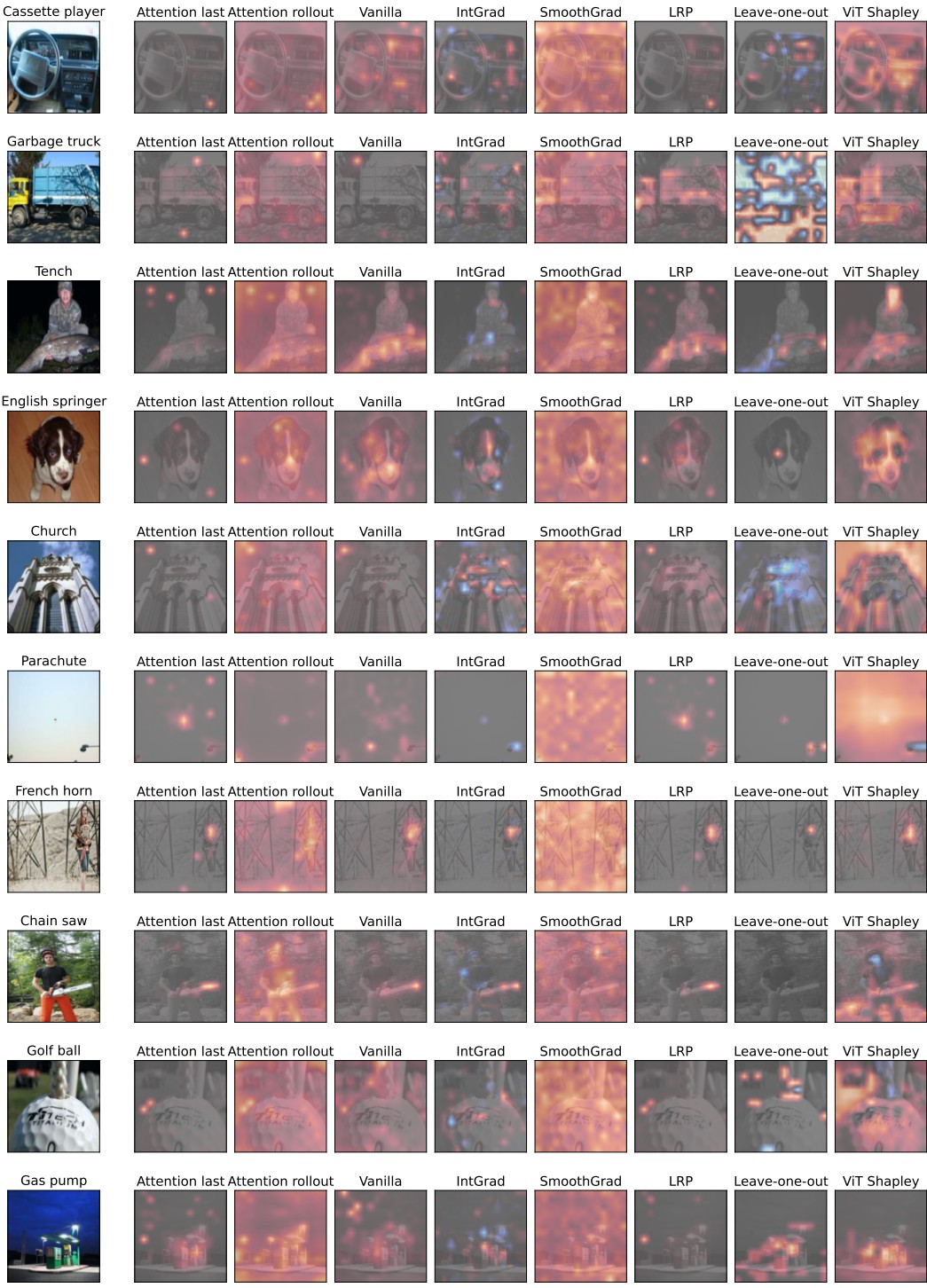

Figure 16: ViT Shapley vs. baselines comparison (ImageNette).

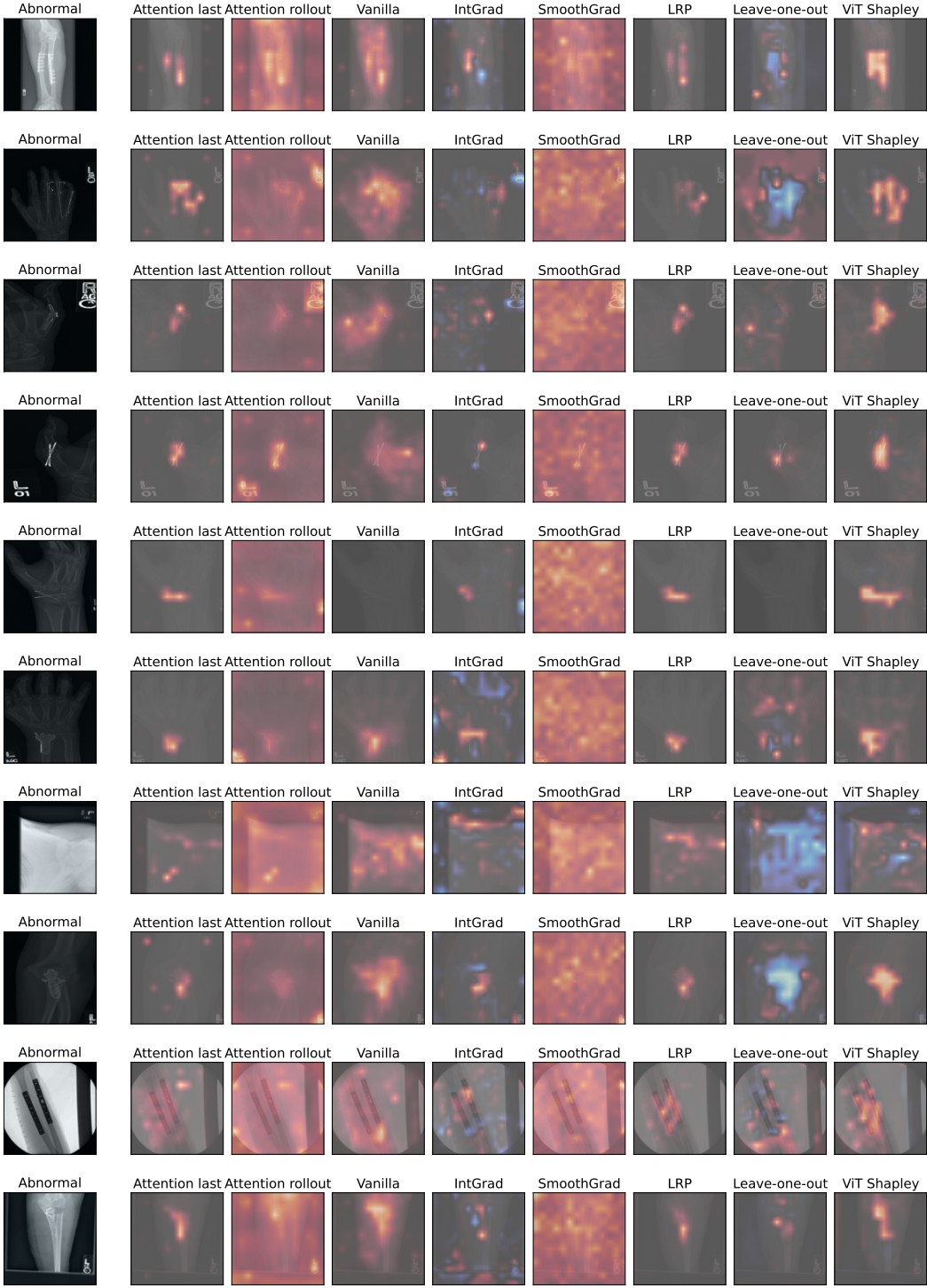

Figure 17: ViT Shapley vs. baselines comparison (MURA).

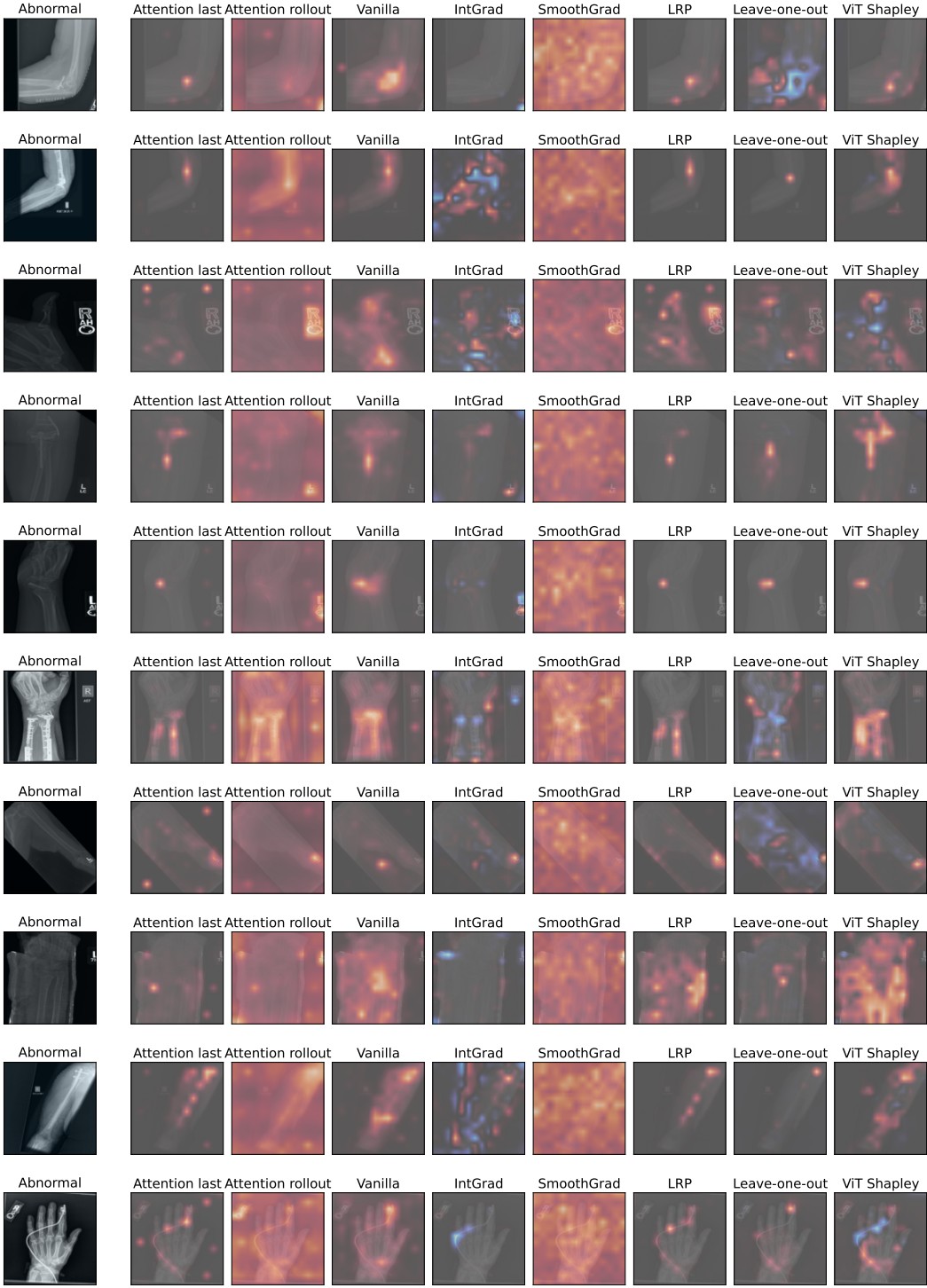

Figure 18: ViT Shapley vs. baselines comparison (MURA).

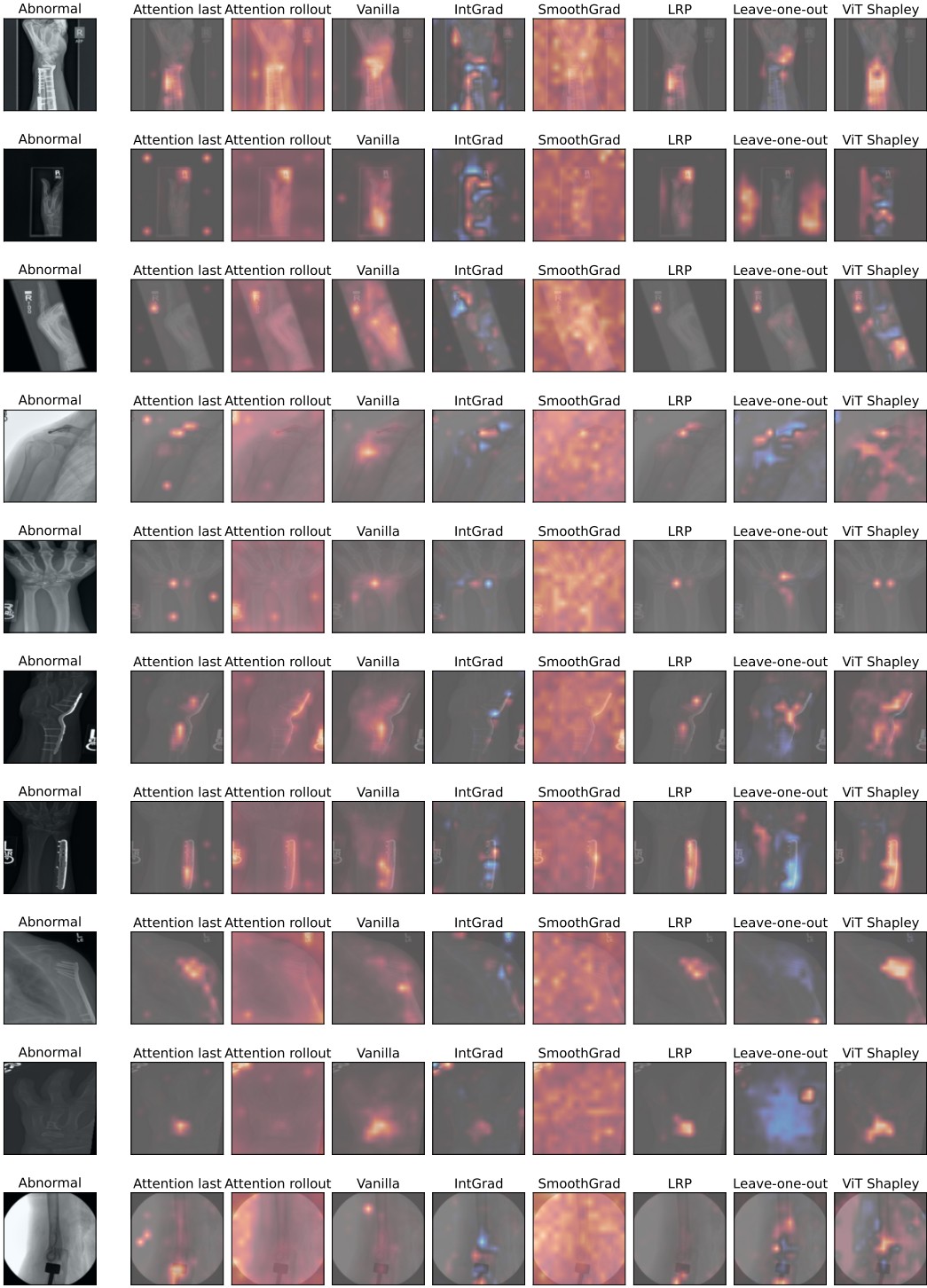

Figure 19: ViT Shapley vs. baselines comparison (MURA).

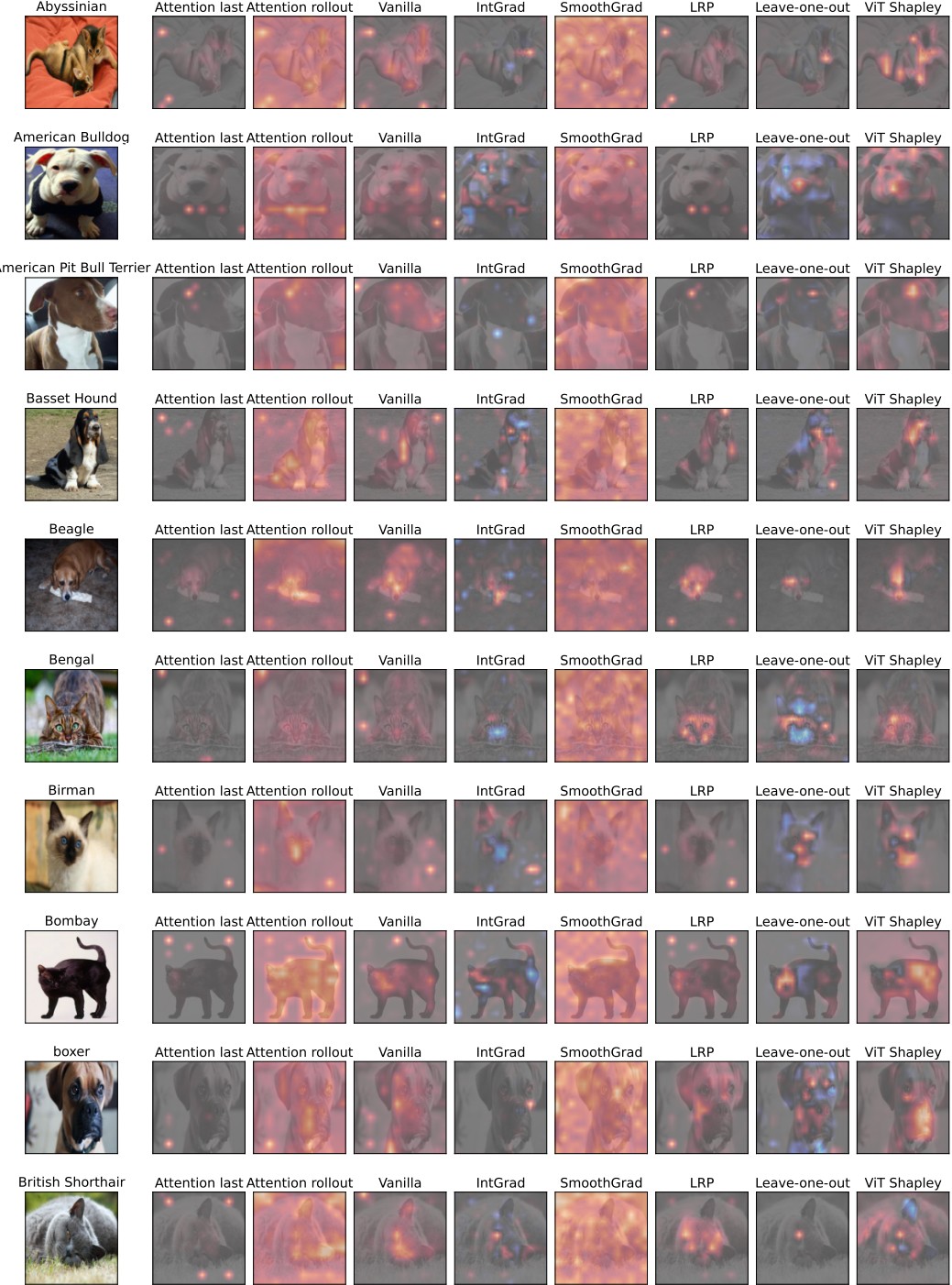

Figure 20: ViT Shapley vs. baselines comparison (Pets).

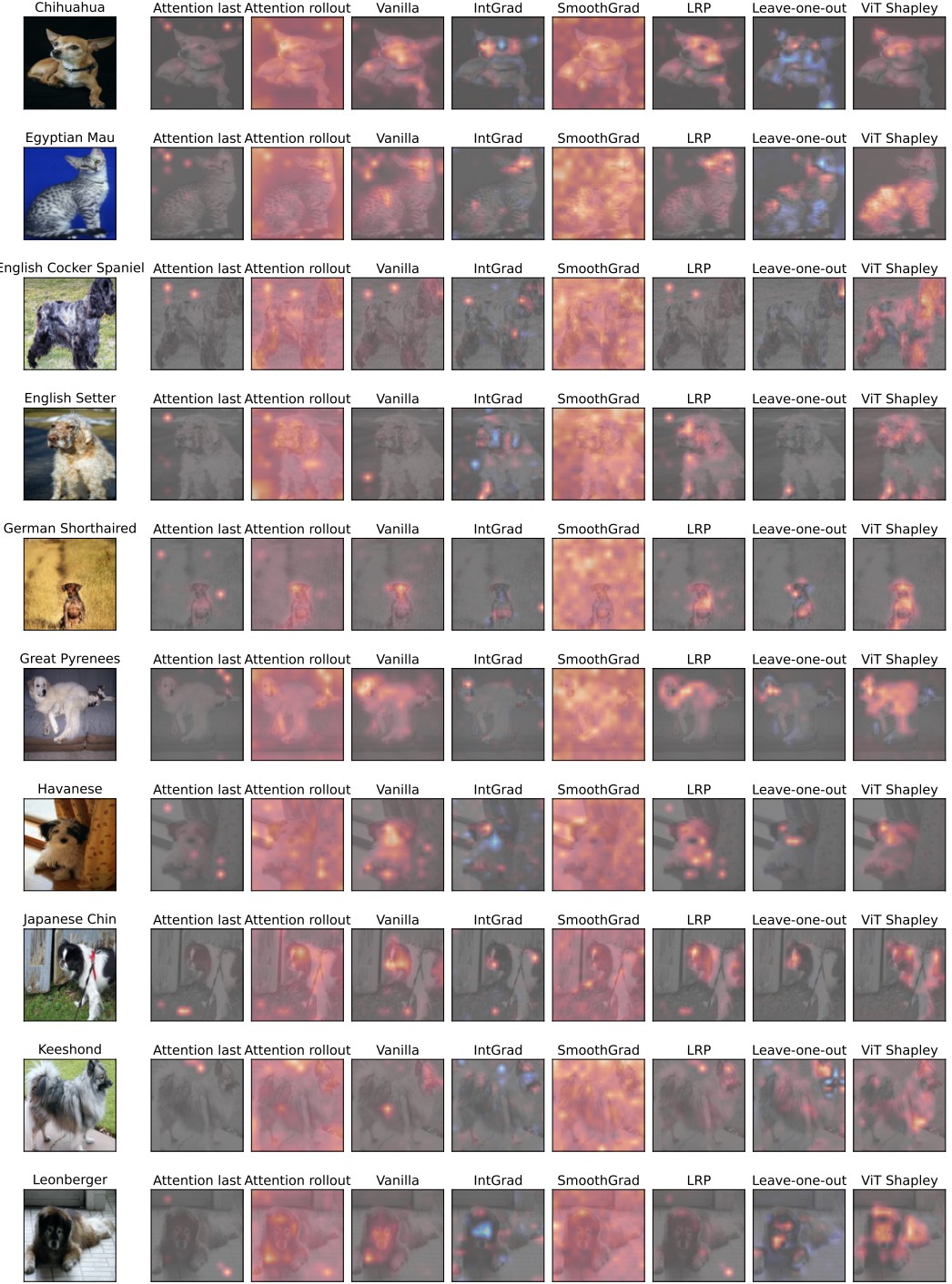

Figure 21: ViT Shapley vs. baselines comparison (Pets).

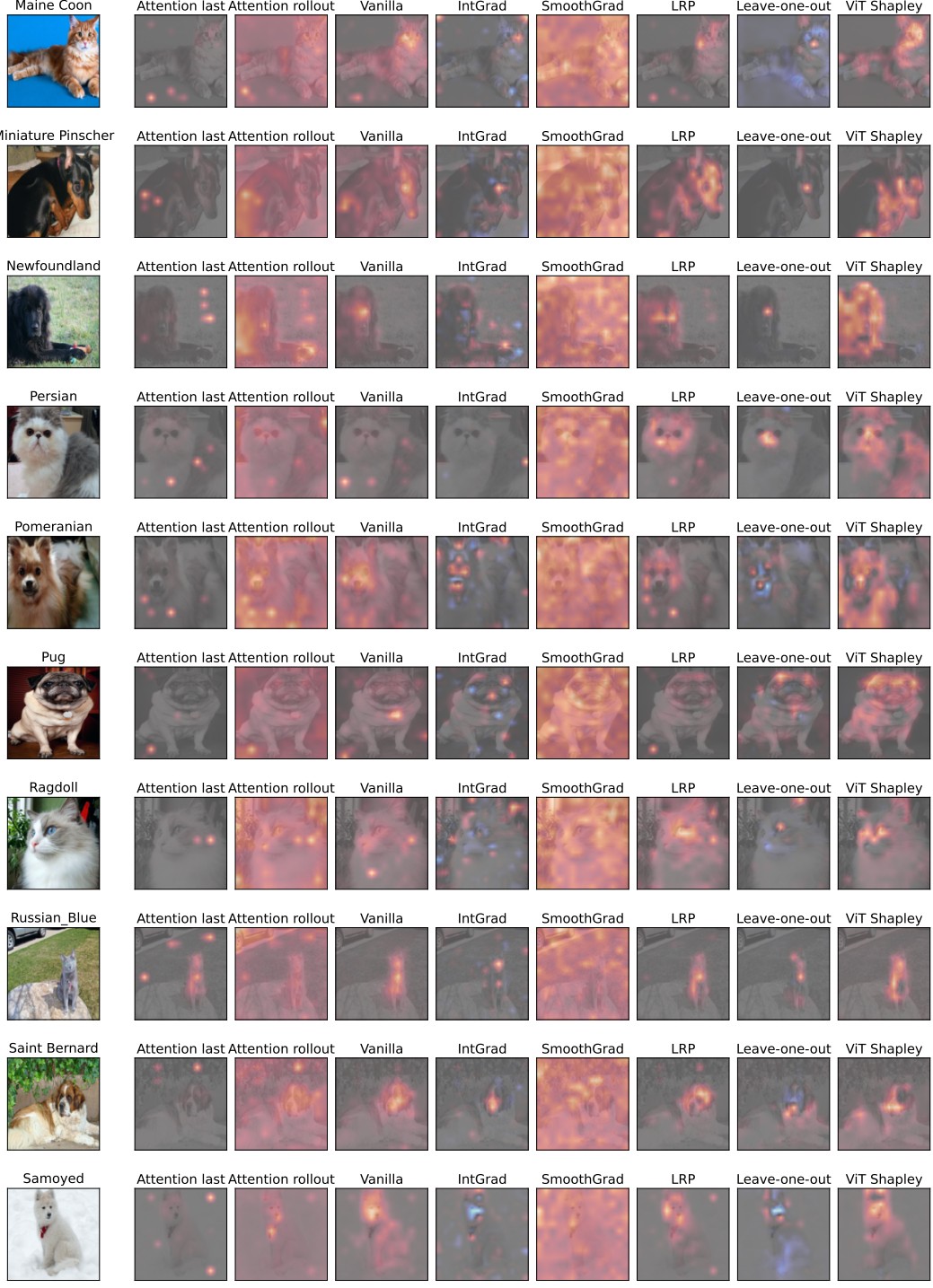

Figure 22: ViT Shapley vs. baselines comparison (Pets).

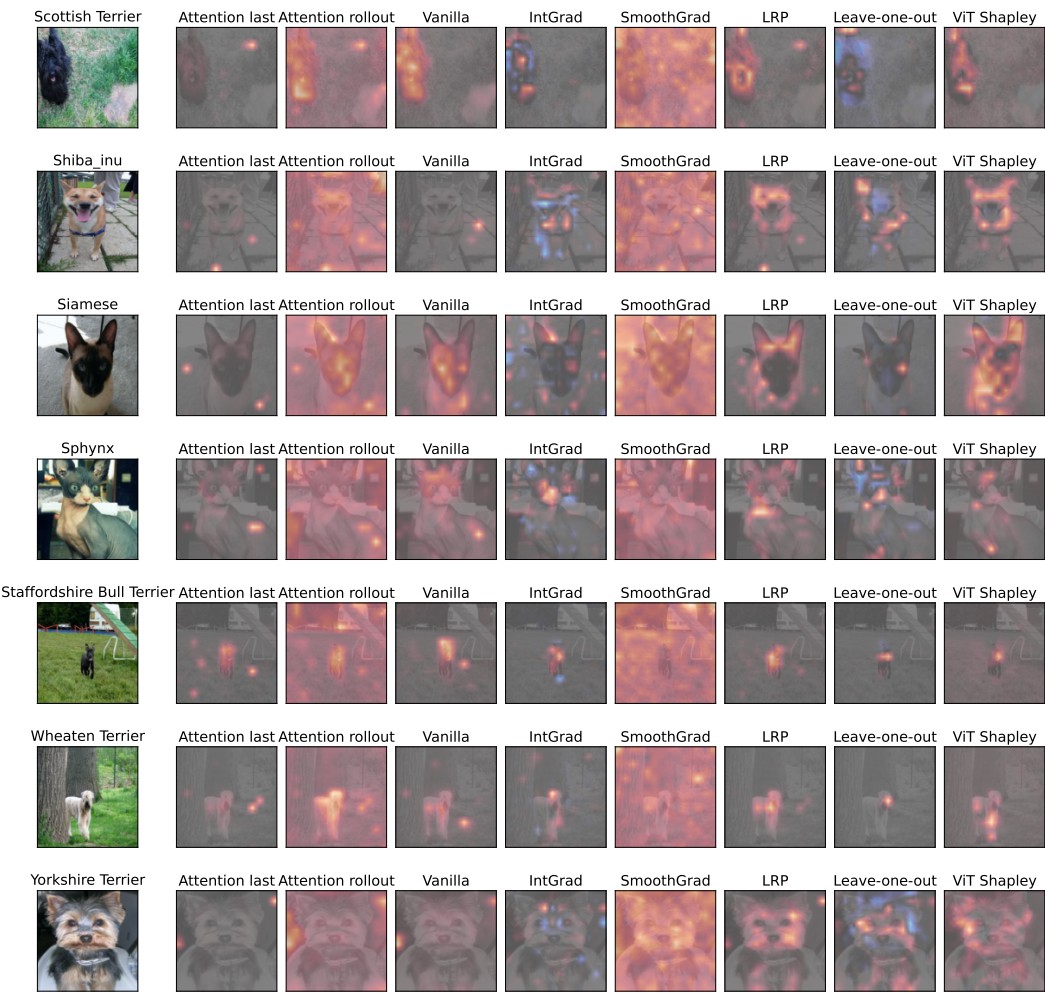

Figure 23: ViT Shapley vs. baselines comparison (Pets).

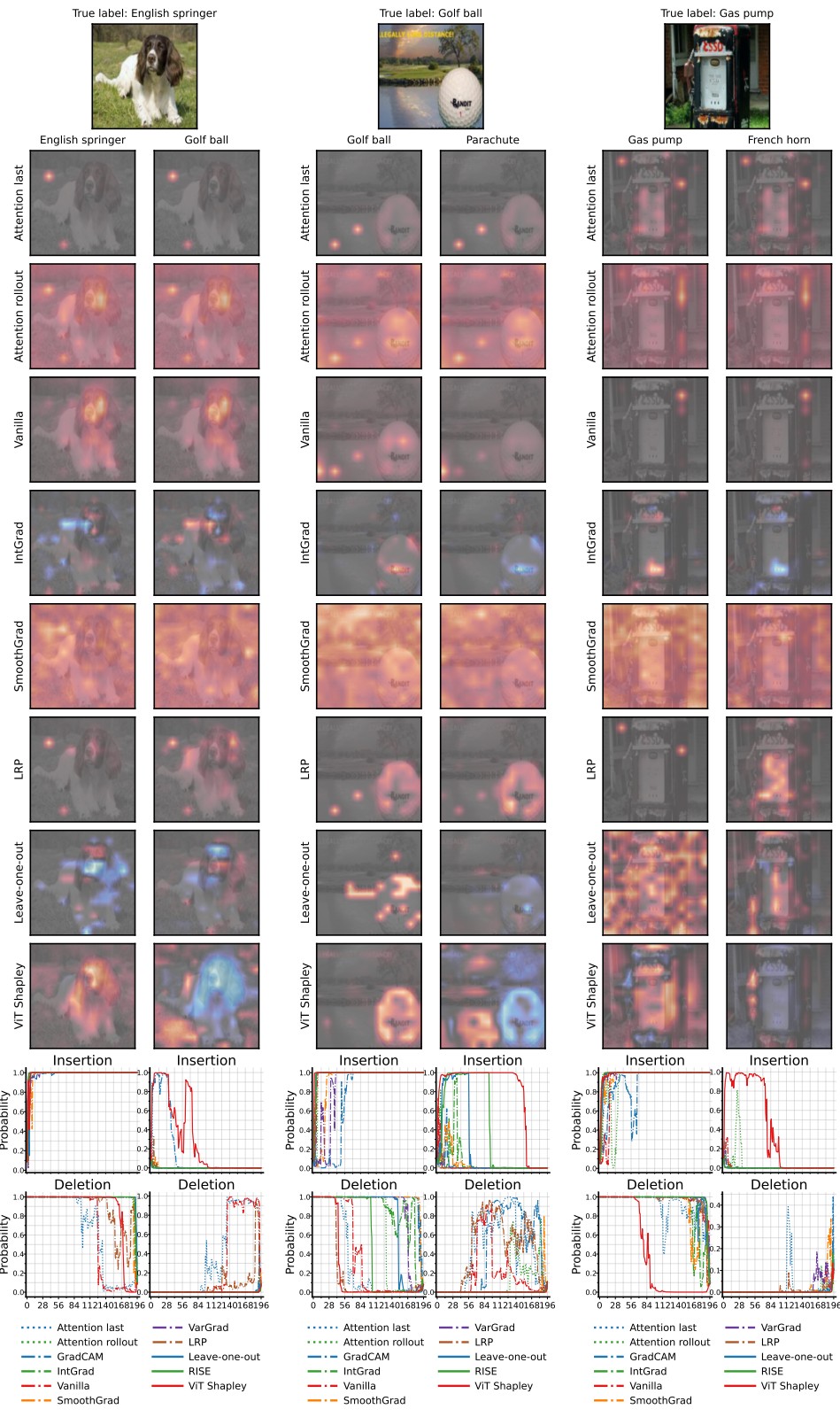

Figure 24: ViT Shapley vs. baselines for non-target classes (ImageNette). **Left:** ViT Shapley shows that the grass in the background provides evidence for the golf ball class. **Middle:** ViT Shapley shows that the sky and its reflection on the water provide evidence for the parachute class. **Right:** ViT Shapley shows that the metallic gas pump handle provides evidence for the French horn class.

