# OpenReview forum: "Learning to Estimate Shapley Values with Vision Transformers"
_ICLR.cc/2023/Conference — ICLR 2023 notable top 25%_

### Official Review · Reviewer_trn3 · 2022-10-16

**Confidence:** 4
**Correctness:** 3
**Technical Novelty And Significance:** 3
**Empirical Novelty And Significance:** 2
**Recommendation:** 6

**Clarity, Quality, Novelty And Reproducibility:**

Clarity is good.

Quality and Novelty are fair.

Reproducibility is unknown.

**Strength And Weaknesses:**

Pros.

1. Confidence intervals of experimental results are provided.

2. Multiple metrics and baseline approaches are considered.

3. The paper is well-written and easy to follow.

Cons.

1. It is very hard to say whether model-agnostic can perform better. More discussions between model-dependent and model-agnostic approaches are needed. In my opinion, involving more ViT-specific information can produce a reliable result.

2. The authors first consider the ViT evaluation with partial information and then leverage a separate, learned explainer model to generate the Shapley value. It seems these two parts are loosely connected.

3. Only one natural image dataset and one medical image dataset are used for experiments. More datasets are needed to support the effectiveness of the paper's proposals.

4. Related works are outdated. More references from 2021 and 2022 should be included.

5. More ViT backbones should be considered.



**Summary Of The Paper:**

Summary.

This paper is dedicated to developing efficient Shapley value approximation for vision transformers. The authors first leverage an attention masking approach to evaluate ViTs with partial information, and then they propose a procedure for generating Shapley value explanations via a separate, learned explainer model. Extensive experiments with diverse comparisons are conducted.

**Summary Of The Review:**

Good idea, but not more discussions and experimental analyses

---

> ### Author Response · Authors · 2022-11-15
> **Reviewer trn3 response**
>
> Thank you for closely examining our work and providing your feedback. We noticed that the pros listed in your review don’t mention some the main contributions of the paper, including our theoretical result bounding the Shapley value estimation error. If you have any questions about these, we are happy to discuss them.
>
> Below, we address the concerns mentioned in your review and how we have attempted to improve the paper.
>
> **Preference for model-specific methods.** You wrote in your review that it’s hard to say whether model-agnostic methods can perform better than model-dependent methods. This seems to invite a philosophical discussion of the two approaches, but we believe an objective and empirical approach to this topic may be more appropriate. Our experiments rely on numerous widely accepted metrics, and the results demonstrate that Shapley values are highly effective. A lengthy discussion of this topic requires more space than we have in our paper, but following your suggestion, we have added the following sentence to our conclusion: “Our experiments provide empirical evidence for the potential for Shapley values as an alternative to attention-based approaches, which remain popular for transformer models.”
>
> Similarly, you wrote “in my opinion, involving more ViT-specific information can produce a reliable result.” We compared ViT Shapley to the main existing attention-based methods, and the results showed that ViT Shapley performs better. Pursuing new attention-based methods is an interesting line of work, but it’s not our focus and there’s nothing more we can do than compare to methods that currently exist.
>
> **Attention masking and explainer are loosely related ideas.** Thank you for pointing this out, it’s a reasonable concern. It’s well known in the Shapley value literature that the two main challenges when computing Shapley-based explanations are 1) properly removing feature information and 2) overcoming the Shapley value’s exponential computational complexity. Our paper is organized around these two challenges, with Section 4 addressing 1) using attention masking and Section 5 addressing 2) with a learning-based estimation approach. The learning-based approach in a way relies on attention masking to query the model with different feature subsets, but ultimately it is only natural that these distinct problems require very different solutions.
>
> To clarify this point in the paper, we cited a recent review paper that clearly describes these two challenges and disentangles how many papers in the literature separately address each one: “Algorithms to estimate Shapley value feature attributions” by Chen et al. (2022).
>
> **Outdated references.** Our paper cites a large number of works that are relevant to our problem, and we would hesitate to remove any of this work because it’s “outdated.” If there are specific works from 2021-2022 that you think should be cited, we would be happy to include them.
>
> **Limited number of ViT backbones.** Our experiments currently test ViT Shapley with two backbones, ViT-Base and ViT-Small. We focused on ViT-Base because it's the most popular backbone (it's #1 and #2 on [Hugging Face](https://huggingface.co/models?pipeline_tag=image-classification&sort=downloads)), and the results with ViT-Small show that the Shapley value function can be estimated even with smaller models. Given that the results are positive for both, we don’t see any reason to expect different performance with other backbones. Note that recent work did not use a large number of backbones: Chefer et al. (2021) only used ViT-Base. However, to alleviate your concern, we are working on adding new results with different ViT backbones, and we will be able to share the results shortly.

---

> > ### Author Response · Authors · 2022-11-16
> > **New ViT backbone**
> >
> > In response to the request for more backbones, we began experiments with ViT-Tiny and ViT-Large using the Imagenette dataset. ViT-Large is still running, but we can share results for ViT-Tiny. Briefly, we replicated the metrics shown in Tables 1 and 2 in the paper (insertion/deletion/faithfulness, for both the true and non-true classes). The results show that with this backbone, ViT Shapley still provides the best performance. This is perhaps surprising, because the Shapley value function should become more difficult to learn with fewer parameters.
> >
> > |                     | Target/Ins. (↑)   | Target/Del. (↓)   | Target/Faith. (↑)   | Non-target/Ins. (↑)   | Non-target/Del. (↓)   | Non-target/Faith. (↑)   |
> > |:--------------------|:-----------------------------|:-----------------------------|:-------------------------------|:-----------------------------|:-----------------------------|:-------------------------------|
> > | Attention last      | 0.917 (0.008)                | 0.702 (0.015)                | 0.649 (0.014)                  | -                            | -                            | -                              |
> > | Attention rollout   | 0.901 (0.009)                | 0.698 (0.015)                | 0.658 (0.015)                  | -                            | -                            | -                              |
> > | GradCAM (Attn)      | 0.898 (0.009)                | 0.814 (0.013)                | 0.636 (0.014)                  | 0.011 (0.001)                | 0.020 (0.001)                | -0.607 (0.012)                 |
> > | GradCAM (LN)        | 0.918 (0.007)                | 0.809 (0.014)                | 0.659 (0.014)                  | 0.034 (0.002)                | 0.006 (0.000)                | -0.616 (0.013)                 |
> > | IntGrad (Pixel)     | 0.928 (0.006)                | 0.852 (0.013)                | 0.293 (0.025)                  | 0.018 (0.002)                | 0.008 (0.001)                | 0.094 (0.014)                  |
> > | IntGrad (Embed.)    | 0.928 (0.006)                | 0.852 (0.013)                | 0.293 (0.025)                  | 0.018 (0.002)                | 0.008 (0.001)                | 0.094 (0.014)                  |
> > | Vanilla (Pixel)     | 0.884 (0.009)                | 0.801 (0.013)                | 0.656 (0.015)                  | 0.015 (0.001)                | 0.020 (0.001)                | -0.629 (0.013)                 |
> > | Vanilla (Embed.)    | 0.902 (0.008)                | 0.752 (0.015)                | 0.659 (0.015)                  | 0.012 (0.001)                | 0.025 (0.001)                | -0.632 (0.013)                 |
> > | SmoothGrad (Pixel)  | 0.908 (0.009)                | 0.733 (0.015)                | 0.659 (0.015)                  | 0.011 (0.001)                | 0.029 (0.002)                | -0.632 (0.013)                 |
> > | SmoothGrad (Embed.) | 0.917 (0.008)                | 0.756 (0.014)                | 0.659 (0.015)                  | 0.010 (0.001)                | 0.025 (0.002)                | -0.632 (0.013)                 |
> > | VarGrad (Pixel)     | 0.914 (0.008)                | 0.754 (0.014)                | 0.648 (0.014)                  | 0.011 (0.001)                | 0.027 (0.002)                | -0.620 (0.012)                 |
> > | VarGrad (Embed.)    | 0.921 (0.008)                | 0.777 (0.014)                | 0.648 (0.014)                  | 0.010 (0.001)                | 0.023 (0.001)                | -0.618 (0.012)                 |
> > | LRP                 | 0.938 (0.007)                | 0.648 (0.016)                | 0.673 (0.014)                  | 0.014 (0.001)                | 0.024 (0.001)                | -0.623 (0.013)                 |
> > | Leave-one-out       | 0.956 (0.004)                | 0.737 (0.019)                | 0.221 (0.039)                  | 0.052 (0.004)                | 0.003 (0.000)                | 0.145 (0.025)                  |
> > | RISE                | 0.970 (0.003)                | 0.602 (0.019)                | 0.658 (0.015)                  | 0.092 (0.005)                | 0.002 (0.000)                | -0.628 (0.013)                 |
> > | ViT Shapley         | 0.981 (0.002)                | 0.457 (0.015)                | 0.694 (0.014)                  | 0.198 (0.006)                | 0.001 (0.000)                | 0.641 (0.011)                  |
> > | Random              | 0.908 (0.008)                | 0.907 (0.008)                | -                              | 0.010 (0.001)                | 0.010 (0.001)                | -                              |
> >
> > We haven’t yet incorporated these results into the paper, but our setup was identical to the other Imagenette experiments except for the classifier and explainer both used the ViT-Tiny backbone.

---

> > > ### Author Response · Authors · 2022-11-19
> > > **ViT-Large**
> > >
> > > Following up on our last update, we're now able to share one more set of results: the same metrics from Tables 1 and 2 but this time using the ViT-Large backbone for Imagenette. Once again, ViT Shapley provides the best results across all the metrics.
> > >
> > > |                     | Target/Ins. (↑)   | Target/Del. (↓)   | Target/Faith. (↑)   | Non-target/Ins. (↑)   | Non-target/Del. (↓)   | Non-target/Faith. (↑)   |
> > > |:--------------------|:-----------------------------|:-----------------------------|:-------------------------------|:-----------------------------|:-----------------------------|:-------------------------------|
> > > | Attention last      | 0.920 (0.009)                | 0.884 (0.016)                | 0.401 (0.022)                  | -                            | -                            | -                              |
> > > | Attention rollout   | 0.922 (0.009)                | 0.932 (0.011)                | 0.423 (0.023)                  | -                            | -                            | -                              |
> > > | GradCAM (Attn)      | 0.938 (0.007)                | 0.922 (0.014)                | 0.395 (0.022)                  | 0.007 (0.001)                | 0.007 (0.001)                | -0.351 (0.018)                 |
> > > | GradCAM (LN)        | 0.956 (0.007)                | 0.918 (0.016)                | 0.424 (0.023)                  | 0.012 (0.002)                | 0.005 (0.001)                | -0.372 (0.020)                 |
> > > | IntGrad (Pixel)     | 0.966 (0.007)                | 0.947 (0.010)                | 0.195 (0.020)                  | 0.006 (0.001)                | 0.004 (0.001)                | 0.114 (0.013)                  |
> > > | IntGrad (Embed.)    | 0.965 (0.007)                | 0.948 (0.010)                | 0.198 (0.020)                  | 0.006 (0.001)                | 0.004 (0.001)                | 0.117 (0.013)                  |
> > > | Vanilla (Pixel)     | 0.898 (0.012)                | 0.918 (0.013)                | 0.411 (0.022)                  | 0.011 (0.001)                | 0.009 (0.001)                | -0.370 (0.020)                 |
> > > | Vanilla (Embed.)    | 0.911 (0.011)                | 0.903 (0.015)                | 0.415 (0.023)                  | 0.010 (0.001)                | 0.010 (0.002)                | -0.373 (0.020)                 |
> > > | SmoothGrad (Pixel)  | 0.946 (0.008)                | 0.862 (0.018)                | 0.424 (0.023)                  | 0.006 (0.001)                | 0.014 (0.002)                | -0.382 (0.020)                 |
> > > | SmoothGrad (Embed.) | 0.958 (0.007)                | 0.856 (0.019)                | 0.423 (0.023)                  | 0.005 (0.001)                | 0.016 (0.002)                | -0.380 (0.020)                 |
> > > | VarGrad (Pixel)     | 0.940 (0.009)                | 0.877 (0.017)                | 0.387 (0.021)                  | 0.007 (0.001)                | 0.012 (0.002)                | -0.350 (0.018)                 |
> > > | VarGrad (Embed.)    | 0.942 (0.008)                | 0.875 (0.018)                | 0.383 (0.020)                  | 0.007 (0.001)                | 0.013 (0.002)                | -0.343 (0.017)                 |
> > > | LRP                 | 0.951 (0.007)                | 0.859 (0.018)                | 0.424 (0.023)                  | 0.006 (0.001)                | 0.014 (0.002)                | -0.381 (0.020)                 |
> > > | Leave-one-out       | 0.970 (0.006)                | 0.938 (0.013)                | 0.123 (0.031)                  | 0.009 (0.002)                | 0.003 (0.000)                | 0.118 (0.020)                  |
> > > | RISE                | 0.978 (0.005)                | 0.879 (0.022)                | 0.425 (0.023)                  | 0.020 (0.004)                | 0.002 (0.000)                | -0.381 (0.020)                 |
> > > | ViT Shapley         | 0.987 (0.004)                | 0.731 (0.021)                | 0.439 (0.023)                  | 0.079 (0.005)                | 0.001 (0.000)                | 0.391 (0.019)                  |
> > > | Random              | 0.956 (0.007)                | 0.956 (0.007)                | -                              | 0.005 (0.001)                | 0.005 (0.001)                | -                              |
> > >
> > > These results aren't yet incorporated in the paper, but we used the same setup as in previous experiments.

---

> ### Comment · Reviewer_trn3 · 2022-11-25
> **Response to the Rebuttal**
>
> Many thanks for all your efforts in rebuttal, especially for the added experiment results. Most of my concerns are solved. I will raise my score.
>
> Meanwhile, for the response "and we would hesitate to remove any of this work because it’s “outdated.”". I never comment to remove any reference. My comment is to ask the author to do a better literature survey to include more recent references during 2021-2022. It is the authors' duty to include both classical and advanced related works!
>
> Best,
>
> Reviewer trn3

---

> > ### Author Response · Authors · 2022-11-27
> > **Thanks**
> >
> > Thank you for taking the time to read our response and adjust your score, we appreciate it! Regarding the related work, apologies if there was a misunderstanding there. Please do let us know if there are related works you think we've missed, we would be happy to take a look.
> >
> > Best,
> >
> > Paper 1982 authors

---

### Official Review · Reviewer_qY28 · 2022-10-23

**Confidence:** 4
**Correctness:** 4
**Technical Novelty And Significance:** 3
**Empirical Novelty And Significance:** 4
**Recommendation:** 8

**Clarity, Quality, Novelty And Reproducibility:**

Well written and clear.
See above.

**Strength And Weaknesses:**

Overall the approach of leveraging ideas from transformer training to compute shapley is interesting and building on the work of Jethani et al. is novel. Authors also present analytical error bounds on shapley approximation which can be useful. The experimental results are convincing.

One possible explanation evaluation approach to consider is the following, where explanations are compared with ground truth:
https://arxiv.org/pdf/2104.14403.pdf

It would be good to discuss how ideas presented in this work could be utilised/extended for text data.

**Summary Of The Paper:**

This work proposes a novel approach to compute Shapley value explanations for vision transformers. To evaluate shapley values for any model we need model's predictions on masked feature inputs. Authors propose a novel way to approximate these by training a ViT model using a loss function designed specifically for Shapley values. The method is evaluated using explainability metrics that do not require ground truth information.

**Summary Of The Review:**

See above.

---

> ### Author Response · Authors · 2022-11-15
> **Reviewer qY28 response**
>
> We would like to thank the reviewer for closely examining our work and providing feedback, and for the largely positive response. We have attempted to address the concerns mentioned in your review, described below.
>
> **Evaluation using ground truth importance.** Thank you for the suggestion, comparisons with ground truth importance are certainly worth discussing. The metrics we used (insertion, deletion, faithfulness, sensitivity-n) are designed to test if explanations highlight features that actually affect the model’s predictions, which makes them well-suited to test an explanation’s correctness for a specific model. The issue with ground truth metrics is that they test the explanation against what the model should depend on, which may not be the case in reality. Thus, such metrics jointly test whether the explanation is correct for the model and whether the model has correct dependencies. The work by Zhou et al. (2021) attempts to address this, but even this approach is imperfect: it cannot guarantee that the model depends on the entirety of the watermark, for example, rather than a part of the watermark.
>
> Thus, we think it is best to use our current set of metrics. We have added some discussion of this point to our appendix section on metrics, where we added a citation for Zhou et al. (2021) – see Appendix G, “Ground truth metrics.”
>
> **Extension to NLP.** As mentioned in the general response, we believe that extending our approach to NLP models is a natural topic for follow-up work, but we did not pursue it here because there are some non-trivial details. Specifically, we think it is important to set up the explainer model so that it outputs word-level Shapley value estimates. This is addressed briefly in our conclusion, which contains the following text: “Future directions involve extending ViT Shapley to transformer models in NLP, operating with arbitrary token groups or superpixels, and accelerating or otherwise improving the explainer model’s training.”

---

### Official Review · Reviewer_eEy2 · 2022-10-24

**Confidence:** 3
**Correctness:** 4
**Technical Novelty And Significance:** 3
**Empirical Novelty And Significance:** Not applicable
**Recommendation:** 8

**Clarity, Quality, Novelty And Reproducibility:**

**Clarity:** The work is written clearly.

**Quality:** The quality is good.

**Reproducibility:** Their approach is well described, including the relevant hyperparameters. However, the authors did not comment whether they plan to release their source code.

**Novelty:** The work is novel. On the more applied side, using the ViT to mask the input partially is novel, and on the theoretical side, Theorem 1 is a novel contribution.

**Minor issues:** All citations are green. When printed in back and white, they are barely readable. Please use the default style.

**Strength And Weaknesses:**

**Strengths:**

- This work provides new theoretical insight into how the optimized objective bounds Shapley values.
- The evaluation is done against a large selection of baselines
- The method outperforms the baselines consistently

**Weakness**:

- The proposed approach could have also been tested on NLP transformer models.
- Training an additional network adds complexity.


**Summary Of The Paper:**

The paper proposes a new way to estimate Shapely values for Vision Transformers. It is required to evaluate the model on partial inputs for estimating Shapely values. For CNNs, this is difficult (e.g. masking areas with gray produces out-of-distribution samples). However, the inputs of vision transformer can be more easily removed, as the attention itself can be masked.



As the estimation of Shapely values is notoriously difficult, the paper proposes to learn them. For this, an existing loss function is used from Jethani et al. (2021). Furthermore, the authors show this loss function to bound the Shapely value estimation error.



The evaluation is done against several baselines on the ImageNette and MURA datasets.

##

**Summary Of The Review:**

This is a good paper. It has both interesting applied contributions (using ViT for masking) and theoretical contributions. Also, the presentation is well done.

---

> ### Author Response · Authors · 2022-11-15
> **Reviewer eEy2 response**
>
> We would like to thank the reviewer for closely examining our work and providing feedback, and for the largely positive response. We have attempted to address the concerns mentioned in your review, described below.
>
> **Extension to NLP.** As mentioned in the general response, we believe that extending our approach to NLP models is a natural topic for follow-up work, but we did not pursue it here because there are some non-trivial details. Specifically, we think it is important to set up the explainer model so that it outputs word-level Shapley value estimates. One naive approach would be to output token-level Shapley values and sum them within word groups, but these would then not be Shapley values. Given that Shapley values are generally appreciated for their mathematical properties, it seems important to give this issue close consideration. We have mentioned it in our conclusion as follows: “Future directions involve extending ViT Shapley to transformer models in NLP, operating with arbitrary token groups or superpixels, and accelerating or otherwise improving the explainer model’s training.”
>
> **Reproducibility.** We can see that you checked our appendix for hyperparameters, thank you for your close attention to detail. Regarding our code, it is available in the supplementary materials and will definitely be released upon publication.
>
> **Difficult to read citations.** Thank you for pointing this out, we apologize for not noticing it ourselves. We have corrected the style used for citations.

---

### Official Review · Reviewer_QpHf · 2022-10-24

**Confidence:** 3
**Correctness:** 4
**Technical Novelty And Significance:** 2
**Empirical Novelty And Significance:** 2
**Recommendation:** 8

**Clarity, Quality, Novelty And Reproducibility:**

- Clarity & Quality

Well written and easy to follow, with a detailed description of the approach.

The writing should make more obvious what is different from the FastSHAP approach, in what respect the fact that an attention-based architecture is studied introduces modification from the original work. Is it just that we can express “attribute” and mask function at the token level instead of the pixel level for images?


- Novelty

It seems to me that the proposed approximation scheme is not limited to a ViT architecture, or even an attention-based architecture. It seems feasible because the input data is represented by a small number of non-overlapping patches, reducing the size of the underlying combinatorics.

- Reproducibility

A link to the code for ViT Shapley has been removed from the submission.

- Miscellaneous questions and comments

Your statement that Attention-based explanations “have not been shown to perform well in vision tasks” should be better motivated.  A better scientific question would have been to state whether attention weights, which introduce soft selection in the decision process,  contain enough information to generate explanation expressed as feature attribution, or not.

It seems to me, as I understand it, that when using a ViT architecture  the number of removable features is the number of patches/tokens ($= d = 196$), which is much smaller than the number of image pixels. Won’t it be possible to compute the exact Shapley values on a few samples for verification of the predictor $\Phi_{ViT}$?

SInce SHAP value prediction results from learning in your approach, what is the impact of sampling on the bound of theorem 1? In other words, the paper provides a bound justifying optimization: can it be extended or modified to account for generalization?

I didn’t understand if the baselines contain the full approach of (Chefer et al., 2021): is it what is called LRP? In fact, I found it quite difficult to compare the performances between the two approaches given that the benchmarks used are different.

The practical use of the approach for real applications seems limited: learning the Shapley value predictor requires a lot of computation. By comparison, (Chefer et al., 2021) seems to me much simpler to implement and requires less computation (no learning, and no extra network at inference time, only gradients and coordinate wise multiplication and sum). Can you elaborate on that?


**Strength And Weaknesses:**

== Strengths ==


- Well funded mathematical result, at least seems to be (I didn’t check the proofs in the appendix).

- Reasonable number of experiments to justify the approach, although on small size image databases (ImageNette & MURA).


== Weaknesses ==

- Limited novelty: rather straightforward application of FastSHAP to a single attention-based architecture, ViT (see the “novelty” section).

- No explicit exploitation of attention information as done in  (Chefer et al., 2021), for instance. Attention is only exploited for the optimization of the Shapley value estimator.

- Small size of the datasets used for evaluation.


**Summary Of The Paper:**

The paper builds on the FastSHAP approach (Jethani et al., 2021) and applies it to the ViT attention-based architecture used in image analysis. The goal of FastSHAP is to learn a dedicated predictor to estimate the Shapley values expressing input feature attribution for explainability purposes. Another contribution is a mathematical justification of the FastSHAP loss used to estimate the Shapley value predictor. The approach is evaluated on two small-size datasets using insertion, deletion and faithfulness metrics.


**Summary Of The Review:**

The main contribution of the paper is to apply the FastSHAP approach (Jethani et al., 2021) to ViT architecture, an attention-based model, and to propose a mathematical justification that the loss used to estimate the Shapley value predictor.

The application to ViT architecture is rather straightforward: deactivate the feature/tokens using the attentional weights. The mathematical justification of the loss seems well funded but is independent from the attention-based architecture. The approach is evaluated on small size databases using standard metrics but is not compared, as it seems to me, to another feature attribution study that explicitly exploits attention to build the explanation (Chefer et al., 2021).

The paper is well written and clear, but the content, for me, is rather at the level of a specialized workshop given the nature of its novelty (application of a known approach to another neural architecture and small theoretical contribution).

---

> ### Author Response · Authors · 2022-11-15
> **Reviewer QpHf response (1/2)**
>
> Thank you for closely examining our work and providing your feedback. We appreciate the numerous good points raised in your review, and we have attempted to address several of them in our revisions to the paper.
>
> **Comparison with Chefer et al. (2021).** As mentioned in the general response, we can confirm that the method from Chefer et al. (2021) was used in our experiments – it’s indeed the method that we refer to as LRP. We noticed that other papers have referred to the method differently, but we chose this name because the paper described itself as extending LRP to transformer models. We apologize if this naming caused any confusion.
>
> **Clarification of claim that attention methods don’t work well.** In your review, you suggested that we take a different approach in claiming that attention methods don’t work well: emphasizing that attention weights do not contain enough information to explain a model’s dependencies. We have ensured that this point is reflected, and our related work section now contains the following sentence: “Attention is a popular interpretation tool, but it is only one component in a sequence of nonlinear operations that provides an incomplete picture of a model’s dependencies (Serrano and Smith, 2019; Jain and Wallace, 2019; Wiegreffe and Pinter, 2019), and direct usage of attention weights has not been shown to perform well in vision tasks (Chefer et al., 2021).” We would also like to clarify that Chefer et al. (2021) is cited here due to their experiments showing that attention from the last layer and attention rollout perform poorly on common metrics (similar to our experiments).
>
> **Differences with FastSHAP.** Our general response enumerates the main contributions of our work and why it is not a straightforward application of FastSHAP. Two of these points were mentioned in our paper’s list of contributions (attention masking and our new bound for the Shapley value estimation error), and for the remaining point we have edited Section 5 to include the following text: “Next, rather than training the explainer from scratch, we fine-tune an existing model that can be either the original classifier or a ViT pre-trained on a different supervised or self-supervised learning task (Touvron et al., 2021; He et al., 2021); ViTs are more difficult to train than convolutional networks, and we find that fine-tuning is important to train effectively (Table 3).” We hope this clarifies the differences with the original FastSHAP work.
>
> **Not exploiting attention mechanism.** In your review, you wrote that attention was only exploited during optimization of the explanation model. This is correct, we exploit the ViT’s self-attention only as a mechanism for withholding feature information. It does not seem to us that attention values are otherwise useful for calculating Shapley values, and particularly for navigating the exponential computational complexity. It’s possible that they are, and there could possibly be an attention-based layer-wise approximation similar to DeepSHAP, but that’s not the focus of our paper and would have to be the subject of future work.
>
> **Limitation to ViT architecture.** The learning-based approach and our characterization of the Shapley value estimation error (Theorem 1) are indeed applicable to beyond ViT models. Our attention masking approach (Section 4) is specific to transformer models, but not limited to ViTs only. As mentioned in the general response, extending ViT Shapley to NLP models is a natural direction for follow-up work, but some modifications are required (parameterizing the explainer to output word-level explanations).

---

> > ### Author Response · Authors · 2022-11-15
> > **Reviewer QpHf response (2/2)**
> >
> > **Shapley value calculation given “only” 196 patches.** This is a good observation – we had the same idea that Shapley values may be possible to calculate efficiently here given the relatively small number of image patches. We of course cannot perform a brute force calculation using all $2^{196}$ subsets, but we may expect a statistical estimator like KernelSHAP to converge without requiring too many iterations. We tested this and provided several experiments in Appendix H. Briefly, KernelSHAP does not converge quickly and seems to require >100,000 model evaluations to converge. These results are mentioned at the end of the experiments section, and we also have a footnote in Section 5 that contains the following text: “The number of model evaluations depends on how fast the estimators converge, and we find that KernelSHAP requires >100,000 samples to converge for ViTs (Appendix H).”
> >
> > **Questions about Theorem 1.** Like you said, our bound is intended to justify optimizing using $\mathcal{L}(\theta)$ by showing that it’s equivalent to minimizing an upper bound on the Shapley value estimation error. The fact that we optimize while performing sampling is harmless – it only means that we’re optimizing with noisy gradients rather than exact gradients (i.e., we’re performing SGD rather than GD). We are not sure how to interpret your question about generalization, but we can clarify that the bound applies specifically to data that follows the original distribution $p(x, y)$.

---

> ### Comment · Reviewer_QpHf · 2022-11-28
> **Response to rebuttal.**
>
> I would like first to thank the authors for providing detailed and clear answers to my concerns. Most of them have been addressed. Here are a few remaining comments.
>
> * Regarding my first criticism about novelty, I am still not convinced that the attention masking approach is really more than a technical trick allowed by the Transformer architecture where held-out input tokens can be globally switched-off (or equivalently removed)  by hard attention. I am not sure, for instance, that other more intertwined attentional architectures like SWIN, could benefit from the same strategy.  The fact that "Jethani et al. (2021) did not consider" Transformers cannot be opposed as a real limitation, as the original FastSHAP approach is agnostic to architecture.
>
> * The other main contribution of the paper is a theoretical bound justifying the approximating loss used in FastSHAP, which is an interesting result for this method. I had an open question - not very clearly expressed, I agree - whether the bound could be extended to introduce issues of statistical significance or generalization (PAC learning like), but this may not be feasible.
>
> * I read the other reviews and mostly agreed with reviewer trn3 concerns (model specificity, limitation of image datasets, connection between learning Shapley values and ViT) which have been answered satisfactorily by the authors.
>
> * I will raise my rating, but still  think that the legacy to FastSHAP (Jethani et al., 2021) is high and should be made more explicit in order to make clearer the two main independent contributions - theoretical bound for the loss and application to ViT.

---

> > ### Author Response · Authors · 2022-11-29
> > **Attention masking and generalization**
> >
> > Thanks very much for reading our response and updating your review, we appreciate it! We are glad to hear that you found our answers satisfactory. Regarding your last couple comments, we may be able to provide satisfactory answers to these as well.
> >
> > First, regarding the attention masking idea, you’re correct that this is a mechanism to switch off input tokens for transformers. It’s an important part of our paper though, because figuring out the right approach for held-out features is half the challenge of using Shapley values (the other half is navigating the exponential complexity). Here, we not only propose attention masking as a natural approach to use with ViTs, but we show that fine-tuning the classifier is important for the predictions with partial inputs to approximate marginalizing out missing features (Section 6.1).
> >
> > Like you said, the fact that Jethani et al. (2021) didn’t consider transformers isn’t necessarily a shortcoming. That work was more focused on the loss function for training the explainer, or navigating the exponential complexity of Shapley values. As for the separate question of how to handle held-out features, it seems reasonable that different models may offer different approaches. This last point relates to your comment about SWIN architectures: we agree that our attention masking approach would need to be modified for SWIN models, as our formulation is focused on the original ViT architecture. However, note that many transformer-specific methods don’t immediately apply to SWIN models, including attention in the last layer, attention rollout, and LRP (Chefer et al., 2021) – along with ViT Shapley, these all of these require some degree of extension.
> >
> > Finally, regarding your question about using Theorem 1 in a generalization context, we think this is possible and we can provide a sketch here. As we understand your question, the goal is to understand the Shapley value estimation error, which can be defined for a model $\phi_{ViT}(x, y; \theta)$ as follows:
> >
> > $\text{SVE} = \mathbb{E} [ || \phi_{ViT}(x, y; \theta) - \phi(v_{xy}) ||_2]$
> >
> > To estimate $\text{SVE}$, one option is to use an external dataset (e.g., the test data in our setup) consisting of samples $(x_i, y_i)$ for $i = 1, \ldots, n$, calculate their exact Shapley values $\phi(v_{x_iy_i})$, and generate a Monte Carlo estimate $\hat{\text{SVE}}_n$.
> >
> > Then, using standard concentration inequalities like Chebyshev or Hoeffding (the latter applies only if we assume an upper bound on the error), we can then get probabilistic bounds of the form $P(| \hat{\text{SVE}}_n - \text{SVE}| > \epsilon) \leq \delta$. This is useful but perhaps computationally costly, because it requires ground-truth Shapley values for the external data. Instead, you can use our Theorem 1 result to bypass the need for ground-truth Shapley values.
> >
> > For this, recall that $\mathcal{L}(\theta)$ represents our weighted least squares loss function. If we know $\mathcal{L}(\theta)$ exactly, Theorem 1 yields the following bound with probability 1:
> >
> > $\text{SVE} \leq \sqrt{2 H_{d - 1} ( \mathcal{L}(\theta) - \mathcal{L}^*)}$
> >
> > If we don’t know $\mathcal{L}(\theta)$ exactly, we can instead use a Monte Carlo estimate $\hat{\mathcal{L}}(\theta)_n$ calculated using samples $(x_i, y_i, s_i)$ for $i = 1, \ldots, n$. Then, again using standard concentration inequalities, we can get bounds of the form $P(|\hat{\mathcal{L}}(\theta)_n - \mathcal{L}| > \epsilon) \leq \delta$. With those, we can say with probability at least $1 - \delta$ that $\mathcal{L}(\theta) \leq \hat{\mathcal{L}}(\theta)_n + \epsilon$.
> >
> > Finally, combining a couple steps from our Theorem 1 proof gives us the following bound with probability at least $1 - \delta$:
> >
> > $\text{SVE} \leq \sqrt{2 H_{d - 1} (\hat{\mathcal{L}}(\theta)_n - \mathcal{L}^* + \epsilon)}$
> >
> > That’s a rough sketch of the relevance to generalization. Naturally, $\delta$ will be a function of $\epsilon$ and depend on the number of samples $n$ used to estimate $\hat{\mathcal{L}}(\theta)_n$, with the rate of convergence to probability 1 depending on whether we use the Chebyshev or Hoeffding inequality.
> >
> > We are happy to add this discussion to appendix D. However, this arguably isn’t the best use-case of our bound because we don't have access to $\mathcal{L}^*$ in practice. So perhaps the more important points are 1) that minimizing $\mathcal{L}(\theta)$ is equivalent to minimizing an upper bound on $\text{SVE}$, thus justifying our training approach, and 2) that $\hat{\mathcal{L}}(\theta)_n$ can be used as a validation loss to perform model selection (this is effectively what we did in our experiments).

---

> > > ### Comment · Reviewer_QpHf · 2022-12-01
> > > **Last comment**
> > >
> > > Thank you for short time reaction to my last comment.
> > >
> > > * I agree that SWIN architectures are rather new and complex objects, and that explaining their behavior is still to be done. My concern about ViT was that the attention masking proposed in your work is perhaps a little bit too specific to this type of network and its extension to other types of attentional architectures not straightforward.
> > >
> > > * I really appreciate the contribution about an updated version of the bound that considers statistical issues. This is exactly the kind of question I had in mind, and  the sketched proof and discussion seem reasonable.

---

> > > > ### Author Response · Authors · 2022-12-02
> > > > **Thanks**
> > > >
> > > > We're glad to hear that's the type of result you were looking for, we'll add it to appendix D. About your other comment - we understand your point, and we can acknowledge in the conclusion that generalizing attention masking to other architectures (specifically CNN-transformer hybrids) is a direction for future work.
> > > >
> > > > Thanks again for engaging!

---

### Author Response · Authors · 2022-11-15
**General response (1/2)**

We would like to thank the four reviewers for providing thorough feedback on our work and for recognizing the paper’s main contributions. Given the overall positive response, we want to focus on alleviating the concerns that were expressed. We believe that these questions/concerns in most cases require better clarification on our part, and we have attempted to improve the paper accordingly.

To briefly summarize, our paper is dedicated to efficiently calculating Shapley values for vision transformer (ViT) models. The two primary challenges in doing so are 1) properly handling held-out image patches and 2) overcoming the Shapley value’s exponential complexity, and we develop solutions for both problems. For the first, we develop an attention masking approach that lets the model better accommodate partial inputs. For the second, we propose learning an explainer model parameterized by a ViT, similar to FastSHAP but with several important innovations (practical and theoretical).

We will use this general response to address the concerns that appeared in multiple reviews, and we will use the individual reviewer responses to answer the remaining questions.

**Novelty.** Reviewers QpHf and trn3 were concerned that our approach has limited novelty (eEy2 and qY28 did not mention this concern), and QpHf specifically thought that our approach was a straightforward application of FastSHAP (Jethani et al, 2021). We do not believe this to be the case, and we can clarify that the biggest differences between the original FastSHAP work and ours are:

- We develop an attention masking approach to handle held out features with ViTs, and we demonstrate via experiments that it works better than other methods. Such an approach is only possible to develop in the context of transformer-based models, which Jethani et al. (2021) did not consider.
- In Jethani et al. (2021), the only theoretical justification for their learning-based Shapley value approximation is related to the loss function’s global optimizer. However, the optimal solution is unlikely to be attained in practice. Here, we advance our theoretical understanding of the learning-based approach by showing that the loss bounds the Shapley value estimation error even when the loss is suboptimal (see Theorem 1). This theoretical result is not straightforward to show (see Appendix D) and is one of our main contributions. It also applies even when using other architectures (non-ViTs).
- ViTs are notoriously difficult to train, so an important part of implementing ViT Shapley is figuring out how to optimize the explainer model effectively. This did not receive much consideration in Jethani et al. (2021), but we investigated several ways of doing so in the ViT context (see Appendix C). Our experiments showed that we get the best results by using a ViT as the explainer (versus a convolutional model like a U-Net), by fine-tuning an existing ViT rather than training from scratch, and by adding an additional self-attention layer.

We recognize that we should have highlighted the final point more prominently in the main text, and we have attempted to fix this. Section 5 now contains the following text: “Next, rather than training the explainer from scratch, we fine-tune an existing model that can be either the original classifier or a ViT pre-trained on a different supervised or self-supervised learning task (Touvron et al., 2021; He et al., 2021); ViTs are more difficult to train than convolutional networks, and we find that fine-tuning is important to train effectively (Table 3).” Note that Table 3 in the appendix compares our different explainer model configurations.

**Reproducibility.** Reviewers QpHf, eEY2 and trn3 mentioned that we had removed a link to our code and did not specify if our code would be released. We apologize for not mentioning this in the submission – our code is available in the supplementary materials, and it will definitely be released upon publication.

**Computational cost.** Several reviewers mentioned that the cost of training an explainer model is a downside to our approach. This is correct – we think it’s the main shortcoming of ViT Shapley, although not a critical one. A learned explainer model provides the benefit of very fast explanations, and more accurate ones than those obtained by other methods (see our experiments), but it does require investing time in training. Ultimately, we believe that the decision of whether to use ViT Shapley depends on one’s needs. If one requires high-quality explanations, fast explanations for a real-time deployment, or explanations for a large number of samples (e.g., an entire dataset), ViT Shapley is a compelling choice. If not, one may prefer to use a different method.

---

> ### Author Response · Authors · 2022-11-15
> **General response (2/2)**
>
> **Comparison with attention-based methods.** Reviewer QpHf was interested in whether we compared ViT Shapley with the method from Chefer et al. (2021), and reviewer trn3 was skeptical of whether model-agnostic methods can outperform ViT-specific methods (i.e., those that use attention values). First, we can confirm for QpHf  that we did compare to the method from Chefer et al. (2021) – it’s what we referred to as LRP in the experiments section. The comparison is straightforward because like ViT Shapley, this method is designed to output class-specific, patch-level importance values.
>
> Regarding trn3’s concern, our view is that there need not be a philosophical debate comparing model-specific and model-agnostic approaches: we can rely on objective metrics to test whether a given method correctly explains ViT predictions. Our experiments provide just such an evaluation, and the results are clear: pure attention-based methods like attention rollout do not work well, LRP (Chefer et al., 2021) which uses attention values differently provides a substantial improvement, but ViT Shapley provides the best performance overall. It's possible that researchers may one day develop a better attention-based explanation, but the evidence suggests that model-agnostic approaches like Shapley values can be highly effective.
>
> **Extension to NLP.** Reviewers eEy2 and qY28 asked about extending ViT Shapley to models for text data. We think this is a great idea and a natural follow-up to our work. However, the extension is non-trivial and that’s the main reason we didn’t explore it here. Specifically, transformers for NLP data typically operate on tokens representing word parts, but explanations should be provided on the word level. Thus, one must figure out how to set up the explainer to output word-level explanations corresponding to groups of tokens. (A concurrent ICLR submission didn't consider this, but we think it’s an important detail: https://openreview.net/forum?id=QcTbkoBycwk) We have ensured that this point is mentioned in our conclusion, which contains the following sentence: “Future directions involve extending ViT Shapley to transformer models in NLP, operating with arbitrary token groups or superpixels, and accelerating or otherwise improving the explainer model’s training.”
>
> **Size of datasets used for evaluation.** Reviewers QpHf and trn3 mentioned concerns about the datasets used in our evaluation. As for the number of datasets, we agree that it would be nice to have more, so we are currently working on adding one more dataset – we will share these results shortly. As for the size of the datasets, it’s worth pointing out that more data is generally a good thing: having more training data would make the explainer model more likely to succeed, not less, although the training cost would likely increase up to a certain point. Ultimately, we would be excited to try ViT Shapley on ImageNet, but this is beyond our current computational resources.
>
> Furthermore, it is perhaps more interesting to assess if a method can work well with moderate amounts of data, which is often the case in medical imaging applications for example. Our experiments show that ViT Shapley works well with our datasets containing roughly 10-40k examples, and we expect its performance would only improve with more data. (Note that Jethani et al. 2021 verified this in an ablation experiment for CNNs, see Figure 8 in their appendix.)

---

### Author Response · Authors · 2022-11-16
**New dataset**

In response to the request to increase our number of datasets, we’ve added one more. We checked which datasets were tested in the original ViT paper (Dosovitskiy et al., 2020) and added the Oxford pets dataset (Parkhi et al., 2012), which contains 37 classes and roughly 8k examples (it's available [here](https://www.robots.ox.ac.uk/~vgg/data/pets/)). The dataset is therefore larger in terms of classes, which is likely the more challenging dimension anyway (fewer samples per class should be harder).

For performance metrics, we replicated those shown in Tables 1 and 2 of our paper: insertion/deletion and faithfulness, calculated for the true class and averaged across the remaining classes. The results are shown below:

|                     | Target/Ins. (↑)   | Target/Del. (↓)   | Target/Faith. (↑)   | Non-target/Ins. (↑)   | Non-target/Del. (↓)  | Non-target/Faith. (↑)   |
|:--------------------|:----------------------|:----------------------|:------------------------|:----------------------|:----------------------|:------------------------|
| Attention last      | 0.886 (0.015)         | 0.504 (0.023)         | 0.545 (0.017)           | -                     | -                     | -                       |
| Attention rollout   | 0.848 (0.015)         | 0.670 (0.025)         | 0.559 (0.017)           | -                     | -                     | -                       |
| GradCAM (Attn)      | 0.844 (0.018)         | 0.749 (0.026)         | 0.527 (0.016)           | 0.005 (0.000)         | 0.008 (0.001)         | -0.499 (0.013)          |
| GradCAM (LN)        | 0.844 (0.015)         | 0.729 (0.028)         | 0.553 (0.017)           | 0.011 (0.001)         | 0.003 (0.000)         | -0.511 (0.014)          |
| IntGrad (Pixel)     | 0.904 (0.014)         | 0.809 (0.023)         | 0.437 (0.017)           | 0.006 (0.001)         | 0.003 (0.000)         | 0.126 (0.008)           |
| IntGrad (Embed.)    | 0.904 (0.014)         | 0.809 (0.023)         | 0.437 (0.017)           | 0.006 (0.001)         | 0.003 (0.000)         | 0.126 (0.008)           |
| Vanilla (Pixel)     | 0.871 (0.015)         | 0.602 (0.025)         | 0.559 (0.017)           | 0.004 (0.000)         | 0.011 (0.001)         | -0.526 (0.014)          |
| Vanilla (Embed.)    | 0.887 (0.015)         | 0.529 (0.026)         | 0.561 (0.017)           | 0.003 (0.000)         | 0.013 (0.001)         | -0.529 (0.014)          |
| SmoothGrad (Pixel)  | 0.898 (0.014)         | 0.491 (0.024)         | 0.564 (0.017)           | 0.003 (0.000)         | 0.014 (0.001)         | -0.532 (0.014)          |
| SmoothGrad (Embed.) | 0.845 (0.017)         | 0.846 (0.018)         | 0.558 (0.017)           | 0.004 (0.000)         | 0.004 (0.000)         | -0.526 (0.014)          |
| VarGrad (Pixel)     | 0.901 (0.014)         | 0.517 (0.024)         | 0.557 (0.017)           | 0.003 (0.000)         | 0.013 (0.001)         | -0.522 (0.014)          |
| VarGrad (Embed.)    | 0.856 (0.016)         | 0.843 (0.018)         | 0.555 (0.017)           | 0.004 (0.000)         | 0.004 (0.001)         | -0.523 (0.014)          |
| LRP                 | 0.898 (0.014)         | 0.479 (0.023)         | 0.567 (0.017)           | 0.004 (0.000)         | 0.013 (0.001)         | -0.529 (0.014)          |
| Leave-one-out       | 0.940 (0.007)         | 0.659 (0.033)         | 0.215 (0.036)           | 0.018 (0.002)         | 0.001 (0.000)         | 0.138 (0.023)           |
| RISE                | 0.948 (0.006)         | 0.510 (0.032)         | 0.559 (0.017)           | 0.032 (0.003)         | 0.001 (0.000)         | -0.523 (0.014)          |
| ViT Shapley         | 0.949 (0.008)         | 0.385 (0.024)         | 0.590 (0.016)           | 0.051 (0.003)         | 0.001 (0.000)         | 0.529 (0.012)           |
| Random              | 0.856 (0.015)         | 0.851 (0.016)         | -                       | 0.004 (0.000)         | 0.004 (0.000)         | -                       |

The results show that ViT Shapley provides the best performance, across all metrics and for both the true and non-true classes. We haven’t yet incorporated these results into the paper, but our setup was identical to the other datasets: we used ViT-Base, split the dataset and used test data only for calculating metrics, etc. This provides more evidence for ViT Shapley’s consistently strong performance.

---

### Decision · Program_Chairs · 2023-01-20

**Decision:**

Accept: notable-top-25%

**Justification For Why Not Higher Score:**

Reviewer concerns about the submission's proximity to work by Jethani et al.

**Justification For Why Not Lower Score:**

The submission is clear and the proposed idea is mathematically sound and well supported empirically. This is not a borderline submission.

**Metareview: Summary, Strengths And Weaknesses:**

The submission presents a new approach to estimating Shapley values for ViT. The approach is trained to amortize the optimization-based formulation of Shapley values using an auxiliary feedforward ViT network, which is initialized from a pre-trained model. The approach is evaluated on ImageNette and MURA and is compared against multiple attention-, gradient-, and removal-based approaches using multiple explanation quality metrics. ViT Shapley is shown to outperform those competing approaches.

The reviewer consensus is that the writing is clear and the proposed idea is mathematically sound and well supported empirically. Some reviewers remain concerned about the submission's proximity to work by Jethani et al., but overall they feel confident that its contributions outweigh those concerns. I recommend acceptance.

**Note From Pc:**

if the above contains the word "oral" or "spotlight" please see: "oral" presentation means -> notable-top-5% and "spotlight" means -> notable-top-25%. As stated in our emails, we are disassociating presentation type from AC recommendations